# Non-viral DNA delivery and TALEN editing correct the sickle cell mutation in hematopoietic stem cells

Arianna Moiani[1] ✉, Gil Letort [1], Sabrina Lizot[1], Anne Chalumeau [2], Chloe Foray[1], Tristan Felix [2], Diane Le Clerre[1], Sonal Temburni-Blake [3], Patrick Hong[3], Sophie Leduc [1], Noemie Pinard[1], Alan Marechal[1], Eduardo Seclen [3], Alex Boyne[3], Louisa Mayer [3], Robert Hong[3], Sylvain Pulicani [1], Roman Galetto[1], Agnès Gouble[1], Marina Cavazzana [4,5,6], Alexandre Juillerat [3], Annarita Miccio [2], Aymeric Duclert[1], Philippe Duchateau [1] & Julien Valton [1] ✉

Sickle cell disease is a devastating blood disorder that originates from a single point mutation in the *HBB* gene coding for hemoglobin. Here, we develop a GMP-compatible TALEN-mediated gene editing process enabling efficient *HBB* correction via a DNA repair template while minimizing risks associated with *HBB* inactivation. Comparing viral versus non-viral DNA repair template delivery in hematopoietic stem and progenitor cells in vitro, both strategies achieve comparable *HBB* correction and result in over 50% expression of normal adult hemoglobin in red blood cells without inducing β-thalassemic phenotype. In an immunodeficient female mouse model, transplanted cells edited with the non-viral strategy exhibit higher engraftment and gene correction levels compared to those edited with the viral strategy. Transcriptomic analysis reveals that non-viral DNA repair template delivery mitigates P53-mediated toxicity and preserves high levels of long-term hematopoietic stem cells. This work paves the way for TALEN-based autologous gene therapy for sickle cell disease.

Sickle cell disease (SCD) is one of the most common inherited diseases worldwide. SCD is characterized by mutations in the *HBB* gene, which encodes the β subunit of hemoglobin (Hb)[1]. The underlying pathophysiology of SCD stems from a single A > T point mutation in exon 1 of the *HBB* gene, which results in the sickle allele β$^s$ (NM_000518.4 (HBB):c.20 A > T), and the formation of the β$^s$-globin chain that is incorporated in sickle Hb (HbS)[2].

Today, most of the approved therapies for SCD remain symptomatic, are non-curative and impart significant side effects. The one notable exception is allogeneic hematopoietic stem cell transplantation (Allo-HSCT), a curative procedure that is unfortunately tarnished by substantial rates of morbidity and mortality[3,4]. Allo-HSCT requires HLA-matched HSPC donors and is reserved for patients with severe disease. The lack of an accessible cure for most patients and the high prevalence of SCD have motivated the development of alternative therapeutic approaches based on ex vivo hematopoietic stem and progenitor cell (HSPC) engineering.

Since the launch of lentiviral (LV)-based autologous gene therapy strategies[5–8], different approaches have sought to leverage engineered nucleases to treat SCD and overcome the genotoxicity risk associated

[1]Cellectis S.A., 8 Rue de la Croix Jarry, Paris, France. [2]Université Paris Cité, Imagine Institute, Laboratory of Chromatin and Gene Regulation During Development, INSERM UMR 1163, Paris, France. [3]Cellectis Inc., 430 East 29th Street, New York, NY, USA. [4]Biotherapy Clinical Investigation Center, Necker Children's Hospital, Assistance Publique Hopitaux de Paris, Paris, France. [5]Human Lymphohematopoiesis Laboratory, Imagine Institute, INSERM UMR1163, Paris Cité University, Paris, France. [6]Biotherapy Department, Necker Children's Hospital, Assistance Publique Hopitaux de Paris, Paris, France. ✉e-mail: arianna.moiani@cellectis.com; julien.valton@cellectis.com

with LV[9–11]. Some nuclease-based gene therapy approaches rely on generating double-strand break (DSB)-induced insertions and deletions (indels) at desired loci (BCL11A or HBG) to reactivate the expression of fetal hemoglobin (HbF) and rescue the missing function of HbA[12–15]. Although these approaches do not prevent the expression of HbS, they are promising and are currently being evaluated in a phase I clinical trial[5]. Other nuclease-based gene therapy strategies consist of exploiting homologous directed repair (HDR) to correct β$^s$ gene defects in the presence of a DNA repair template based on either adeno-associated virus 6 vectors (AAV6)[16,17] or single-stranded oligonucleotide (ssODN)[18–20]. This direct and precise correction of HBB represents an alternative approach with several potential advantages, notably that it promotes the expression of endogenous HbA rather than a surrogate with different physicochemical properties and oxygen affinity (HbF) and that it decreases the levels of HbS and its negative downstream effects.

While nuclease-mediated gene correction strategies are appealing, they are not devoid of challenges. One of these challenges lies in the method used to vectorize the DNA repair template. AAV6 is able to deliver large quantities of DNA to the nucleus and to elicit HDR-dependent gene correction in the presence of an engineered nuclease[21,22]. However, recent reports have raised safety and efficacy concerns over AAV6 related to potential genotoxicity[23] and the ability of AAV6 to mediate the integration of inverted terminal repeat (ITR) sequences at both on-target and off-target sites in the genome[24,25]. AAV6 has also been shown to activate the DNA damage response (DDR) and p53 pathways, which are associated with an impairment of hematopoietic stem cell (HSCs) fitness and engraftment capacity[25,26]. Further challenges lie in the choice of the engineered nuclease and the gene editing process used to promote β$^s$ gene correction. Depending on these two parameters, a substantial fraction of engineered HSPCs could end up harboring inactivated, rather than corrected, HBB alleles. Such an outcome could promote the development of a β-thalassemic phenotype[18] and reduce the proportion of corrected therapeutic cells in the final product. Thus, in light of these challenges, it will be vital to carefully select and optimize the gene editing tool, DNA repair template, and cell culture process to generate an efficient HSPC-derived gene therapy product to treat SCD.

Here we report a preclinical proof of concept of an autologous ex vivo gene therapy approach that leverages a TALE nuclease (TALEN) and a DNA repair template to precisely correct the β$^s$ gene responsible for SCD. We describe an optimized HSPC engineering protocol that results in high frequency of β$^s$ gene correction with a low frequency of β$^s$ gene knock out. This protocol promotes therapeutic HbA expression and phenotypic rescue of red blood cells (RBCs) derived from edited homozygous HbSS patients' HSPCs. Further characterization of edited RBCs by single-cell RNAseq analysis confirmed the generation of therapeutically relevant RBCs and low prevalence of a β-thalassemic phenotype, demonstrating the efficiency of our β$^s$ gene correction process. Finally, we provide a comprehensive characterization of the choice of the DNA delivery template by directly comparing AAV6 to ssODN in clinically relevant contexts and show evidence supporting the use of non-viral DNA template for efficient gene correction in LT-HSCs. The results described here provide the basis for the therapeutic development of a TALEN-based ex vivo gene therapy approach to treat SCD.

## Results

### TALEN-mediated gene editing at the HBB locus in mobilized HSPCs from healthy donors

With the aim of developing an autologous gene therapy strategy to treat SCD, we designed two TALEN: TALEN-HBB$_{ss}$ and TALEN-HBB$_{ββ}$, which are specific for the mutant and wild type versions of HBB exon 1, respectively (Supplementary Fig. 1a and Supplementary Table 1). We first tested TALEN-HBB$_{ββ}$ in HSPCs mobilized from healthy donors (HDs) to optimize the TALEN-mediated gene editing protocol and to select the most suitable DNA template delivery method to repair HBB via homologous directed repair (HDR).

As a first step, we sought to directly compare the gene editing efficiency obtained with viral and non-viral strategies based on either AAV6 or as ssODN. To do so, we designed two DNA repair templates containing the sickle-to-wild type mutation and additional silent mutations (to avoid TALEN-mediated cleavage of the corrected sequence) for use as templates for HBB gene modification in HDs or β$^s$ gene correction in HbSS patients' HSPCs (Supplementary Table 1). Gene editing efficiencies displayed in the following sections were assessed by digital droplet PCR (ddPCR) or AmpliconSeq.

We first optimized the viral and non-viral gene editing protocols by starting from previous reports[17,27] (Fig. 1a) and adjusted the doses and timing of TALEN-encoding mRNA and ssODN transfection and the doses of AAV6 necessary to elicit HBB gene correction. This first round of optimization generated up to 38.5% or 30.23% HBB gene correction with AAV6 and ssODN, respectively (Supplementary Fig. 1b–d). However, this correction was associated with a substantial frequency of insertions and deletions of 1 to multiple base pairs (indels) at the HBB locus and an HDR/indel ratio close to 1 with both approaches (Supplementary Fig. 1e). In an effort to develop a robust and clinically relevant gene editing protocol for ex vivo β$^s$ gene correction, we carried out a second round of optimization. This optimization was twofold. First, we sought to reduce the frequency of indels in the HBB coding gene to mitigate the risk of generating β-thalassemic cells[18] and to improve the therapeutic potential of the corrected cells. We then explored GMP-compatible culture media and electroporation buffers.

To that end, we co-delivered TALEN-HBB$_{ββ}$ and an mRNA named HDR-Enh01[28–30], which encodes an indirect NHEJ inhibitor, and observed a significant increase in HDR in both the ssODN and AAV6 protocols compared to the standard non-optimized protocol (Supplementary Fig. 2a, b, 34.3% ± 1.8% vs 22.6% ± 1.2% average HDR frequencies ± SD, respectively, for ssODN and 42.3% ± 6.1% vs 33.3% ± 6.7% average HDR frequencies ± SD, respectively, for AAV6). This effect was associated with a significant reduction in indels frequency (Supplementary Fig. 2a, b, 38.7% ± 3.9% vs 19.2% ± 2.0% average indel frequencies ± SD, respectively, for ssODN and 32.4% ± 2% vs 16.8% ± 2.1% average indel frequencies ± SD, respectively, for AAV6) and an increase in the HDR/indel ratio (Supplementary Fig. 2c). To anticipate the clinical development of this approach, we tested this optimized protocol including HDR-Enh01 in GMP-compatible conditions (GMP-compliant culture medium and electroporation buffer). Unexpectedly, we observed a drop in cell viability when using either AAV6 or ssODN (Supplementary Fig. 2d, 71.6% ± 0.7% and 39.6% ± 4.4% average frequencies of viability ± SD, respectively) under GMP-compatible conditions compared to standard R&D conditions. We thus further optimized our gene editing process, adding an mRNA named Via-Enh01, which encodes an antiapoptotic protein[31–33], to the TALEN and HDR-Enh01 mRNAs. The addition of Via-Enh01 led to a significant increase in HSPC viability compared to HDR-Enh01 alone or the standard protocol under GMP-compatible conditions (Supplementary Fig. 2d, 82.3% ± 2.3% vs. 39.6% ± 4.4% and 91.4% ± 1.1% vs. 71.6% ± 0.7% average frequencies of viability for ssODN and AAV6 ± SD, respectively). Importantly, the addition of Via-Enh01 contributed to the maintenance of high level of HDR frequency and HDR/indel ratio in GMP-compatible conditions in ssODN protocol (Supplementary Fig. 2e, f, 32.2% ± 3.8% vs 18% ± 6.3% HDR and 1.8 ± 0.2 vs 1 ± 0.3 average ratio ± SD in presence or absence of Via-Enh01, respectively), justifying the addition of both mRNAs to the final optimized protocol.

We then examined the efficiency and reproducibility of the optimized TALEN-mediated gene correction process. All the following experiments were carried out under R&D conditions. Our optimized process (including both HDR-Enh01 and Via-Enh01) was reproducible in nine different HDs and led to significantly higher HDR efficiencies and reduced indels frequencies compared to non-optimized

(standard) protocol, with no major differences between ssODN and AAV6 (Fig. 1b, 41.5% ± 5.3% vs. 46.5% ± 5.2% average frequency of HDR alleles ± SD, respectively and 19.1% ± 2.1% vs. 16% ± 1.4% average frequency of indel alleles ± SD, respectively). The HDR/indel ratio was significantly increased upon optimization with both ssODN and AAV6 (Fig. 1c, from 0.6 ± 0.1 to 2.2 ± 0.4 and from 1 ± 0.1 to 2.9 ± 0.5, average ratios ± SD, respectively).

In addition, in the context of both viral and non-viral DNA delivery methods, our optimized process did not affect HSPC viability (>95% average viable cells) or purity ( > 95% average CD34+ cells) assessed by flow cytometry 4 days after editing (Fig. 1d). Indeed, HSPC viability and purity were comparable between the optimized and standard processes, confirming the value of integrating HDR-Enh01 and Via-Enh01 into the editing protocol (Fig. 1d).

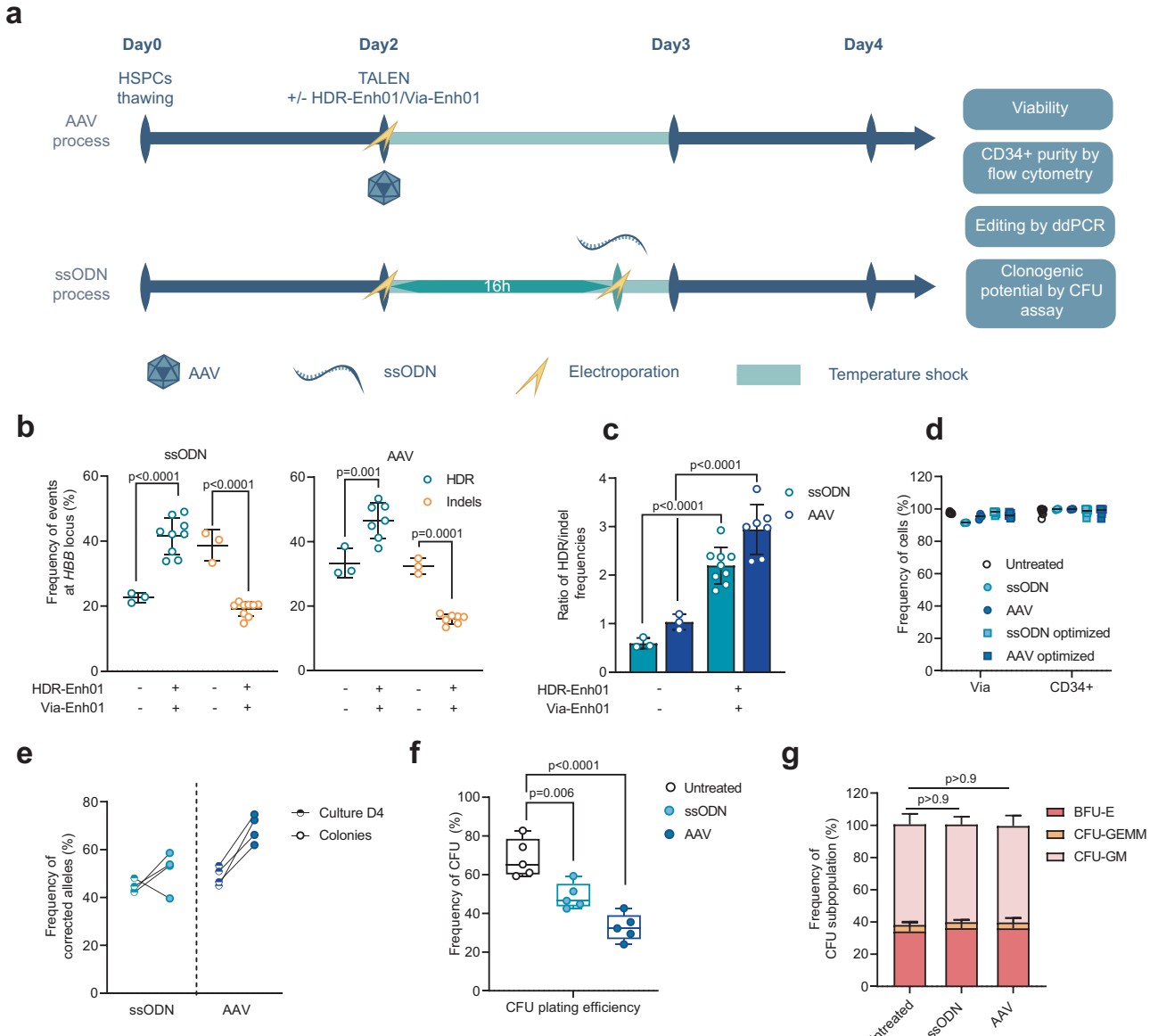

**Fig. 1 | Optimization of TALEN gene correction protocol at the *HBB* locus in HSPCs mobilized from healthy donors. a** Representative schema of viral (AAV) and non-viral (ssODN) editing protocols in HSPCs and experimental design. Frequency of homology-directed repair (HDR, blue) and insertion/deletion (indels, yellow) events (**b**) or HDR/Indel ratio (**c**) at the *HBB* locus measured at day 4 by ddPCR in PLX HSPCs edited with TALEN coupled with ssODN (left plot) or AAV (right plot) in the presence (+) or absence (−) of HDR-Enh01 and Via-Enh01. Data are expressed as average ± SD, for *n* = 3 (-), *n* = 9 (+, ssODN) and *n* = 7 ( + , AAV) biologically independent experiments and donors. A Kruskal-Wallis followed by Dunn's multi-comparison test. *P*-values are indicated. **d** Frequency of viable cells and CD34-expressing cells evaluated at day 4 by flow cytometry in ssODN- or AAV-edited cells, subject to the optimized (dark blue) or non-optimized (light blue) protocol, compared to untreated cells for up to 9 donors. **e** Frequency of HDR allelic events in BFU-E colonies derived from ssODN-edited (light blue) or AAV-edited (dark blue) HSPCs compared to HSPCs from culture at day 4. Data are expressed as average

frequency of corrected alleles, for cells edited by ssODN and AAV6, *n* = 4 biologically independent experiments and donors for each experimental group. **f** Box plots showing the plating efficiency, measured as the frequency of CFU colonies obtained 14 days after seeding, normalized for the number of seeded HSPCs, obtained for HSPCs edited by ssODN and AAV6 or untreated controls, *n* = 5 biologically independent experiments and donors for each experimental group. On each box plot, the central mark indicates the median, the bottom and top edges of the box indicate the interquartile range (IQR), and the whiskers represent the maximum and minimum data point. One-way ANOVA with Tukey's comparison test. *P*-values are indicated. **g** Average frequency ± SD of myeloid (CFU-GM), erythroid (BFU-E), and multipotential (CFU-GEMM) colonies comparing untreated, ssODN-edited, and AAV-edited HSPCs, *n* = 5 biologically independent experiments and donors for each experimental group. One-way ANOVA with Tukey's comparison test. *P*-values are indicated. Source data are provided as a Source data file.

Finally, our gene editing process achieved high frequency of HDR in burst-forming unit-erythroid (BFU-E) progenitors (Fig. 1e, 51.3% ± 7.1% and 68.7% ± 5.0% average frequency of HDR alleles for ssODN and AAV6, respectively), although the process resulted in a 2- and 1.4-fold decrease in clonogenic potential in AAV- and ssODN-edited HSPCs, respectively, compared to untreated control (Figs. 1f, 32.8% ± 6.3% and 48.8% ± 6.0% average frequency of CFU ± SD for AAV and ssODN edited HSPCs, respectively, vs 68.5% ± 8.8% average frequency of CFU ± SD for untreated). No lineage skewing was observed by CFU-C assays (Fig. 1f, g). Overall, these experiments demonstrated that an optimized TALEN-mediated editing protocol coupled with either viral or non-viral mediated DNA delivery can achieve highly efficient gene correction in clinically relevant HSPCs, while mitigating the risk of frequent *HBB* indels.

## Non-viral DNA delivery leads to high levels of HDR at the HBB locus in LT-HSCs

We next investigated whether the viral and non-viral DNA delivery strategies can edit LT-HSCs. We used plerixafor-mobilized (PLX) HSPCs from three healthy donors as a clinically relevant source of CD34+ cells and assessed the engraftment of edited LT-HSCs and hematopoietic reconstitution in Busulfan-conditioned immunodeficient NCG mice (Fig. 2a). Consistent with previous reports[22,34], ssODN-mediated editing led to higher levels of engraftment in bone marrow (BM) than did AAV6 at 16 weeks after injection (Fig. 2b, 58.2% vs 2.2% median of hCD45+ chimerism, respectively). Interestingly, one- or two-step electroporation (EP) did not affect the fitness of LT-HSCs as the engraftment levels were similar in untreated and EP controls. Likewise, engraftment of HSPCs edited with ssODN was similar to the control groups (Fig. 2b). Based on multilineage engraftment analysis by flow cytometry, the human graft mostly generated CD19 + B-cells and, to a lesser extent, myeloid and CD3 + T-cells (Fig. 2c). To assess allelic editing in engrafted LT-HSCs, we performed ddPCR on genomic DNA extracted from hCD45+ cells sorted from BM. Notably, ssODN performed significantly better than AAV6, reaching a median HDR allelic frequency of 30.5% vs 9.1%, respectively (Fig. 2d, left panel, 29.2% ± 8.7% vs 14.4% ± 14.8% average HDR frequency ± SD, respectively), and showing a smaller drop in HDR frequency compared to the in vitro input median frequency of 45% for ssODN vs. 50% for AAV6 (Fig. 2d, left panel, 45% ± 2.4% vs 50.2% ± 2.8% average frequency of HDR ± SD, respectively). No enrichment of indels was observed in either group between the in vitro input and in vivo output (Fig. 2d middle panel, 20.3% vs 18.4% and 16.8% vs 3.9% median or 20.6% ± 0.5% vs 20.6% ± 8% and 16.5% ± 0.6% vs 14.2% ± 18.2% average frequency of indels at input vs output for ssODN and AAV6, respectively). The ratio of HDR/indel did not change significantly for both groups (2.2 vs 1.6 and 3.0 vs 2.5 median frequency of indels at input vs output, for ssODN and AAV groups, respectively) although some outliers harboring ratio <0.5 were observed in the AAV group (Fig. 2d, right panels). In addition, ssODN resulted in similar editing compared to AAV6 in CFU derived from engrafted cells, and no major differences were observed between BFU-E and CFU-GM (Fig. 2e, 28.7% ± 8.1% vs 15.7% ± 5.0% average frequency of HDR alleles ± SD, respectively, in BFU-E and 27.8% ± 7.4% vs 25.6% ± 8.9% average frequency of HDR alleles ± SD, respectively, in CFU-GM).

In all, these data show that non-viral mediated DNA delivery outperformed viral-mediated editing both in terms of engraftment and HDR efficiency in clinically relevant cells (Fig. 2f).

## Non-viral DNA delivery maintains higher proportion of bona fide HSC than does viral DNA delivery

To better understand the discrepancies in engraftment and editing between viral and non-viral DNA template deliveries in vivo, we performed single-cell RNA sequencing (scRNAseq) on PLX-HSPCs used for in vivo injection. We characterized untreated (UT), ssODN, and AAV6 edited samples from two healthy donors multiplexed using hashtag

technology[35] to account for donor-to-donor variability (Fig. 3a). Using a droplet-based approach, we generated 5'scRNA-seq data from 15,554 cells, detecting a median of 4184 genes/cell.

To identify cell subpopulations, we used Azimuth[36-38], which enabled us to assign up to 97.2% cells to a specific HSPCs subset, leaving only 2.8% cells unassigned. We were able to identify within all samples a more primitive HSPCs subset, which we named HSC-enriched. The HSC-enriched subset, consisting of >95% cells in G0/G1 phase (Supplementary Fig. 3b), was validated by looking at the expression of common RNA markers previously described to be associated with primitive HSCs[39-45] (Fig. 3b, c and Supplementary Fig. 3a). Interestingly, we found a greater proportion of HSC-enriched cells in ssODN-edited cells than in AAV6-edited cells (Fig. 3b, c, 37.7% ± 5% vs 19.3% ± 1.6%, average frequency ± SD of HSC-enriched cells, respectively). In contrast, the proportion of differentiating progenitors such as lympho-myeloid primed progenitors (LMPPs) was increased in AAV6 compared to ssODN sample (Fig. 3b, c, 41.9% ± 1.1% vs 26.4% ± 2.0% average frequency ± SD of LMPP cells, respectively). This observation was reproducible within the two donors and was well correlated with engraftment levels obtained with the same samples, though it was not noticeable using standard flow cytometry analysis based on CD34 + /CD38-/CD90 + /CD133+ markers (Fig. 3d and Supplementary Fig. 3c, 0.917 and 0.089 $R^2$ correlation for scRNAseq and flow cytometry, respectively). These observations demonstrate the potential of scRNAseq to quantify the pool of primitive HSCs and the advantages of scRNAseq over conventional flow cytometry methods in detecting this subpopulation. No other HSPC subpopulation changed significantly between different samples, confirming that our editing process did not induce any lineage bias with the exception of the pool of primitive cells. Consistently with this first observation, we found a greater proportion of quiescent cells in ssODN-edited HSPCs compared to AAV6-edited HSPCs (Fig. 3e, f, 57.0% ± 1.9% vs 37.9% ± 3.6% average frequency of cells in G0/G1 phase ± SD, respectively). Unexpectedly, the UT cells had the lowest proportion of HSC-enriched and quiescent cells (Fig. 3e, f, 5% ± 0.4% frequency of HSC-enriched ± SD and 17.6% ± 0.3% frequency of cells in G0/G1 phase ± SD).

To better characterize the fitness of cells edited in the presence of AAV6 and ssODN, we investigated the transcriptional status of the edited HSC-enriched population (HSC) and progenitors (HPC) and compared it to UT cells. Consistent with published data[26], analysis of differentially expressed genes (DEGs) and gene set enrichment analysis (GSEA) revealed significant upregulation of genes belonging to the p53 pathway, the innate immune response, interferon (IFN) response, and inflammation after AAV6 and ssODN gene editing and at similar levels in HSC and HPC (Fig. 3g and Supplementary Fig. 3d, e). Interestingly, genes involved in p53 pathway activation were upregulated to a significantly higher level in AAV6- than in ssODN-edited cells (Fig. 3g). Consistent with this, AAV6-edited cells displayed the greatest proportion of cells expressing the p53 pathway among the two experimental groups. (Fig. 3g). On the other hand, ssODN-edited HSCs and HPCs showed a greater upregulation and a greater proportion of cells expressing genes involved in IFN response and inflammation (Fig. 3g and Supplementary Fig. 3e).

GSEA also showed several negatively enriched pathways in edited samples compared to UT. These pathways include oxidative stress, mitochondrial metabolism, protein synthesis, and the cell cycle (Supplementary Fig. 3e, f). Interestingly, these pathways were less enriched in the ssODN-edited group than in the AAV6-edited group and expressed at lower levels in HSCs than in HPCs (Fig. 3g and Supplementary Fig. 3e, f).

## Highly efficient TALEN-mediated correction of sickle mutation in non-mobilized HbSS patient HSPCs

To assess whether the optimized editing protocol described here would efficiently correct sickle-cell HSPCs, we purified HSPCs from

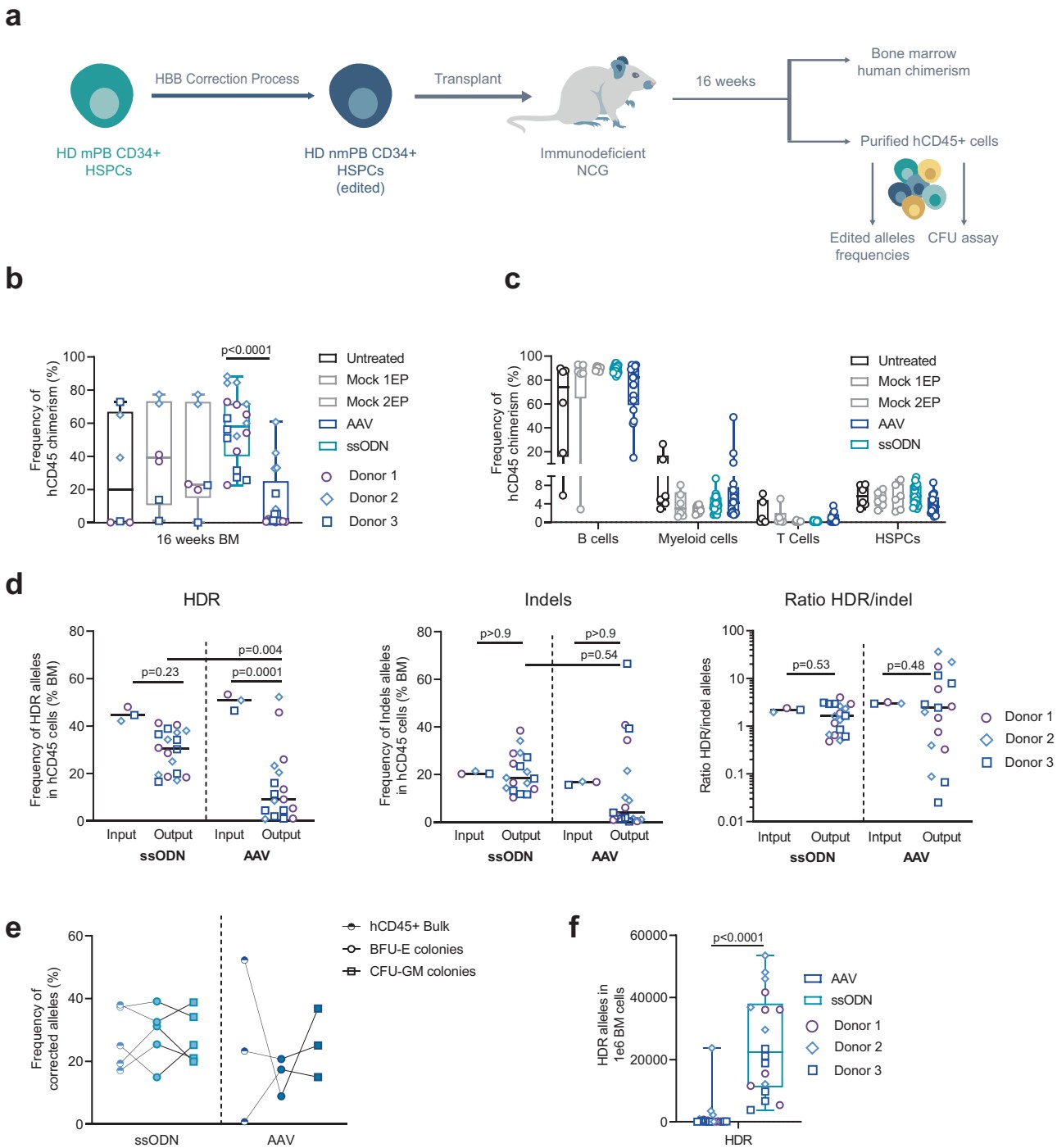

**Fig. 2 | Long-term engraftment and hematopoietic reconstitution of edited plerixafor-mobilized HSPCs from healthy donors in NCG mice. a** Representative schema and experimental design of xenotransplantation of HSPCs in NCG mice. **b** Human chimerism measured by flow cytometry at 16–18 weeks after injection. On each box plot, the central mark indicates the median value of chimerism ($n = 6$ mice per control group and $n = 18$ mice per edited group from a total of 3 HD HSPC donors), the bottom and top edges of the box indicate the interquartile range (IQR), and the whiskers represent the maximum and minimum data point. Two-way ANOVA followed by Tukey's multi-comparison test. *P*-values are indicated. **c** Multilineage engraftment in bone marrow (BM) assessed by flow cytometry 16–18 weeks after injection. Box plots parameters are defined as in (**b**). **d** Frequency of HDR (left plot) or indels (middle plot) and ratio of HDR/indel (left plot) alleles determined in hCD45+ cells obtained from BM 16–18 weeks after injection (Output) or in HSPCs before injection (Input). Black lines represent median values ($n = 18$ mice per edited group from a total of 3 HD HSPCs donors). Two-way ANOVA

followed by Bonferroni multi-comparison test. *P*-values are indicated. **e** Frequency of HDR allelic events in hCD45+ cells obtained from BM at 16–18 weeks after injection BFU-E and CFU-GM colonies derived from engrafted hCD45+ cells for ssODN-edited or AAV-edited groups. Data are expressed as frequency of corrected alleles, $n = 5$ and $n = 3$ biologically independent experiments and donors for cells edited by ssODN and AAV6, respectively. Black lines connect paired samples. **f** Number of HDR-edited alleles in $1 \times 10^6$ bone marrow cells obtained in each mouse, 16–18 weeks after engraftment. On each box plot, the central mark indicates the median value of HDR or indel-edited alleles in $1 \times 10^6$ BM cells ($n = 18$ mice per edited group from a total of 3 HD HSPC donors), the bottom and top edges of the box indicate the interquartile range (IQR), and the whiskers represent the maximum and minimum data point. Mann–Whitney two-tailed non-parametric unpaired test with a confidence interval of 95%. *P*-value is indicated. Source data are provided as a Source data file.

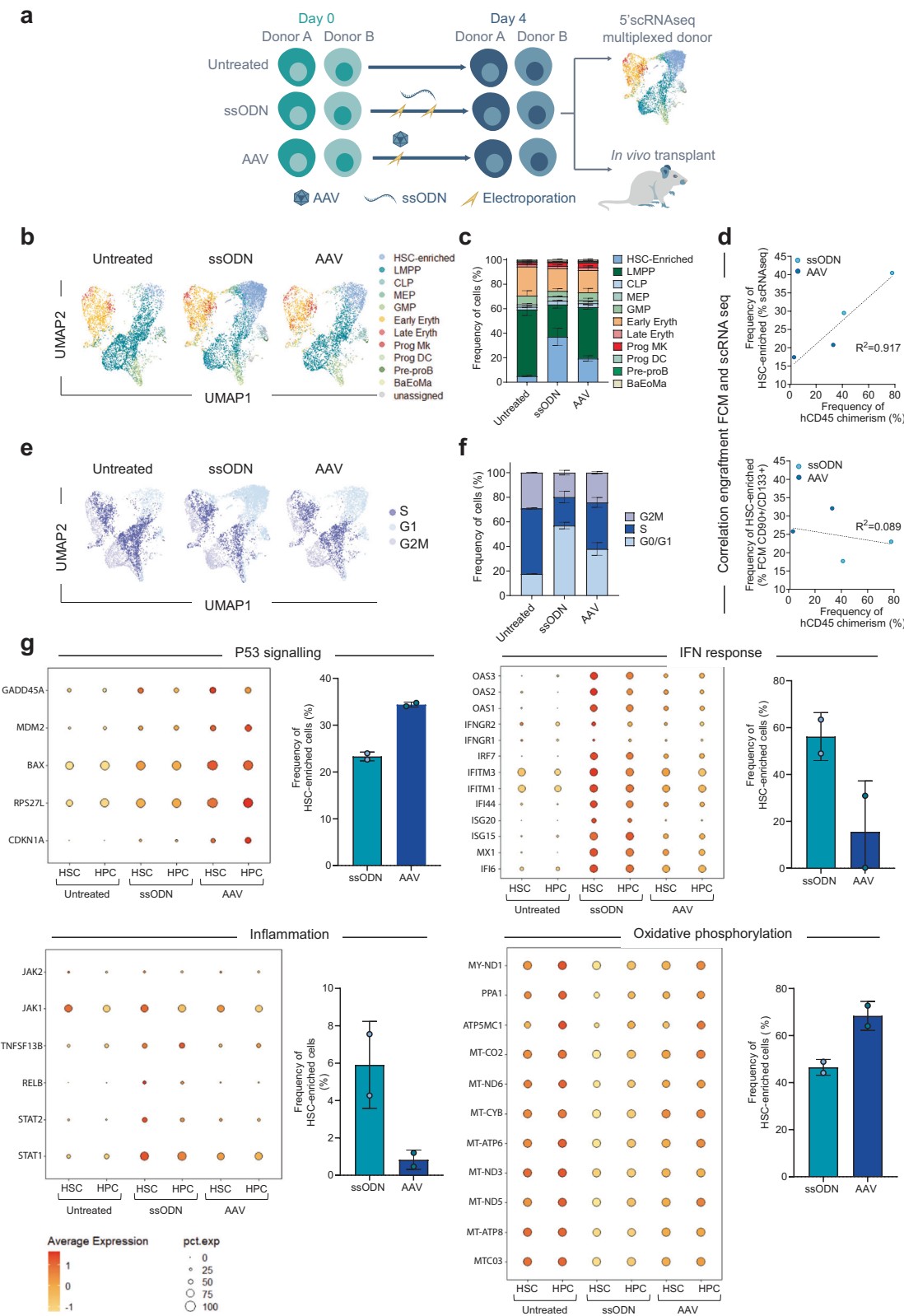

nine non-mobilized homozygous sickle patients (nmHbSS) and assessed the efficiency and reproducibility of gene editing using the TALEN-HBBss (Fig. 4a).

We achieved 44.3% ± 5% (ssODN) and 53.2% ± 7.7% (AAV6) average frequency of βˢ gene correction while maintaining a low frequency of indels (Figs. 4b, 21.4% ± 2.3% and 17.3% ± 3.1% average frequency of indels ± SD in ssODN and AAV6, respectively), high viability (>90%

viable cells), and high purity (>90% CD34+ cells) in all tested patient cells (Fig. 4a–d). These data confirmed the advantage of our optimized protocol compared to the standard counterpart (Fig. 4b, 29.6% ± 2.1% and 40.3% ± 8.4% average frequency of HDR ± SD, in ssODN and AAV6, respectively, and 40.3% ± 2.8% and 33.4% ± 2.5% average frequency of indels ± SD in ssODN and AAV6, respectively). Our protocol optimization significantly increased the HDR/indel ratio in HbSS HSPCs from

**Fig. 3 | Single-cell RNAseq characterization of the different editing protocols in plerixafor-mobilized HD HSPCs before in vivo injection. a** Representative schema and experimental design used for 5′scRNAseq analysis. **b** UMAP (Uniform Manifold Approximation and Projection) plot showing aggregated 5′scRNAseq data from non-edited and edited HSPCs analyzed the day of in vivo injection, $n = 2$ independent biological donors. Clusters and associated cell types (defined in the Methods section) are indicated by name and colors. **c** Average frequency ± SD of assigned cell types obtained out of n = 2 independent biological replicates coming from 2 donors. Two-way ANOVA followed by Bonferroni multi-comparison test. *P*-values are indicated. **d** Linear correlation between the frequency of HSC-enriched cells defined by scRNAseq data (upper plot) or by flow cytometry (FCM, lower plot) and the average frequency of hCD45+ chimerism obtained in BM 16–18 weeks after in vivo injection. Each dot represents data obtained from one HSPCs donor. For hCD45+ chimerism, the data from mice injected with the same donor were pooled, and the average frequency is shown. **e** UMAP plot showing aggregated 5′scRNAseq

data illustrating the clusters and associated cell cycle phases of non-edited and edited HSPCs analyzed the day of in vivo injection. **f** Average frequency ± SD of cells associated with each cell cycle phase. **e** and **f** were plotted out of $n = 2$ independent biological replicates from 2 donors. Two-way ANOVA followed by Bonferroni multi-comparison test. *P*-value is indicated. **g** DotPlots from scRNAseq data representing gene expression profiles for selected relevant genes belonging to p53, IFN response, Inflammation, and oxidative phosphorylation pathways (rows) among hematopoietic stem cells (HSC) and the rest of progenitor cells (HPC). The size and the color of each dot represent the percentage of cells expressing a given gene and the average expression of the gene, respectively. On the side of each DotPlot is illustrated the average frequency ± SD of HSC-enriched cells (as defined in Fig. 3c) expressing selected relevant genes belonging to the p53, IFN response, Inflammation, and Oxidative phosphorylation pathways, for $n = 2$ independent biological replicates from 2 HSPC donors. Source data are provided as a Source data file.

0.7 ± 0.1 and 1.2 ± 0.3 to 2.1 ± 0.2 and 3.2 ± 0.6 (average HDR/indel ratio ± SD) when mediated by ssODN and AAV6, respectively (Fig. 4c). ssODN did not lead to significant differences in progenitor cell viability and caused no lineage skewing compared to UT control as assessed by CFU-C assays (Fig. 4e, f). In contrast, AAV6 induced a significant decrease in clonogenic potential compared to ssODN and UT control (Fig. 4e). Our optimized protocol significantly reduced the proportion of HSPC-derived BFU-E clones harboring bi-allelic indels (less than 10% of all clones in either AAV6- or ssODN-based editing) and enabled reaching high frequency of corrected BFU-E clones (Fig. 4g, 58% and 75% in ssODN and AAV6, respectively). Overall, our gene editing process led to efficient and reproducible $\beta^S$ gene correction in nmHbSS HSPCs, elicited therapeutically relevant fraction of corrected clones, and mitigated bi-allelic $\beta^S$ gene inactivation.

### Sickle cell phenotype correction in fully mature red blood cells derived from edited HbSS patient HPSCs

To verify the therapeutic potential of edited nmHbSS HSPCs, we differentiated them into fully mature red blood cells (RBCs) and assessed Hb expression as well as the sickling properties of the edited samples (Fig. 5a). Because producing bi-allelic indels in the $\beta^s$ gene might generate β-thalassemic cells in treated patients, we generated a β-thalassemic-like control group (β-thal) in HSPCs, differentiated it and compared the properties of these cells to those of corrected and uncorrected RBCs. The β-thal control cells were generated using a TALEN protocol that promotes up to 80% *HBB* indels and results in 60% of clones harboring bi-allelic indels at *HBB* (Supplementary Fig. 4a).

Our results showed that while the enucleation process and the expression of early erythroid markers (CD36 and CD71) were similar in Mock EP control and in corrected RBCs, enucleation was altered in the β-thal control (Supplementary Fig. 4b). Reverse Phase-HPLC, and Cation Exchange-HPLC measurements showed clinically relevant HbA expression in all corrected RBC groups (Supplementary Fig. 4c, left panel 55.4% ± 5.5% and 59% ± 8.2% average frequency of HbA ± SD for ssODN and AAV6, respectively), with a substantial reduction in HbS expression compared to Mock EP control cells (Fig. 5b and Supplementary Fig. 4c, left panel, 24.5% ± 14.4%, 29.8% ± 7.7%, 93.6% ± 2.9% average frequency of HbS ± SD for ssODN, AAV6 and Mock EP control, respectively). Similar results were observed for pooled BFU-E colonies (Supplementary Fig. 4c, right panel). Most importantly, the α-globin/non-α-globin ratio in corrected RBCs was comparable to that of the Mock EP control but significantly different from that of the β-thal control, which was characterized by the reactivation of HbF, a high proportion of HbF+ cells, and a reduced level of total β-like globins (Fig. 5b and Supplementary Fig. 4d). Moreover, we observed alpha precipitates in the β-thal control group that were only barely detectable in the corrected RBCs (Supplementary Fig. 4e, 6.2% ± 0.6%, 1.4% ± 0.1% and 1% ± 0.1% average frequency of alpha precipitates ± SD for β-thal control, ssODN- and AAV6-edited samples, respectively). Furthermore, we found that the

proportion of apoptotic RBCs was higher in the β-thal control than in the Mock EP control (Supplementary Fig. 4f, 27.1% ± 2.7% vs 14.8% ± 0.4% average frequency of apoptotic cells ± SD, respectively) and low in the corrected RBCs (Supplementary Fig. 4f, 16.3% ± 0.9% and 18.7% ± 4.4% average frequency of apoptotic cells ± SD for ssODN and AAV, respectively). All the corrected RBCs showed significantly lower frequencies of sickle cells relative to the Mock EP control as assessed by sickling assays (Fig. 5c and Supplementary Fig. 4g, 96.2% ± 1.2%, 40.5% ± 14.1% and 41.8% ± 11.9% average frequency of sickling cells ± SD for Mock EP, ssODN and AAV samples, respectively).

We then performed scRNAseq analysis in differentiating RBCs to compare corrected RBCs to non-corrected RBCs and to the β-thal control. Using a droplet-based approach, we generated 5′ scRNAseq data from 48,280 erythroid cells at two time points to ensure detection of a β-thalassemic signature and were able to assess the genotype from the *HBB* RNA expression coupled with transcriptomic profile at the single cell level. First, we confirmed the low proportion of bi-allelic indels in ssODN- and AAV6-edited RBCs (Fig. 5d, e, 9% and 4.9% of cells, respectively) and a high proportion of corrected RBCs (57.4% and 66.3% respectively). This indicated that the high levels of gene correction observed in the HSPC before differentiation were maintained in RBCs. When assessing gene expression, we observed that bi-allelic indels found in the β-thal control led to a reduced expression of *HBB* and to an elevated expression of *HBG1* and *HBG2* (Supplementary Fig. 4h). In contrast, corrected and Mock EP control maintained normal levels of *HBB*, *HBG1*, and *HBG2* expression (Fig. 5f and Supplementary Fig. 4h). Bi-allelic indels were also associated with an upregulation of heat shock proteins (HSP), such as *HSPA1A*, which are typical signatures of β-thalassemic phenotype[46] as well as ribosomal protein genes and with a downregulation of heme synthesis genes (Fig. 5d, f). Interestingly, the cell population overexpressing *HSPA1A* was also observed in corrected RBCs but was restricted to cells harboring bi-allelic indels (Fig. 5d). Overall, the transcriptomic profile of the corrected RBCs was similar to the Mock EP and different from β-thal, suggesting that our editing protocol does not induce major transcriptional deregulation in RBCs (Fig. 5f).

Taken together, our results show that our optimized gene editing protocol promotes a high degree of phenotypic correction and mitigates the generation of β-thalassemic phenotypes or transcriptional changes in fully mature RBCs. Due to the comparable efficiencies and functional outcomes obtained with ssODN and AAV6 and the greater ability of ssODN to preserve edited LT-HSCs, we used ssODN for all subsequent experiments.

### Non-viral DNA gene editing leads to correction of the sickle mutation in HbSS patients-derived engrafted cells

We next tested whether corrected nmHbSS HSPCs could engraft long term and reconstitute hematopoiesis in immunodeficient NBSGW mice relative to a mock double-electroporated control (Fig. 6a). For

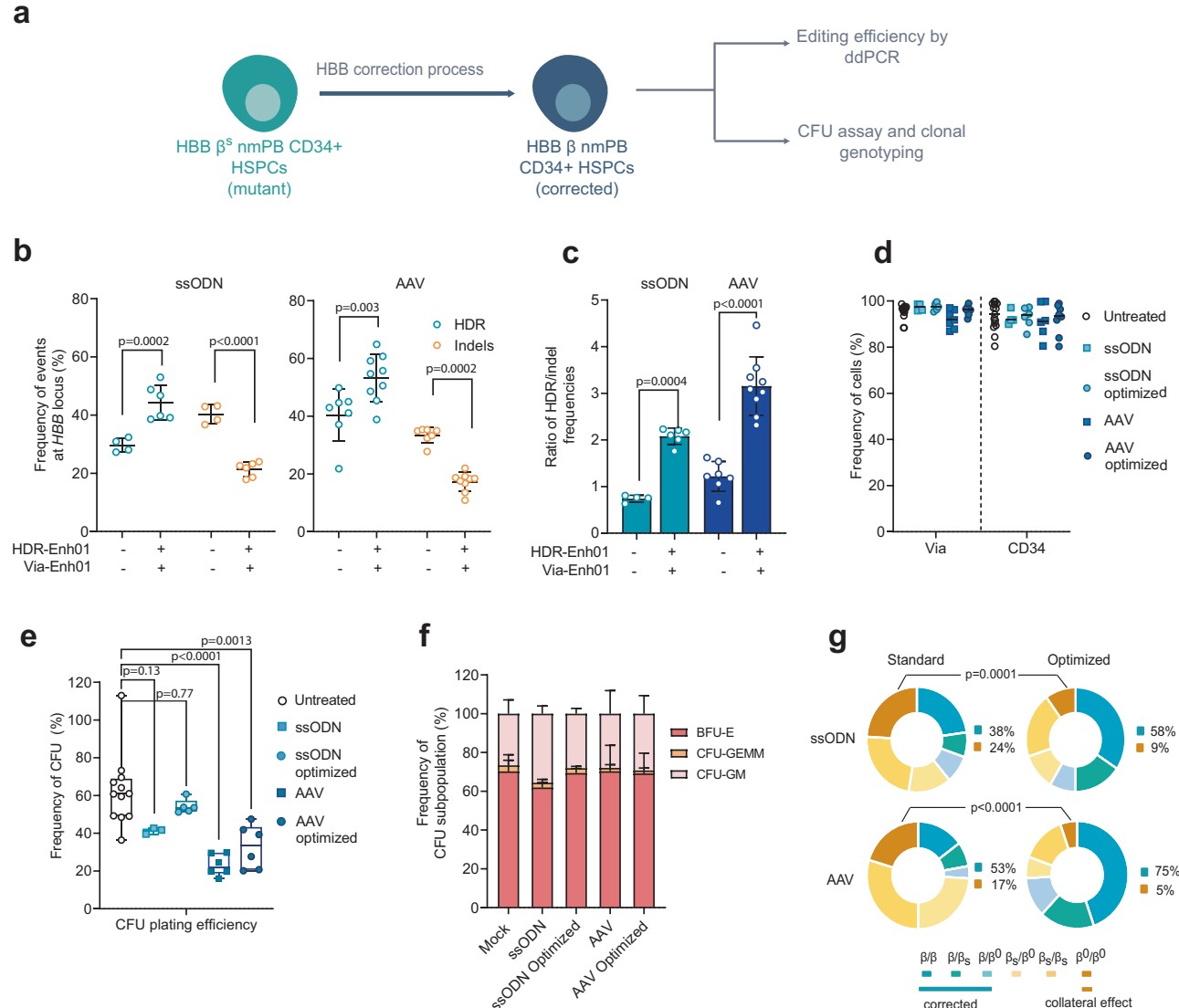

**Fig. 4 | TALEN gene correction protocol at the *HBB* locus in non-mobilized HSPCs from HbSS patients. a** Representative schema of gene correction assessment in HSPCs. Frequency of homology-directed repair (HDR) and insertion/deletion (indels) allelic events (**b**) or ratio HDR/indel (**c**) at *HBB* locus measured at day 4 by ddPCR in PLX HSPCs edited with TALEN and ssODN or AAV in presence (+) or absence (−) of HDR-Enh01 and Via-Enh01. Data are expressed as average ± SD for $n = 4$ (−, ssODN), $n = 7$ (−, AAV), $n = 6$ (+, ssODN) and $n = 9$ (+, AAV) biologically independent experiments and donors. Kruskal-Wallis followed by Dunn's multi-comparison test. *P*-values are indicated. **d** Frequency of viable cells and CD34-expressing cells evaluated at day 4 in HSPCs edited by ssODN optimized ($n = 6$) and non-optimized ($n = 4$) protocols or by AAV optimized ($n = 10$) and non-optimized ($n = 7$) protocols compared to untreated cells ($n = 15$). **e** CFU plating efficiency obtained for HSPCs edited by ssODN optimized ($n = 5$) and non-optimized ($n = 3$) protocols or by AAV optimized ($n = 6$) and non-optimized ($n = 6$) protocol compared to untreated cells ($n = 12$). On each box plot, the central mark indicates the median plating efficiency, the bottom and top edges of the box indicate the

interquartile range (IQR), and the whiskers represent the maximum and minimum data point. Each point represents one experiment performed with a given donor. One-way ANOVA with Tukey's comparison test. *P*-values are indicated. **f** Average frequency of CFU-GM, BFU-E and CFU-GEMM colonies ± SD obtained for untreated HPSCs ($n = 12$) or HSPCs edited by ssODN optimized ($n = 5$) or non-optimized ($n = 3$) protocols or AAV optimized ($n = 6$) or non-optimized ($n = 6$) protocols. One-way ANOVA with Tukey's comparison test. *P*-value are documented in the Source data file. **g** Frequency of allelic editing events detected in single BFU-E colonies enumerating bi-allelic genotype as $\beta$ = corrected allele, $\beta^s$ = sickle allele, $\beta^0$=indels allele. Data represent editing events detected in ssODN-edited groups (optimized and non-optimized protocols, $n = 6$ and $n = 4$ independent biological experiments and donors, respectively) and in AAV-edited groups (optimized and non-optimized protocols, $n = 6$ independent biological experiments and donors in both conditions). One-way ANOVA with Tukey's comparison test. *P*-values are indicated. Source data are provided as a Source data file.

this, we analyzed BM human chimerism and allele correction frequencies 16 to 18 weeks after transplantation. Corrected HbSS HSPCs showed robust engraftment capacity with a median human chimerism comparable to the mock-electroporated sample control (Fig. 6b, 23.2% vs 6.5% median or 30.5% ± 29.6% vs 16.2% ± 19.7% average frequency ± SD, respectively, Mann-Whitney non-parametric unpaired test, *p*-value is indicated). The human graft was multilineage and consisted of mostly lymphoid (CD19+ and CD3+) and myeloid cells (Fig. 6c, CD14+,

CD15+, and CD11b+). The cells engrafted in BM showed a relevant percentage of corrected alleles (Fig. 6d left panel, 22.2% vs 49.3% median frequency or 21.0% ± 18.6% vs 49.3% ± 4% average frequency ± SD of corrected alleles at output vs input) and no sign of enrichment for indels (Fig. 6d middle panel, 7.4% vs 23.7% median frequency or 13.5% ± 13.1% vs 23.5% ± 0.1% average frequency ± SD of indels alleles at output vs input), consistent with a stable ratio of HDR/indel between output and input conditions (Fig. 6d right panel). Overall, these data

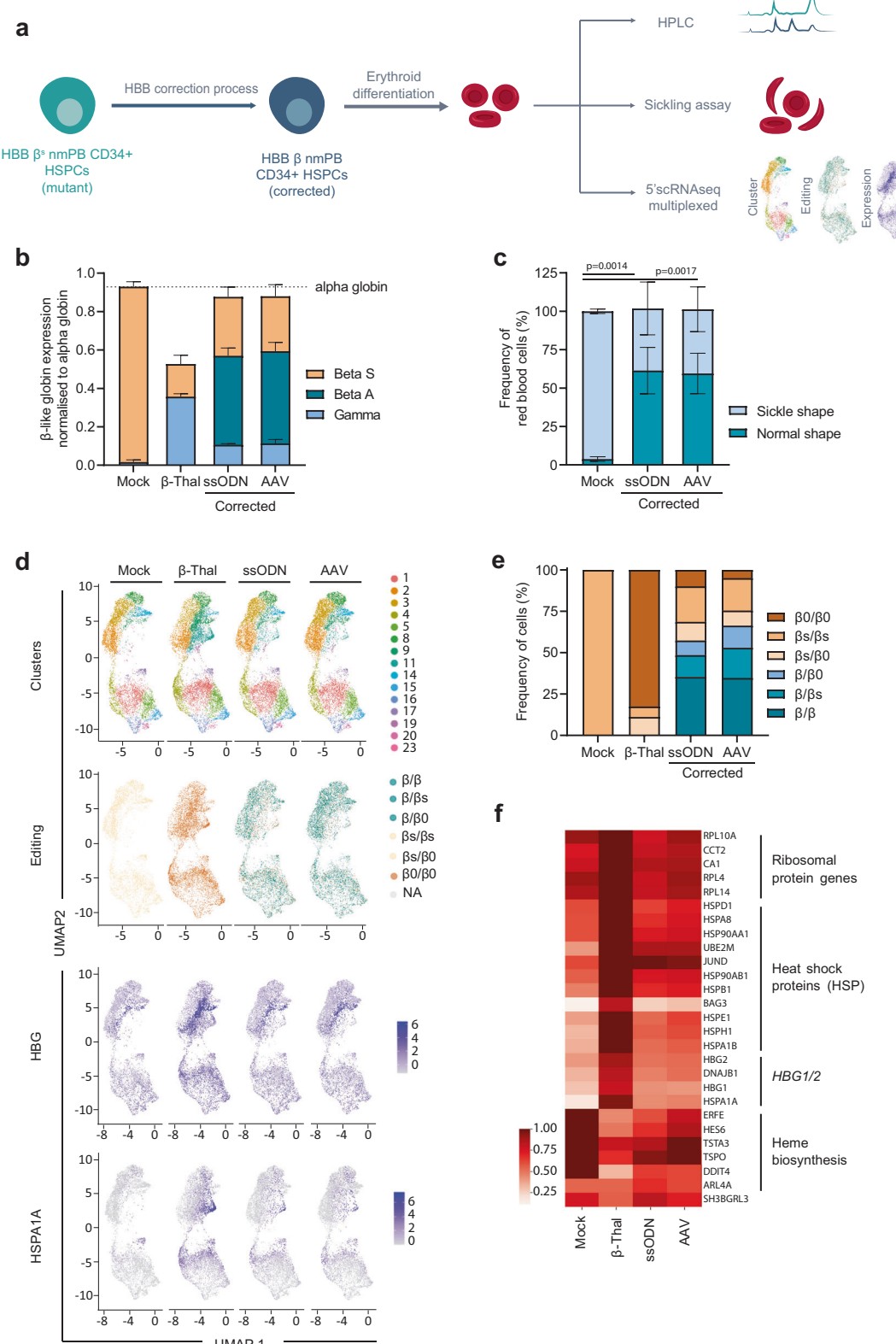

confirmed that our gene editing protocol can correct LT-HSCs derived from patients.

**The optimized TALEN editing protocol results in a low frequency of Off-target and adverse events in HSPCs in vitro and in vivo**
To examine the potential off-target cleavage at the highest dose of TALEN-HBB$_{ss}$, we analyzed edited primary T-cells from HbSS patients

with Oligo Capture Assays (OCAs), a genome-wide approach that has been previously described as unbiased[47] (Fig. 7a). This approach was able to confidently identify on-target sites (i.e., within the β$^s$ gene) and 192 putative off-target sites in the genome (Fig. 7b). We further quantitatively investigated the putative off-target sites using Sur-eSelect methodology[48] on HSPCs from 2 HbSS donors edited by TALEN-HBB$_{ss}$ (Fig. 7a, c, in the absence or presence of HDR-Enh01 and

**Fig. 5 | Correction of the sickle phenotype in erythroid cells differentiated from edited HSPCs. a** Representative schema of gene correction protocol in HSPCs followed by erythroid differentiation and experimental design. **b** (RP) HPLC quantification of globin chains in erythroid cells derived from non-mobilized HSPCs from HbSS patients corrected with ssODN or AAV and compared to Mock and β-Thal controls. β-Like globin chains are normalized to α-globin chains. Data are expressed as average ± SD, for $n = 3$ independent biological experiments and donors. **c** In vitro sickling assay measuring the proportion of sickled RBCs under hypoxic conditions (0% $O_2$). Plots represent the percentage of sickle- and normal-shape RBCs derived from Mock control, ssODN-edited, and AAV-edited cells. Data are expressed as average ± SD, for $n = 3$ independent biological experiments and donors. One-way ANOVA with Tukey's comparison test. *P*-values are indicated. **d** UMAP (Uniform Manifold Approximation and Projection) plot comprising aggregated 5'scRNAseq data from differentiated erythroid cells analyzed 11- and 13-

days post differentiation onset. Data represent the ssODN-edited and AAV-edited samples compared to Mock and β-Thal controls. Clusters determined by Louvain unsupervised clustering are indicated by different colors (Clusters), clusters representing *HBB* allelic genotype as determined by expressed *HBB* mRNA are indicated by different colors (Editing). Normalized expression of *HBG* and *HSPA1A* mRNAs are shown. The color gradient indicates the average expression of the indicated gene in each cell according to the scale shown. **e** Frequency of erythroid differentiated cells harboring each bi-allelic genotype as assigned in scRNAseq data for Mock, β-Thal, ssODN-edited, and AAV-edited samples. β = corrected allele, $β_s$ = sickle allele, $β^0$ = Indel allele. **f** Heatmap showing expression (normalized expression values log transformed) of the top deregulated genes comparing Mock, β-Thal, ssODN-, and AAV-edited samples. General categories to which a list of genes belong are shown. Source data are provided as a Source data file.

Via-Enh01). This analysis confirmed only one off-target site, located in the *HBD* gene. TALEN-HBB$_{ss}$ cleavage activity at the *HBD* gene in HbSS patients' HSPCs was low compared to the on-site cleavage activity (Fig. 7c, left panel, 54.2% ± 6.3% vs 1.1% ± 0.3% average frequency of indels ± SD at the *HBB* on-site and the HBD off-site, respectively, in the absence of HDR-Enh01 and Via-Enh01). Detailed analysis of the molecular events generated at the HBB on-site enabled to decipher the contribution of NHEJ- and microhomology mediated end joining- (MMEJ) events to the overall indels pattern illustrated in Fig. 7c (Fig. 7c, middle and right panels). These results showed that the presence of HDR-Enh01 and Via-Enh01 skewed the NHEJ/MMEJ ratio from 1.8 ± 0.3 to 0.8 ± 0.2 (average ratio ± SD), a trend consistent with ref. 28.

Simultaneous cleavage of *HBB* and *HBD* genes is known to elicit genetic rearrangements[49,50]. We further confirm that aspect in an unbiased and qualitative fashion by using the CAST-Seq methodology[49] on HSPCs from 3 HbSS patients, edited by TALEN-HBB$_{ss}$ in the presence of HDR-Enh01 and Via-Enh01. Our results showed that such treatment elicited deletion, inversion, and translocation events between the HBB on-site and HBD off-site, without promoting any detectable rearrangement in other part of the genome (Fig. 7d).

We then quantitatively assessed the frequency of such rearrangements by ddPCR using gDNA from HbSS patients' HSPCs edited by TALEN-HBB$_{ss}$, in the presence or absence of ssODN, Via-Enh01, and HDR-Enh01 (Fig. 7e–g). TALEN cleavage activity promoted formation of deletions, inversions, and translocations between *HBB* and *HBD* with frequencies of 1.3% ± 0.4% (deletion), 0.33% ± 0.04% (inversion) and 0.91% ± 0.71% (translocation), respectively (Fig. 7g, average frequency of events ± SD, obtained at day-4, in the absence of Via-Enh01 and HDR-Enh01). The frequency of genomic rearrangements was found either unchanged or decreased by the presence of Via-Enh01 and HDR-Enh01, and this pattern was observed in the presence or in the absence of the ssODN repair template (Fig. 7g). In addition, it significantly decreased from culture samples to those obtained from BM of mice, 16 weeks after onset of edited HSPC infusion (Fig. 7g). Similar conclusions were reached with HD HSPCs or HbSS patients' HSPCs edited by our viral gene correction process (Supplementary Fig. 5). This suggested a negative selection for adverse events in LT-HSCs (Fig. 7g, and Supplementary Fig. 5, compare day-4 and day-7 to WK16 dataset).

Nuclease-mediated genome editing has been recently found to promote losses and gains of heterozygosity in primary cells[51–56]. We thus sought to qualitatively assess that aspect in HbSS patients' edited cells. For that purpose, HSPCs from 2 HbSS patients were edited by our optimized non-viral gene editing process (TALEN-HBB$_{ss}$, ssODN, Via-Enh01, and HDR-Enh01) and analyzed by InferCNV scRNAseq (Fig. 8). This analysis was performed on edited cells obtained before or after differentiation into RBCs (4- and 15-days post thawing) and compared to mock cells controls. InferCNV plots showed no detectable loss or gain of heterozygosity compared the mock cells at the two time points (Fig. 8c), in contrast with the InferCNV positive control[53] showing a loss of chromosome 14 induced by TRAC-specific CRISPR-CAS9 treatment

of T-cells (Fig. 8d). Similar conclusions were reached with HbSS patients' HSPCs edited by our optimized viral gene editing process (TALEN-HBB$_{ss}$, AAV, Via-Enh01 and HDR-Enh01, Supplementary Fig. 6).

Nuclease mediated genome editing was also reported to promote enrichment of *p53*-deficient clones in edited cell population after several rounds of divisions[57–59]. While this phenomenon has been mostly characterized in immortalized hRPE1, hPSC, hiPS, and hES cell lines, we believe it was important to verify that aspect in the edited primary HSPCs generated in our study, given their potential for clinical applications. To address this question, we sought to characterized HSPCs edited by our non-viral optimized engineering process incorporating HDR-Enh01 and Via-Enh01, by high throughput sequencing of their *p53* exome (Fig. 8e). Because potential expansion and enrichment of *p53*-deficient clones may arise after several cell divisions, we compared the *p53* exome of untreated HSPCs to the ones obtained for successfully edited HSPCs gathered 4 days post editing and 16 weeks post NCG mouse engraftment onset. Our *p53* exome sequencing results obtained with 3 HSPC donors didn't show any detectable enrichment of common driver mutations between untreated and edited conditions, using the recommended detection threshold of 1% (Fig. 8e). Of note, a single nucleotide *p53* variant was identified in the mock and edited experimental groups of donors 1 and 3. It corresponds to a natural variant frequently identified in healthy donors (rs1042522, worldwide median prevalence of 66.4%, Supplementary Fig. 7). Similar results were obtained with healthy donor HSPCs successfully edited by our GMP-compatible viral and non-viral processes in the presence or in the absence of HDR-Enh01 and Via-Enh01 and cultivated in vitro for a total of 7 days (Supplementary Fig. 7). Altogether, our results suggest that the TALEN-mediated non-viral gene correction treatment incorporating HDR-Enh01 and Via-Enh01, neither generates detectable loss or gain of heterozygosity nor promotes detectable *p53* driver mutation enrichment in HSPCs.

## Discussion

In this study, we developed a TALEN-based gene editing process that enables highly efficient gene correction of the sickle cell mutation in patient-derived HSPCs. This correction translates into clinically relevant HbA expression and correction of sickle phenotype in mature RBCs. We optimized our protocol to prevent bi-allelic indels in *HBB* and to mitigate the risk associated with *HBB* inactivation. We also carried out a comprehensive side-by-side comparison of viral and non-viral HDR template delivery platforms and revealed that while both approaches led to a similar efficiency of $β^s$ gene correction, they display distinct transcriptomic patterns associated with sharp differences of in vivo engraftment of edited HSPCs. The non-viral HDR template delivery method mitigated p53 activation and better preserved the pool of HSC-enriched cells compared to its viral counterparts. This delivery method enabled the maintenance of gene correction after in vivo engraftment without any evidence for positive selection of indels or adverse events at the *HBB* locus.

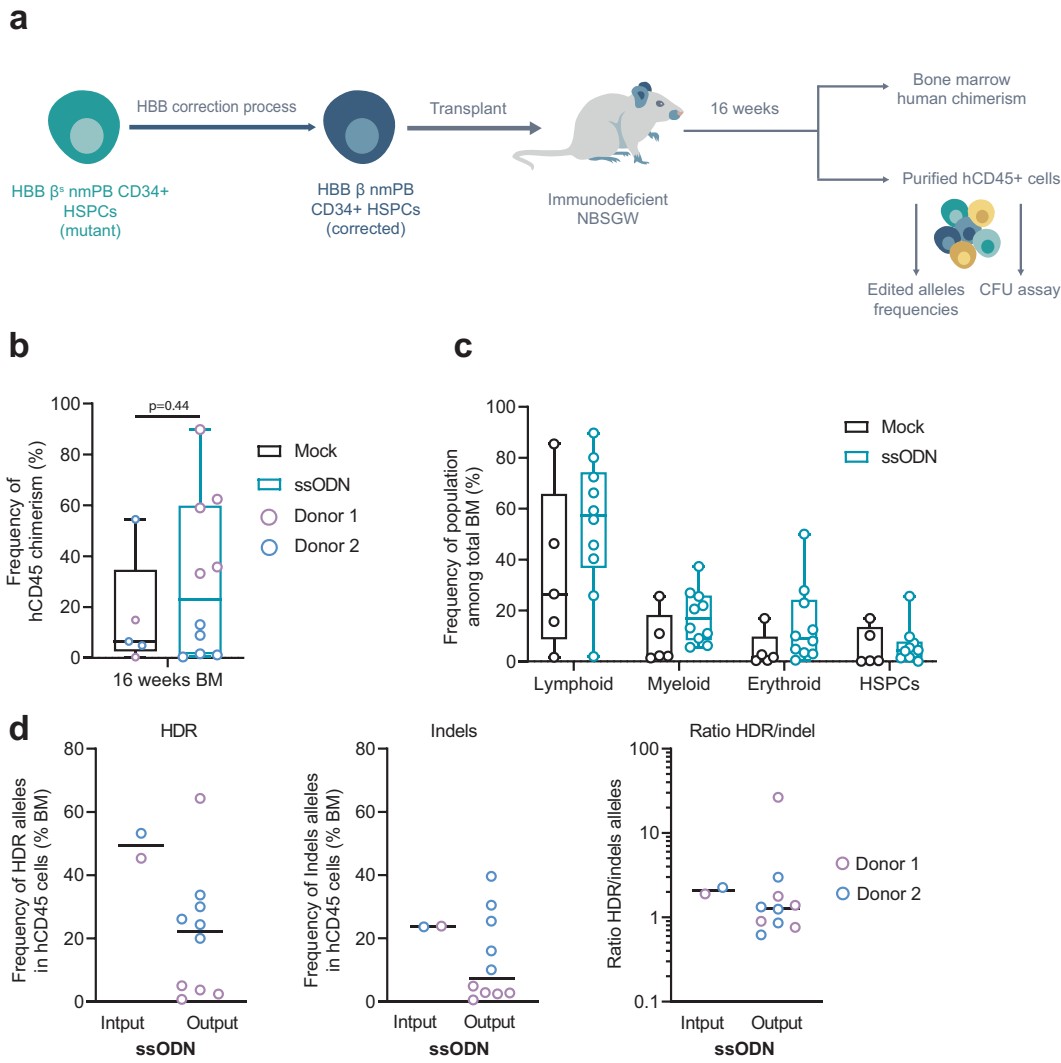

**Fig. 6 | Long-term engraftment and hematopoietic reconstitution of edited non-mobilized HSPCs from HbSS patients in NBSGW mice. a** Representative schema and experimental design of xenotransplantation of HSPCs in NBSGW mice. **b** Human chimerism assessed as hCD45+ cells in bone marrow (BM) by flow cytometry at 16–18 weeks after injection. Mock electroporated and ssODN-edited HSPCs groups are shown. Lines represent median values ($n = 5$ mice per control group and $n = 10$ mice per edited group from a total of 2 HbSS donors). Mann-Whitney two-tailed non-parametric unpaired test with a confidence interval of 95%. *P*-value is indicated. **c** Multilineage engraftment in bone marrow (BM) assessed by flow cytometry 16–18 weeks after injection to evaluate the frequency of B cells (CD19+), Myeloid cells (CD11+, CD14+, C15+), T-cells (CD3+), or HSPCs (CD34+)

among the graft (hCD45+) gathered from mock and ssODN experimental groups ($n = 5$ mice and $n = 10$ mice, respectively, generated from a total of 2 HbSS donors). On each box plot illustrated in B and C, the central mark indicates the median frequency, the bottom and top edges of the box indicate the interquartile range (IQR), and the whiskers represent the maximum and minimum data point. **d** Frequency of HDR (left plot), Indel alleles (middle plot), and ratio HDR/indel alleles (right plot) evaluated in gDNA from hCD45+ cells obtained from BM 16–18 weeks after injection (Output) or in gDNA obtained from HSPCs before injection (Input). Black lines represent median values ($n = 10$ mice per edited group from a total of 2 HbSS donors). Source data are provided as a Source data file.

Most nuclease-mediated gene correction approaches rely on generating a DSB at the targeted locus. In the context of β$^s$ gene correction in HSPCs, these approaches bear the risk of generating LT-HSCs with bi-allelic indels, a genotype potentially associated with a β-thalassemic phenotype[18]. In addition, these approaches may generate cells harboring large deletions or unwanted genomic adverse events[60]. Therefore, increasing the ratio of HDR/indel is of the utmost importance for β$^s$ gene correction, not only to achieve a safer profile, but also to increase the therapeutic potency of edited HSPCs. With this in mind, we incorporated in our gene correction protocol a genetically encoded inhibitor of the NHEJ pathway named HDR-Enh01[28–30]. This addition significantly increased the frequency of β$^s$ gene correction and reduced the frequency of bi-allelic indel clones to less than 10%. The efficient mitigation of *HBB* inactivation is consistent with a very recent study published during the revision of our manuscript, that reported a

similar trend using the same inhibitor delivered as peptide, along with CRISPR-CAS9 and viral-DNA template to correct *HBB*[61].

Using HDR-Enh01 as NHEJ inhibitor is not the only strategy to improve the ratio of HDR/indel during gene correction/insertion processes. Indeed, several encouraging approaches, using small molecules[62–64], engineered nucleases fused to NHEJ inhibitor protein domains[65,66] or temporal control of DNA repair[67,68] where recently reported to achieve this goal to variable extents in cell lines, primary differentiated cells and HSPCs. In the context of our study, the inhibition of NHEJ-mediated inactivation of *HBB* by HDR-Enh01 presents an advantage over DSB-based *HBB* correction strategies reported earlier[16,18,19,34,69,70]. Actually, multiple former studies performed with different engineered nucleases specific for the *HBB* locus (CRISPR-CAS9, and Zinc Finger nuclease) and different *HBB* repair templates (AAV, IDLV and ssODN), report a wide range of absolute frequencies

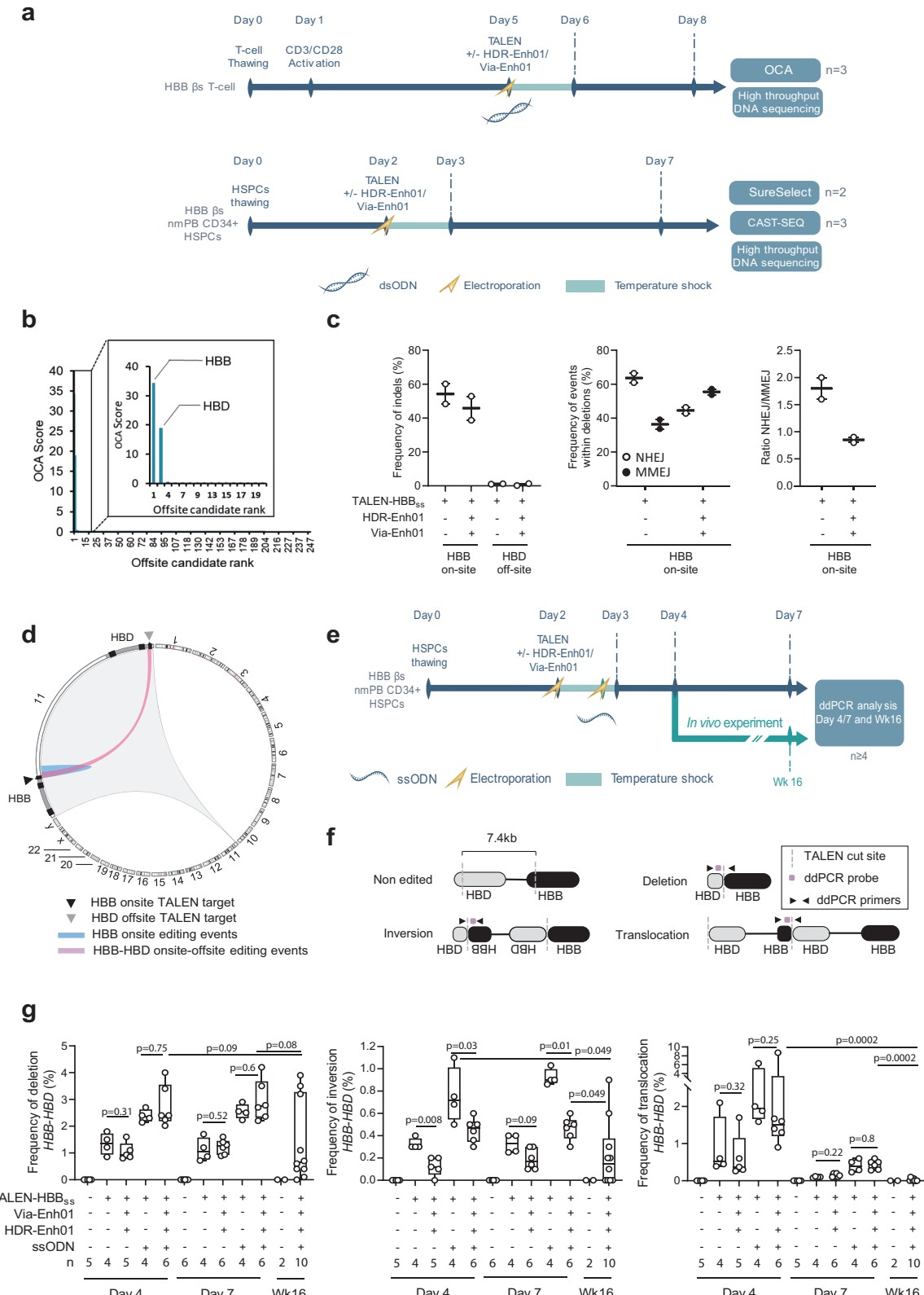

of HDR and indels at the *HBB* locus (from 10% to 65–70%, and from 10% to 60%, respectively) and HDR/indel ratio <1 in most of the cases[16,18,19,34,69,70]. Together, these earlier works indicate that the nature of the engineered nuclease, the DNA repair template, the handling of the cells and the timing of the gene editing process, markedly influence the therapeutic potential as well as the safety of edited HSPCs.

Nevertheless, we further characterized the phenotypic outcome associated with the low frequency of *HBB* indels and compared it to a β-thalassemic control in fully differentiated RBCs. Using a comprehensive set of analytical read out including flow cytometry, HPLC and 5'scRNAseq analysis, we confirmed that the high frequency of *HBB* indels present in the β-thalassemic control led to an impaired α-/non-α-globin chain ratio, reactivation of HbF and elevated apoptosis in

**Fig. 7 | TALEN off-target sites analysis and genomic rearrangements after editing at *HBB* locus. a** Representative schema to identify on-target and off-target TALEN-HBB$_{ss}$ activity in HbSS patient T-cells by OCA, to quantify them in non-mobilized HSPCs from HbSS patients by SureSelect and assess their consequence on the genomic integrity of HSPCs from HbSS patients by CAST-SEQ. **b** Average OCA score obtained for the on-site (*HBB*) as well as the 247 first candidate off-sites (*n* = 3 independent biological experiments and donors). OCA score for the first 20 candidates showing the on-target (*HBB*) and the unique off-target site identified (*HBD*) are shown enlarged in a separate box. **c** Left panel, frequency of Indels obtained by high-throughput DNA sequencing of TALEN-HBB$_{ss}$ on-target and off-target sites identified by OCA, in presence (+) or absence (−) of HDR-Enh01 and Via-Enh01. Middle and right panels, frequencies of NHEJ- and MMEJ-dependent deletions and ratio of NHEJ- over MMEJ-dependent events, respectively, detected and computed within all deletion events obtained in presence (+) or absence (−) of HDR-Enh01 and Via-Enh01. Each point represents the frequency of indels obtained in one independent experiment performed with a given HSPC donor (*n* = 2 independent biological experiments and donors, 5 technical replicates). **d** Circos plot representing the editing events occurring at *HBB* on site and between *HBB* on-site and *HBD* off-site, obtained by CAST-SEQ (*n* = 3 independent biological experiments and donors). **e** Representative schema of the experience performed to characterize the genomic rearrangements occurring at the *HBB-HBD* locus by ddPCR. **f** Experimental ddPCR strategy to detect deletion, inversion and translocation events at the *HBB-HBD* locus. **g** Frequencies of deletions (left panel), inversions (middle panel), translocations (right panel) at the *HBB-HBD* locus assessed by ddPCR. On each box plot, the central mark indicates the median value, the bottom and top edges of the box indicate the interquartile range (IQR), and the whiskers represent the maximum and minimum data point. The number of biological replicates performed with a given HbSS donor is documented at the bottom of each plot. Mann–Whitney non-parametric two-tailed unpaired test. *P*-value are indicated. Source data are provided as a Source data file.

RBCs. This pattern was associated with upregulated expression of β-thalassemic transcriptomic signatures as well as signs of delayed erythropoiesis. In stark contrast, these phenotypic hallmarks of β-thalassemia were significantly reduced in the corrected RBCs. These results confirmed the importance of preventing bi-allelic *HBB* indels to ensure correct erythropoiesis and to increase the overall therapeutic potential of the final gene edited product.

We also confirmed the high specificity of the TALEN nuclease activity, detecting only one off-target site at the *HBD* locus. Due to its high similarity to the *HBB* locus, this off-site cleavage event was expected. However, despite this high similarity, less than 1% of *HBD* alleles displayed indels. The proximity of *HBB* and *HBD* cleavage sites on chromosome 11 promoted the formation of genomic rearrangements including deletions, inversions, and translocations between *HBB* and *HBD* as previously reported[50]. However, those events occurred at low frequencies in HSPCs and were negatively selected in engrafted cells recovered 16 weeks after NCG transplantation onset. Furthermore, we didn't detect any enrichment of *p53* driver mutation as well as loss or gain of heterozygosity in edited cells, suggesting a favorable safety profile of our optimized TALEN-mediated β$^s$ gene correction process in HSPCs from HbSS patients. Nevertheless, further work assessing tumorigenicity in vivo is needed to thoroughly assess the potential genotoxic risk borne by genomic rearrangements in LT-HSCs.

To ensure lifelong treatment of a genetic blood disorder, a therapeutic gene editing approach must correct a sufficient number of LT-HSCs. We demonstrated that our TALEN-based editing protocol led to successful gene correction in progenitor cells while preserving high cell viability and in vivo long-term engraftment capacity. Substantial numbers of HSCs are known to be lost during transplantation due to apoptotic cell death controlled by proteins from the Bcl-2 family and a the lack of signals derived from the stem cell niche[71]. The transient overexpression of Via-Enh01 in our protocol resulted in increased viability of edited progenitors and potentially preserved the engraftment capability of edited HSCs as previously reported[31–33].

Nonetheless, we observed different engraftment outcomes in edited cells when using viral and non-viral DNA template deliveries, with a significant reduction in engrafted edited cells when using AAV6. These discrepancies have already been reported[22,34], but a thorough assessment of their root cause has not. One recent study highlighted the importance of the choice of DNA template delivery method[25]. This study showed that AAV6-mediated gene delivery exacerbates p53-dependent DDR burden as well as traps ITRs and compared it to IDLV-mediated gene delivery, which is associated with an improved p53 response and better engraftment capacity at the expense of lower editing efficiencies.

Here, we compared viral- and non-viral-based DNA delivery methods side-by-side in a clinically relevant setting. While both methods promoted high gene correction frequencies in vitro, non-viral mediated DNA delivery more efficiently targeted LT-HSCs than did its viral counterparts. Differences of editing processes, cells handling, and length of the DNA templates used in each process could be part of such discrepancy, but other factors are also likely to influence the overall functionalities of edited HSPCs. A comprehensive scRNAseq characterization of the gene edited products before in vivo injection allowed us to decipher some of the underlying reasons.

We observed a higher proportion of HSC-enriched subset in ssODN- compared to AAV6-edited HSPCs. This signature was correlated with a higher proportion of quiescent cells as well as a higher in vivo engraftment of ssODN-edited cells, indicating that the non-viral DNA delivery better preserved the pool of LT-HSCs. However, maintaining a higher proportion of primitive and quiescent HSCs is not the only parameter leading to efficient engraftment. Indeed, while displaying low proportion of primitive and quiescent HSC, the UT control unexpectedly showed high frequency of in vivo engraftment. While this may be explained by the high proliferation/differentiation rate of non-edited HSPCs (as suggested by the greater proportion of HSPCs in S phase in the UT compared to edited samples), this unexpected result indicates that other factors may play a key role in the repopulation capacity of edited-HSPCs.

The fitness of edited HSC appears to be an important factor for successful engraftment. Indeed, we observed more activation of the p53 pathway in AAV6-edited HSPCs than in ssODN-edited HSPCs. This phenomenon suggests that AAV6-edited HSPCs display lower fitness as reported earlier[25,26]. It also suggests that non-viral DNA delivery better preserved the fitness of HSCs by mitigating the toxicity associated with the gene editing process. Consistent with this, ssODN-edited HSPCs showed a higher expression of stemness markers as well as lower oxidative metabolism and protein synthesis, three transcriptomic signatures usually associated with primitive HSCs[72,73].

The observed high IFN response in edited HSPCs may also decrease their engraftment capacity, as was recently reported in patients with chronic granulomatous disease[74]. However, we did not observe this effect in our dataset, especially with the ssODN-edited HSPCs, which displayed the highest level of IFN response activation among all experimental groups. Because HSCs can activate the IFN response and reacquire their quiescent state later on[75], we hypothesize that our editing process triggers a transient and reversible IFN response that does not greatly affect the engraftment capacity of edited HSCs.

Elevated IFN responses are correlated with increases in the expression of inflammation genes. The inflammation pathway is usually activated upon a DNA sensing response triggered by the direct delivery of mRNA and DNA payloads to the cytoplasm[76]. Interestingly, the extent of this response was lower in AAV6- than in ssODN-edited HSPCs. This difference is probably due to the well-established ability of AAV6 to evade DNA sensing factors by delivering its DNA payload directly into the nucleus[77]. Nevertheless, DNA sensing and the resulting

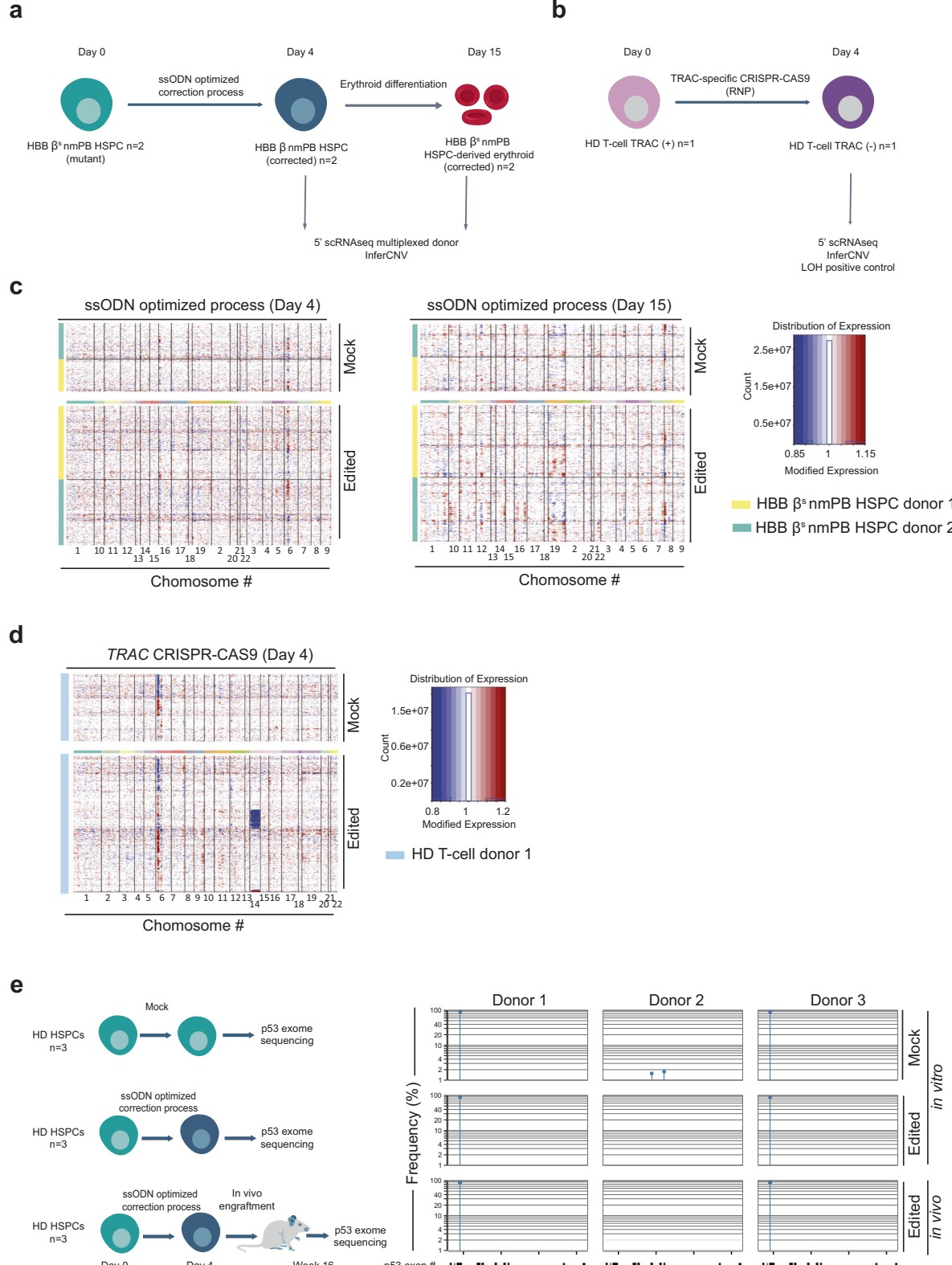

inflammation response may still account for the loss of HDR-edited cells observed in vivo, although additional work is needed to address this point.

Noteworthy, the different conclusions drawn out of our scRNAseq package are based on the transcriptomic analysis of 2 independent HD HSPCs. Additional work using multiple other HD and HbSS patient HSPCs, is now needed to confirm our conclusions on the mode of action of both DNA delivery strategies and on their impact on HSPCs fitness. That being said, although AAV6 appears to be the most efficient delivery method for integrating a large DNA template, this study and earlier elegant works[22,78] suggest that ssODN should be the preferred choice for short DNA template delivery. Indeed, ssODN provides a safer avenue for editing HSPCs because it mitigates p53 activation, limits DDR burden and increases HDR efficiency in LT-HSCs.

**Fig. 8 | Characterization of TALEN-mediated genomic adverse events after optimized non-viral editing at *HBB* locus in HSPCs. a** Representative schema of scRNAseq InferCNV analysis performed to identify the potential large genomic rearrangement including gain and loss of heterozygosity in HSPC, *n* = 2 HbSS donors transfected with mRNA encoding TALEN-HBB$_{ss}$, Via-Enh01, HDR-Enh01 and with ssODN. **b** Representative schema of experimental design to verify the ability of scRNAseq InferCNV to detect loss of heterozygosity in T-cells. **c** scRNAseq InferCNV results obtained from samples gathered 4 days post HSPC thawing (Day 4) and after erythroid differentiation (Day 15). Each individual row corresponds to a single cell and each column corresponds to a specific gene and its genomic position, grouped by chromosome. Red color represents increase in gene expression, while blue color represents decrease in gene expression. **d** scRNAseq InferCNV results obtained from T-cell sample edited by TRAC-specific CRISPR-CAS9 and gathered 4 days post editing (Day 4). The color code is similar as in (**c**). Typical examples of LOH are illustrated by the blue lines seen across chromosome 14, where the TRAC CRISPR-CAS9 is designed to cut. **e** *P53* exome sequencing obtained from the genomic DNA of untouched HD HSPCs (mock in vitro) or of HD HSPCs edited by the optimized non-viral gene editing process and gathered 4 days post editing (edited in vitro) or 16 weeks after NCG mice engraftment onset (edited in vivo). The frequency and exonic position of *p53* variants obtained in each donor (*n* = 3 independent biological experiments and donors) and in each experimental group are indicated. The detection threshold was set to 1% according to the *p53* exon sequencing kit's manufacturer guidelines. The *p53* variant identified in the mock and edited experimental groups of donors 1 and 3 corresponds to a natural single nucleotide variant frequently identified in healthy donors (rs1042522, worldwide median prevalence of 66.4%, Supplementary Fig. 7b).

Our preferred optimized protocol, based on TALEN and ssODN, achieved high gene correction efficiencies in vitro in HSPC population (Fig. 1b, 41.5% ± 5.3% HDR and 19.1% ± 2.1% indels, average frequency ± SD) and in vivo (Fig. 2b, d, 29.2% ± 8.7% HDR, 20.6% ± 8% indels, average frequency ± SD and 58.2% median engraftment) using PLX-mobilized HSPCs from HD. These results were confirmed in vitro in HSPC population (Figs. 4b and 6d, 44.3% ± 5% and 49.3% ± 4% HDR, 21.4% ± 2.3% and 23.7% ± 1% indels, average frequency ± SD), in BFU-E colonies (Fig. 4g, 58% of clones harboring mono and biallelic HDR and 9% of clones harboring biallelic indels) and in vivo (Fig. 6d, 21.0% ± 18.6% HDR, 13.5% ± 13.1% indels, average frequency ± SD and 23.2% median frequency of engraftment) using non-mobilized HSPCs from HbSS patients. Genetic correction of β$^s$ in HSPCs form HbSS patient translated into an efficient rescue of HbA (Supplementary Fig. 4c, 55.4% ± 5.5% of HbA rescued in RBCs, relative concentration ± SD) and a significant decrease of HbS (Supplementary Fig. 4c, from 93.6% ± 2.9% to 24.5% ± 14.4%, relative concentration of HbS ± SD, in RBCs), and of sickle RBCs (Fig. 5c, from 96.2% ± 1.2% to 40.5% ± 14.1%, average frequency of sickle RBCs ± SD) confirming that substituting HbS with HbA expression was sufficient to inhibit the Hb polymerization and the sickling process. Because one corrected *HBB* allele is sufficient to generate functional RBCs and because a threshold of 20% donor chimerism in the bone marrow is necessary for clinical benefit[4,79,80], this work provides evidence that coupling non-viral DNA delivery to TALEN editing has the potential to elicit a positive therapeutic outcome for SCD patients.

For this preclinical proof of concept, we chose the TALEN gene editing technology because of its high specificity and editing efficiency. However, other gene editing tools could theoretically be incorporated in a similar protocol[81–84]. Indeed, ex vivo gene therapy strategies based on DSB-free gene editing tools including base and prime editors, also displayed a potential clinical benefit for the treatment of SCD. Base editing at *HBB* locus was reported to elicit an average of 44 to 80% conversion of β$^S$ to β$^G$ allele (Makassar β-globin allele), with indels ranging from 1.2 to 2.8% and bystander editing events <2%[81], while prime editing elicited an average of 27% conversion of β$^S$ to β allele with an average of indels of 4.6% in plerixafor-mobilized HbSS patient HSPCs in vitro[82]. Conversion events were found to be maintained after engraftment in immunodeficient mouse models indicated that both editing strategies hold therapeutic potential for SCD treatment[81,82]. However, while these strategies offer a better edit-to-indel ratio, the genotoxic consequences of their DNA and RNA off-target activity has only recently been started to be characterized[81,82,85–87]. Given the high level of LT-HSC targeting obtained by gene correction or base/prime editor strategies for SCD, the strategy conferring the safest and most therapeutic benefit for the treatment of SCD will have to be determined in clinical trials.

To conclude, we posit that the TALEN gene editing protocol coupled with non-viral DNA delivery presented in this comprehensive preclinical dataset has the potential to be further developed for a therapeutic application targeting SCD.

## Methods
The research described in this manuscript complies with the regional investigational review board (reference: DC 2022-5364, CPP Ile-de-France II "Hôpital Necker-Enfants malades").

### CD34 + HSPCs sourcing
Frozen CD34+ HSPCs purified from healthy donor G-CSF-mobilized and Plerixafor-mobilized peripheral blood were purchased from All-Cells (Almeda), Hemacare (Los Angeles) or New York Blood Center (New York). CD34+ HSPCs derived from HbSS patients were recovered from erythrocytapheresis bags provided by Hôpital Necker-Enfants malades (Paris, France). Written informed consent was obtained from all adult patients. The study was approved by the regional investigational review board (reference: DC 2022-5364, CPP Ile-de-France II "Hôpital Necker-Enfants malades").

### CD34+ cell isolation
Mononuclear cell isolation was performed on erythrocytapheresis bags by diluting 2x red blood cell suspension with a medium containing PBS (Gibco, #70011044), 2% heat inactivated FBS (Gibco, #10082147), and 1 mM EDTA (Invitrogen, #15575020). Diluted red blood cell suspension was then distributed into Sepmate tubes (StemCell #85450) containing 15 mL of density gradient medium (StemCell, #07861). Tubes were centrifuged 10 min at 12,000 × *g*, the supernatant was poured in a new 50 mL tube, brought to a volume of 45 mL with medium containing PBS, FBS, and EDTA, and centrifuged 7 min at 300 × *g*. All pellets were pooled together to perform CD34+ cell isolation using CD34 progenitor kit according to the manufacturer's recommendations (Miltenyi #130-046-703). HbSS CD34+ cells were cryopreserved in medium containing FBS and 10% DMSO (Sigma, #D2438) in liquid nitrogen.

### CD34 + HSPC culture
After thawing, CD34+ HSPCs from healthy donors and HbSS patients were cultured at a concentration of 0.3 × 10$^6$ cells/mL in complete medium: StemSpan II (Stemcell, #09655), 1X CD34 expansion supplement (Stemcell, #02691) and 1X penicillin-streptomycin (Gibco, #15140-122) at 37 °C, 5% CO$_2$.

For GMP-compatible conditions: cells were cultured in GMP Stem Cell Growth Medium (SCGM, CellGenix, Freiburg, Germany) supplemented with human cytokines (Cellgenix GMP-grade), TPO (100 ng/mL), Flt3 (300 ng/mL), SCF (300 ng/mL), and IL-3 (60 ng/mL).

HSPCs were assessed for viability by Nucleocounter or by the expression of CD34 and viability marker by flow cytometry 2 days after gene editing. Flow cytometry staining was performed in PBS, 0.5% BSA, 2 mM EDTA with a CD34 VioBlue, Clone#REA1164 diluted 1/50 (Miltenyi, #130-124-459) antibody and a fixable viability marker e780 diluted 1/1000 (eBioscience, #65-0865-18). The gating strategy used to analyze cellular suspension is documented in supplementary Fig. 8.

HSPC pellets were recovered from culture 2 days or 5 days after TALEN electroporation for genomic DNA extraction and evaluation of editing efficiency.

Aliquots of HSPCs from healthy donors were frozen 2 days after gene editing for xenotransplantation experiments and for scRNAseq analysis.

## TALENs and DNA donor templates

Plasmids of the 4 TALEN arms, containing a T7 promoter and a polyA sequence (120 residues), were produced after assembly and linearized for mRNA in vitro transcription. TALEN mRNAs were produced by TriLink.

For viral-mediated HDR repair, a DNA template containing the corrected nucleotide for *HBB* repair and 5 silent mutations flanked by 300-nt left and right homology arm sequences was designed and carried by an AAV6 vector.

For non-viral mediated repair, a DNA donor template sequence containing the corrected nucleotide for *HBB* repair and 4 silent mutations flanked by 80-nt left and right homology arm sequences was designed. A 197-nt long single-stranded oligonucleotide was sourced from Integrated DNA Technology (IDT) and contained phosphorothioate modifications on the two first bases at both the 3' and 5' edge of the sequence. All sequences are available in Supplementary Table 1.

## Transfection and transduction of HSPCs

Two days after thawing, the cells were washed twice in BTXpress buffer and resuspended at a final concentration of $10 \times 10^6$ cells/mL in the same solution. The cellular suspension ($1 \times 10^6$ cells) was mixed with 15 μg mRNA encoding each TALEN arm in the presence or absence of 4 μg mRNA and 1 μg mRNA encoding for HDR-En01 and Via-Enh01, respectively, in a final volume of 100 μl. The cellular suspension was transfected in 4-mm cuvette gap size using PulseAgile™ technology. The electroporation program consisted of two 0.1 ms pulses at 1000 V/cm followed by four 0.2 ms pulses at 130 V/cm. Immediately after electroporation, the HSPCs were transferred to a new plate containing prewarmed medium at a concentration of $2 \times 10^6$ cells/mL and incubated for 15 min at 37 °C.

For transduction experiments, 15 min after TALEN electroporation, HSPCs were cultured in the presence or absence of AAV6 particles (MOI = $7.5 \times 10^5$ viral genome/cell) containing the *HBB* gene repair template at a concentration of $2 \times 10^6$ cells/mL and incubated for 15 min at 37 °C. HSPCs were then incubated at 30 °C overnight.

For ssODN transfection experiments, HSPCs were kept in culture at 30 °C for 16 h after TALEN electroporation, and a second electroporation was performed in the same conditions described above, using $1 \times 10^6$ cells in presence or absence of 1000 pmol of ssODN in a final volume of 100 μl. HSPCs were electroporated following the same conditions and program described above. After ssODN transfection, HSPCs were seeded at a concentration of $2 \times 10^6$ cells/mL and incubated at 30 °C overnight. The following day, cells were seeded at a density of $0.3 \times 10^6$ cells/mL in complete medium and cultured at 37 °C in the presence of 5% $CO_2$.

mRNA, ssODN electroporation and AAV6 transduction in GMP-compatible conditions were performed with the following modifications. Cells were washed and resuspended in an in-house made GMP-compliant electroporation buffer for both mRNA and DNA electroporation. 500 pmol ssODN were used for electroporation in 100 μl volume.

## Evaluation of editing efficiency and adverse events by ddPCR of edited HSPCs in cell culture, HSPCs-derived erythrocytes and bone marrow samples

Genomic DNA (gDNA) was extracted using the Qiagen kit DNeasy Blood and Tissue Kit (Qiagen, #69506) or QIAamp DNA Micro kit (Qiagen, #56304) according to the manufacturer's instructions. BFU-E clones were lysed using DNA Extract All Reagents kit (Thermo Scientific, #4403319) according to the manufacturer's instructions. gDNA quantifications are performed using NanodropOne device (ThermoFisher).

Assessment of editing rates was performed by digital droplet PCR (ddPCR) using QX200 Droplet Generator and analyzed using QX200 Droplet Reader (Bio-Rad), on 50 ng of purified gDNA or lysate (for CFU) using a triplex assay allowing the quantification of edited, non-edited, and NHEJ-mediated indel events. This triplex assay was designed with a PCR amplicon encompassing the TALEN cutting site and *HBB* correction in which one of the primers could not bind to any homology donor template and three probes: one probe detecting edited sequences, one drop-off probe situated at the TALEN cutting site identifying indel events, and one probe for reference[88]. Assessment of adverse events was performed based on protocol described by Long et al.[50] using 100 ng of purified gDNA. All primers and probes sequences were reported in Supplementary Table 2. ddPCR was performed following the BioRad protocol. Genomic DNA was combined with 1 μl of EcoRI restriction enzyme (ThermoScientific, #FD0274), 10 μM each of target primer and FAM probe mix, 10 μM each of reference primer and HEX probe mix, 1xddPCR Supermix probe without dUTP (Bio-Rad, UK), and nuclease-free water in a final volume of 20 μl. Detailed sequences for primers and probes are available in Supplementary Table 2.

## Identification and detection of candidate off-site targeting by oligo capture assay and high-throughput DNA sequencing

Oligo capture assay (OCA) was used as previously described[47] to assess the specificity of TALEN-HBB$_{ss}$ activity. Briefly, primary T cells derived from SCD-patients were co-transfected with mRNA encoding TALEN-HBB$_{ss}$ and double-strand oligodeoxynucleotide (dsODN) and expanded for 6 days. Genomic DNA was recovered, sheared, end-repaired/A-tailed, processed, and analyzed by high-throughput DNA sequencing as described previously[47]. Briefly, next-generation sequencing Y-adapters (TruSeq Annealed Adapter, Illumina) containing the P5 sequence were ligated to the ends of sheared DNA. Nested, anchored PCR using dsODN-specific and adapter-specific primers were used to specifically amplify the plus and minus strand of DNA fragments that incorporated the dsODN (Supplementary Table 2). The first round of PCR used the adapter-specific (P5_1) and dsODN-specific primers (Supplementary Table 2). The second round of PCR used the adapter-specific (P5_2) and dsODN-specific primers along with a third oligo named P7 that added the barcode and P7 sequence to the ends of the DNA (Supplementary Table 2). Equal amounts of these separate PCRs were pooled together and sequenced using Illumina MiSeq ($2 \times 150$ bp). The resulting sequences were then mapped onto the human genome (GRCh38) to identify potential off-target sites. The frequency of insertion and deletion events (indels) generated at potential off-target sites were then quantitatively assessed using high-throughput DNA sequencing of candidate off-site-specific PCR amplicons obtained from HSPCs edited with TALEN-HBB$_{ss}$ (without dsODN).

## Deep sequencing/Amplicon sequencing for on- and off-target detection/*p53* exome sequencing

For deep sequencing, 100 ng of genomic DNA was used per 50 μL reaction with Phusion High-Fidelity PCR Master Mix (NEB). The PCR conditions were set to 1 cycle of 30 s at 98 °C; 30 cycles of 10 s at 98 °C, 30 s at 60 °C, 30 s at 72 °C; 1 cycle of 5 min at 72 °C; hold at 4 °C. The PCR product was then purified with Omega NGS beads (1:1.2 ratio) and eluted into 30 μL of 10 mM Tris buffer pH 7.4. The second PCR, which incorporates NGS indices, was then performed on the purified product from the first PCR. Fifteen microliters of the first PCR product were used in a 50 μL reaction with Phusion High-Fidelity

PCR Master Mix (NEB). The PCR conditions were set to 1 cycle of 30 s at 98 °C; 8 cycles of 10 s at 98 °C, 30 s at 62 °C, 30 s at 72 °C; 1 cycle of 5 min at 72 °C; hold at 4 °C. Purified PCR products were sequenced on MiSeq (Illumina) on a 2 × 250 nano V2 cartridge. Detailed sequences for primers are available in Supplementary Table 2. *P53* genomic integrity was assessed using xGen™ TP53 Amplicon Panel kit (IDT) and sequenced using NextSeq (Illumina). Genomic DNA samples (20 ng/sample) were PCR-amplified in technical duplicates (per donor and per condition) using the manufacturer protocol. The sequence reads obtained from high throughput DNA sequencing were trimmed for quality and sequencing adapters using Trim Galore and aligned on the human genome (assembly GRCh38) using Bowtie2[89] in very sensitive local mode. Regarding the dataset obtained from cell gathered after in vivo mouse engraftment, sequence reads were also aligned on the mouse genome (assembly mm10) to detect potential contamination by mouse genomic sequences. The reads were discarded when the mapping quality was higher for the mouse alignment than for the human alignment. Doing so, 1.30% ± 0.93 of the mouse sequence reads (average frequency ± SD) were filtered out from our dataset. The human sequence reads were then analyzed for potential *p53* variants. The *p53* variants were called using Lofreq[90] with a minimum baseQ of 20 for all bases, and a minimum mapping quality of 20. Finally, we filtered out the variants that were not called in both technical replicates and supported by less than 2000 sequence reads.

### CAST-SEQ analysis
The protocol use to analyze the genomic DNA of HSPCs edited by TALEN-HBB$_{ss}$ was adapted from ref. 49 using the CAST-SEQ oligonucleotides documented in Supplementary Table 2. Regarding dataset bioinformatic post-processing, the sequenced reads were trimmed for quality with the trim_galore tool, and the mates were assembled using the FLASh software. The resulting sequences were mapped on the Human genome (release GRCh38) using the Bowtie2[89] software with the -k2 option (allowing for up to two alignments per sequence). We filtered the sequences, keeping those with one alignment at the expected location in the onsite and the other at least 300 bp away. The start position of these second alignments were used as the structural event location. After checking the reproducibility of the events, we binned their positions (bin size = 50 bp) and used the orientation to determine the probable adverse events (deletion, inversion, or trans-location). Events were plotted using the CIRCOS tool.

### Colony forming unit (CFU) assays
CD34+ HSPCs recovered two days after electroporation or human CD45+ recovered from mouse bone marrow 16 weeks after in vivo injection, were plated in methylcellulose (Stemcell, #04435) for Colony Forming Unit (CFU) assays. A total of 200–500 cells (CD34+) or 200,000–300,000 (hCD45+) were resuspended in 100 μL of Stemspan II and transferred to an aliquot of 1 mL of methylcellulose, mixed, and plated in a Smartdish well (Stemcell, #27371). Cells were cultured for 12–14 days in methylcellulose according to the manufacturer's instructions. At the end of the culture, colonies were automatically counted using a Stemvision (StemCell) automated colony counter to assess plating efficiency (number of colonies counted at day 14/number of cells plated at day 0). Bulk and single BFU-E colonies were picked at day 14 and lysed to obtain gDNA to evaluate editing efficiency.

### HPLC
HPLC experiments were performed on CFU-derived BFU-E bulk (~25 colonies) and HSPC-derived erythroid cells (400,000 cells). Hemoglobin tetramers and monomers were quantified using cation-exchange (CE) HPLC, and globin chains were assayed using reverse-phase (RP) HPLC as previously described[14].

### Single-cell RNAseq
Single-cell mRNA barcoding and library generation were performed following the 10X Genomics protocol from the Chromium Next GEM Single Cell 5′ Kit v2 (#1000263). In order to distinguish donor-by-donor variability, samples from different donors were multiplexed following a previously described cell-hashing protocol[35]. Briefly, 300,000 CD34+ cells or erythroid differentiated cells were thawed and incubated with Human TruStain FcX™ Fc Blocking reagent (Biolegend, #422301) for 10 min at 4 °C. Afterwards, the cells were stained with TotalSeq-C Hashtag antibodies diluted at 1/50 in Cell Staining buffer (Biolegend, #420201) for 30 min at 4 °C. After washing, different conditions bearing different hashtags were pooled and droplets were generated. Cells were loaded into Chromium Single-Cell Chip (10X Genomics) at a target capture rate of ~10,000 individual cells per sample.

Gene Expression and Cell Surface libraries were checked using a Bioanalyzer High Sensitivity DNA kit (Agilent, #5067-4626) according to the manufacturer's recommendations. Libraries generated were sequenced by Institut du Cerveau (ICM) sequencing platform using a NovaSeq 600 sequencer from Illumina following 10X recommendations.

Sequencing reads were demultiplexed and aligned to the human reference genome (GRCh38), using the CellRanger pipeline v6.1.2.

### Single-cell RNAseq bioinformatic analysis
Raw sequences were processed by Cell Ranger v6.1.2 to generate the count matrix, using either count or multi-option depending on the cases. Cell with less than 200 unique molecular identifiers (UMI) or more than 8000 UMI were filtered out, as well as cells with more than 10% mitochondrial genes. Data were normalized, and the 2000 most variable features were retained using Seurat v4.3.0, before doing a PCA. When required, data were batch-corrected using Seurat CCA approach. Cell type determination was done with Azimuth in R using the "bonemarrowref" reference. For the sake of simplicity, cell type acronyms were used in Fig. 3 to name the different HSPCs sub-populations identified. We used HSC-enriched for Hematopoietic Stem Cells; LMPP for Lymphoid Primed Multipotent Progenitors; CLP for Common Lymphoid Progenitors; MEP for Megakaryocytic Erythroid Progenitors; GMP for Granulocyte Monocyte Progenitor; Early Eryth for Early Erythroid Progenitors; Late Eryth for Late Erythroid Progenitors; Prog MK for Progenitor Megakaryocyte; Prog DC for Progenitor Dendritic Cells; Pre-proB for Progenitor and Precursor B cells and BaEoMa for Basophil Eosinophil Mast Progenitors. Clustering was done using FindNeighbors and FindClusters functions from Seurat followed by UMAP projection, and cell cycle state was assessed with CellCycleScoring function. Differential gene expression was assessed using Seurat FindMarkers with default parameters but setting no minimum for the log fold change and min.pct. Gene Set Enrichment Analysis (GSEA) was carried out using Reactome and KEGG pathways retrieved from MSigDB with the R package msigdbr. The editing status of the *HBB* gene in cells obtained 11- and 13-days post differentiation onset, was studied using an ad hoc python script after realignment of the reads using a dynamic programming algorithm with parameters adapted to detect big deletions. Editing values obtained at both time points were then aggregated and plotted in Fig. 5. The same sample and data management approach was adopted to plot the level of *HBG* and *HSPA1A* expression.

InferCNV was performed using the protocol and the CRISPR-CAS9 LOH positive control described in ref. 53. Briefly, PBMC were thawed at day 0 and activated at day 1 by dynabeads (25 μL/1 × 10⁶ CD3 + T-cells) and allowed to grow for 3 days. At day 4 cells were passaged with new media and were used at day 5 to perform the electroporation of the Ribonucleotide protein (RNP) complex ito activated T-cells. The RNP complex was generated by mixing CAS9 with CRISPR targeting TRAC locus (61 μM and 50 μM final concentration, respectively, Supplementary Table 2). 25 μL of the RNP solution was mixed with 100 μL of activated

T-cell (25 × 10⁶ cell/mL, final concentration) in BTXpress buffer and electroporated right away using the PulseAgile™ apparatus. Transfected T-cells were allowed to grow for 4 days before being recovered and analyzed by scRNAseq using the protocol described earlier.

### In vitro differentiation of CD34 + HSPCs into erythrocytes

Two days after TALEN transfection, HSPCs were plated at a concentration of 0.5 × 10⁵ cells/mL to start the erythroid lineage differentiation as described by Giarratana et al.[91]. First, cells were plated for 6 days in a serum-free medium containing SCF at 100 ng/mL (CellGenix, #001418-050), IL3 at 5 ng/mL (CellGenix, #001402-050), EPO at 3 UI/mL (Stemcell, #78007) and Hydrocortisone at 10 μM (Sigma, #H0888-1G). For the second step of differentiation, HSPCs were co-cultured on an MS5 cell layer in a medium containing EPO at 3 UI/mL for 3 days. Cells were kept in MS5 co-culture with medium containing 10% of human AB serum (Biowest, #S4190-100) for a total of 21 days differentiation.

To assess the quality of erythroid differentiation, HSPC-derived erythrocytes were stained for flow cytometry analysis at different time points during the differentiation protocol, using the following antibodies: CD36 V450, Clone#CB38, diluted 1/20 (BD, #561535), CD71 FITC, Clone#M-A712, diluted 1/50 (BD, #555536), CD233 PE, Clone#BRIC 6, diluted 1/50 (IBGRL, #9439), CD235a PECy7, Clone#GA-R2, diluted 1/100 (BD, #563666), CD49d APC, Clone#9F10, diluted 1/20 (BD, #559881), viability dye 7AAD, diluted 1/100 (BD, #559925), and Draq5 1/500 (eBioscience, #65-0880-96). The gating strategy used to analyze cellular suspension is documented in Supplementary Fig. 9.

Aliquots of erythroid differentiating cells were frozen at day-11 and day-13 for scRNAseq.

### Sickling assay

Twenty-one days after the beginning of erythroid differentiation, HSPC-derived erythroid cells were exposed to an oxygen-deprived atmosphere (0% O₂), and the time course of sickling was monitored in real time by video microscopy. Images were captured every 20 min for at least 80 min using an AxioObserver Z1 microscope (Zeiss) and a 40× objective. Images of individual fields were taken throughout all stages and processed with ImageJ to determine the percentage of non-sickled Red Blood Cells (RBC) among the total RBC population. A total of 500 to 1000 cells were counted per condition.

### Transplantation of CD34 + HSPCs into NCG or NBSGW mice

Frozen aliquots of mobilized peripheral blood CD34+ cells from healthy donors obtained 2 days after editing were sent to TRANS-CURE bioServices (Archamps, France) for xenotransplantation into 5-weeks-old female NOD*Prkdc*ᵉᵐ²⁶ᶜᵈˢ² *Il2rg*ᵉᵐ²⁶ᶜᵈ²²/NjuCrl (NCG) mice. Pre-transplant conditioning was based on Busulfan (Sigma, #B1170000) according to the TRANSCURE bioServices protocol. Non-mobilized CD34+ cells derived from HbSS patients were transplanted 2 days after editing in 8-week-old female NOD.Cg-Kitᵂ⁻⁴ᴶTyr+Prkdcˢᶜⁱᵈ Il2rgᵗᵐ¹ᵂʲˡ/ThomJ (NBSGW) mice. Busulfan (Sigma, #B1170000) conditioning was performed using a dose of 15 mg/kg body weight through an intraperitoneal injection one day before HSPC transplantation. NCG and NBSGW mice followed an acclimatation period of 7 days prior being used in experiments. Mice were housed by groups of 2-to-5 in ventilated cages (type II, 16 × 19 × 35 cm, floor area = 500 cm²) at a constant temperature $T$ = 22 ± 2 °C, at a constant relative humidity $RH$ = 55 ± 10%, a photoperiod of 12:12-h light-dark cycle 7 am:7 pm, with water and food available ad libitum. Mice were monitored daily for unexpected signs of distress. Body weight was measured once a week. Mice with a cumulative clinical score ≥7 were euthanized. For a mouse with a body weight loss >20% associated to a clinical score <6, the veterinarian was consulted. The decision to euthanize an animal in pain or having reached the ethical limit was at the sole decision of the veterinarian. Mice showing a body weight loss >10 and <20%, were fed with nutritionally fortified food until recovery to 10% body weight loss.

Edited and non-edited control HSPCs were transplanted via tail vein (NCG) or retro-orbital (NBSGW) injection at 0.25 or 0.7 × 10⁶ cells for NCG mice and 5 × 10⁵ cells for NBSGW mice. For all xeno-transplantation studies, 16–18 weeks after transplantation, mice were sacrificed, and peripheral blood, bone marrow, spleen, and thymus were recovered. One hundred thousand cells from each organ were harvested, and chimerism was assessed by flow cytometry using the following antibodies: mouse CD45 Vioblue, Clone#REA737, diluted 1/50 (Miltenyi, #130-110-664), human CD45 APCVio770, Clone# REA747, diluted 1/50 (Miltenyi, #130-110-635) and viability dye 7AAD, diluted 1/200 (BD, #559925). The gating strategy used to analyze cellular suspension is documented in supplementary Fig. 10.

Human CD45+ cells from bone marrow were sorted using human CD45 Microbeads (Miltenyi, #130-045-801) following the manufacturer's recommendations, and gDNA was extracted to assess gene editing stability after engraftment. For NCG xenotransplantation, an aliquot of sorted human CD45+ cells was plated for CFU assays.

All procedures and animal housing for NCG xenotransplantations were performed at TransCure bioServices (Archamps, France) and were reviewed and approved by the local ethics committee (CELEAG). All procedures for NBSGW xenotransplantations were performed at the Hospital Necker animal facility in compliance with the French Ministry of Agriculture's regulations on animal experiments and were approved by the regional Animal Care and Use Committee APA-FIS#2019061312202425_v4. Animal experiments abode by the ARRIVE guidelines.

### Multilineage engraftment

Cells recovered from mouse bone marrow 16–18 weeks post injection were assessed for multilineage engraftment in the lymphoid compartment using CD19 FITC, Clone#J3-119, diluted 1/100 (Beckman Coulter, #A07768) and CD3 APC, Clone#REA613, diluted 1/50 (Miltenyi, #130-113-135) antibodies, the granulocyte compartment using CD15 PE, Clone#80H5, diluted 1/50 (Beckman Coulter, #IM1954U), CD14 PECy7, Clone#MφP9, diluted 1/50 (BD, #562698) and CD11b APC, Clone#M1, diluted 1/100 (BD, #553312) antibodies, and the erythroid compartment using CD36 FITC, Clone#CB38, diluted 1/50 (BD, #555454), CD235a PE, Clone#GA-R2, diluted 1/50 (Invitrogen, #12-9987-82) and CD71 APC, Clone#M-A712, diluted 1/50 (BD, #551374) antibodies.

Cells were incubated in a buffer containing PBS, 0.5% BSA, 2 mM EDTA and antibodies for 15 min at room temperature and then washed two times in 1X PBS. Cells were resuspended in PBS containing 2% FBS and analyzed by flow cytometry. The gating strategy used to analyze cellular suspension is documented in supplementary Fig. 11.

### Instrument and software used for data acquisition and analysis

Cell viability was acquired using NucleoCounter® NC-250 (ChemoMetec) and analyzed using NucleoView™ software v4.3. CFU colonies were detected using a STEMvision apparatus (STEMCELL Technologies) and analyzed using STEMvision Colony marker (STEMCELL Technologies) v2.0.3.0. DNA quantification was performed using a NanodropOne device (ThermoScientific) and analyzed using NanoDrop QC software v1.6.198. Flow cytometry analysis were aquired using MACSQuant 10 (Milteniy Biotech), BD CANTO (Becton Dickinson Bioscience), Fortessa X20 (Becton Dickinson Bioscience) or Gallios (Beckman coulter) flow cytometers. Macsquantify software v2.11 (Milteniy Biotech), BD FACS DIVA software v9.0 (BD), FlowJo software, or FlowJo v10.8.1 (Treestar), were used to analyze flow cytometry dataset. Digital droplet PCR were prepared using QX200 Droplet Generator (Bio-Rad) and analyzed using QX200 Droplet Reader (Bio-Rad). ddPCR results were analyzed by QuantaSoft™ Software v1.7 (Bio-Rad). Single-cell formulations were obtained using Chromium single cell system (10X Genomics), RNA Seq high throughput DNA sequencing was performed using MiSeq,

NextSeq or NovaSeq systems (Illumina). Sequence read were demultiplexed and aligned to the human reference genome (GRCh38), using the CellRanger pipeline v6.1.2 (10X Genomics) and Azimuth was used to identify cell subpopulations. HPLC experiments were performed using Nexera X2 HPLC system (Shimadzu) equiped with SIL-30AC autosampler. HPLC chromatograms were analyzed using LC solution software v5.51 (Shimadzu). GraphPad Prism software v9.2.0. was used to plot most of the dataset illustrated in this study.

### Statistical analysis
Comparisons of numerical variables between two groups were carried out using Mann–Whitney U tests as specified in the figure legends. One-way ANOVA with Tukey's multiple comparisons test or Kruskal–Wallis followed by Dunn's multiple comparisons test were used to analyze three or more groups with one variable and two-way ANOVA with Bonferroni post-test in case of two variables. P-values are indicated in the plots. All statistical analyses were performed using GraphPad Prism v.9.4 (GraphPad).

### Reporting summary
Further information on research design is available in the Nature Portfolio Reporting Summary linked to this article.

## Data availability
The authors declare that the data supporting the findings of this study are available in the article and in the supplementary information files. The raw single cell transcriptomic dataset, p53 exome sequencing, CAST-SEQ and SureSelect dataset are available on NCBI SRA under the BioProject PRJNA1117889. Source data are provided with this paper.

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

## Acknowledgements

We would like to thank Cellectis publications team for reviewing the manuscript and their valuable advice. We would like to also thank Sandra Rozlan and Emilie Dessez for helping to purify the HbSS HSPCs used in this study.

## Author contributions

A.Mo., P.D., A.J., and J.V. conceived the study. AMo designed experiments, analyzed and interpreted data. G.L. and S.Li. performed and analyzed most of the experiments included in the study. A.C. and T.F. performed and analyzed HPLC and in vivo NBSGW experiments. C.F., N.P., and S.Le. setup GMP-compatible protocol and performed experiments. A.Ma. supervised C.F., N.P., and S.Le. and designed experiments. D.L.C. performed and analyzed ddPCR for genomic rearrangements. S.T.B., P.H., L.M., and E.S. designed and performed experiments. A.B. designed the TALENs. R.G. and A.G. supervised D.L.C. and contributed to the design of the in vivo NCG study. M.C. and A.Mi. contributed to the design of the experimental strategy, data interpretation, and provided patient samples. D.L.C., L.M., and R.H. performed the p53 exome sequencing. S.P. analyzed the p53 exome sequencing and CAST-SEQ dataset. A.D. analyzed scRNAseq, Ampliconseq and off-target data. A.J., P.D., and J.V. contributed to the design of the experimental strategy and data interpretation. A.Mo. and J.V. supervised the study and wrote the manuscript with the help from all authors. J.V. supervised the collaboration between Cellectis and the Imagine Institute.

## Competing interests

A.Mo., G.L., C.F., DL.C., S.T.B., A.Ma., S.Le., N.P., R.G., L.M., R.H., S.P., A.B., E.S., A.J., A.D., P.D., and J.V. are current employees and equity holder at Cellectis; M.C. and A.Mi. received research funds from Cellectis. S.Li., P.H., and A.G. were employees and equity holders at Cellectis at the time the work was conducted. TALEN® is a Cellectis patented technology. The remaining authors declare no competing interests.
