## [Peer Review File · Nature Communications]

REVIEWER COMMENTS

Reviewer #1 (Remarks to the Author):

Moiani et al use the TALEN technology to develop a gene editing process for the treatment of sickle cells disease via homology-directed repair. In their work they compare viral vs non viral DNA template for DNA template delivery. Their results convincingly show that non-viral DNA delivery combined to TALEN gene editing reduces toxicity due to viral DNA delivery and results in superior HBB gene correction in long-term HSC. The paper is well written, contains very solid data thanks and experiments have been performed in a rigorous and complete way. Their work represent a significant advance in terms of translational development, considering also that they set up the procedure with GMP grade reagents and in a very thorough way. However, the comparison between viral and non-viral methods for sickle (Romero et al 2019, Patthabi et al, 2019), and the toxicity of AAV have been previously shown and limit somehow the novelty of this paper.

-

Major comments

- The authors showed that the optimized editing protocol can decrease indels in the cell product and in the xenograft and thus reduce the probability of generating a beta-thal phenotype in sickle patients. This point is absolutely well taken and their results outperform those reported previously in similar studies (Romero et al; Patthabi et al). Yet, some clarifications should be provided to strengthen it further, as it is one of the most relevant novelty aspects of the paper:

o Can the authors verify whether the optimized editing protocol is having any impact on the frequency of rearrangements, such as translocations and large deletions? It is possible that NHEJ inhibition favors other DNA repair pathways, such as MMEJ, and thus potentially lead to an increased frequency of genotoxic outcomes.

o In Fig. 6, there is high variability among mice in terms of HDR editing efficiency and indels, ranging from 0 to 60%. Are the mice with the lowest HDR editing efficiency those showing the highest indels frequency, and vice versa? If this is the case, it means that there is a 50% probability of acquiring a beta-thal like phenotype. Viceversa, are the mice with the highest HDR editing also those with the higher indel frequency? This could reflect high variability of the editing procedure and/or extremely low graft clonality, a potentially worrisome finding regarding the safety of this approach.

o In Fig. 5 and Suppl. Fig. 4, the authors generate and use beta-thal like HSPCs as control of a condition in which high level (>60%) of biallelic knock-out could lead to an undesired phenotype of the erythroid progeny, with pathogenic potential. Yet, this control is somehow an extreme scenario. Using HSPCs edited with the standard editing condition or with previously published editing protocols is the most appropriate control to claim “that our optimized process represents a substantial improvement on existing approaches”.

- In Fig. 2B, one would expect the engraftment of untreated and mock treated animals to be higher than ssODN and AAV ones, in line with the data on colonies in Fig. 1F and with previous publications. Similar observation applies also to Fig 6B. Can the authors add to the text a brief explanation/speculation on why this was not the case? Moreover, since the experiment was performed transplanting cells separately from three different donors, authors could use a color/symbol code to identify mice transplanted with HSPCs from the same donor in each experimental group. This could help interpreting the data.
- In the untreated control of the scRNA-seq analysis, the % of HSC-enriched cells is very low compared to the treated groups. Can the authors explain and comment on this? While the lower % of quiescent cells could be explained by the more proliferating status of unmanipulated cells (as mentioned in the paper), this observation is more difficult to interpret, despite being in line with the engraftment data shown in Fig. 2B.

Minor comments:

- The acronyms HDR-Enh01 and Via-Enh01 should be substituted with the name of the actual protein encoded by the electroporated mRNA.
- The letter "E" for the panel in Fig. 6 is missing.
- Numbers in the statement related to Figure 6D-E in the main text are wrong.
- When discussing base and prime editing strategies, this recent ref could be added:
<https://doi.org/10.1038/s41587-023-01915-4>

Reviewer #2 (Remarks to the Author):

Summary

The present manuscript improves therapeutic double-strand-break-based editing of HSPCs based on the TALEN platform, in the context of sickle-cell disease, using preclinical/clinically relevant materials and methods, and employing comprehensive and cutting-edge analysis technology.

On the plus side, the authors detail TALEN and viability- and HDR-enhancing mRNA sequences, compare viral and non-viral donors to allow reproduction. They explore transition from research- to GMP-suitable reagents and include in their analyses single-cell RNA-Seq and transplantation experiments in immunodeficient mice, besides a range of solid functional analyses and exemplary off-target analyses in therapeutically relevant cells. Efficiencies and tolerability reached and the reported off-target behavior compare very favorably with other recent work on DSB-based therapies.

On the minus side, single-cell RNA-Seq and its conclusions are based on multiplexed sequencing of two sickle-cell patient samples. Also, many of the developments on their own are incremental, and the Results text is at time too speculative, while the Discussion text omits any concrete comparison to efficacy and safety data for competing therapies, e.g. using same-target CRISPR/Cas and DSB-independent platforms.

Overall and despite minor shortcomings, the combination of analyses and improvements presented in the manuscript significantly contributes to our understanding of DSB-based therapies and their greater safety and efficacy. This is all the more important for allowing more generalised statements going forward, the present study being based on data for TALENs instead of the far more commonly used CRISPR/Cas platform.

Major Comments

(1)

The use of Via-Enh01 increases survival of cells post-HDR. However, it also increases the likelihood of enriching the edited cell population for p53-pathway-deficient cells. In line with original observations around p53-related changes during DSB-based editing (e.g. PMID 29892067) and with ambitions for clinical application based on GMP-compliant conditions, this should be monitored for the viable post-editing cell population, with and without Via-Enh01, for GMP-compliant culture conditions.

(2)

The authors should refrain from conjecture or excessive interpretation in the Results section, as I will illustrate here for three examples.

First, this applies to some statements in lines 251 to 307, where scRNAseq data for n=2 and duplexed sequencing are at times over-interpreted and are generalised without a disclaimer about the potential limitations. If such a disclaimer would be better at home in the Discussion, then so would be an interpretation (currently in Results) going from downregulated mitochondrial metabolism and protein synthesis, over preservation of more quiescent primitive cells, to the assertion that “Overall, these data show that the non-viral mediated DNA delivery strategy preserves more engraftment potential” (lines 303 to 304), ahead of any presentation of data for engraftment later in the manuscript. What is more, besides differences in buffer and cell handling, the template delivered by AAV is much longer than the ssODN, which might have toxicity effects on the treated cells that are unrelated to “viral” characteristics. Overall, current lines 497-534 are a better place for corresponding conclusions and discussion.

Second, the interpretation in the Results section that observing fewer sickling cells after correction confirms that HbA expression suffices to inhibit Hb polymerization (lines 362 to 363) would be more suited for the Discussion, besides ignoring the likely more fundamental effect of corrected cells producing less HbS.

Third, understandably the manuscript does not take a range of major existing protocols (e.g. PMID 37294917) to compare their impact on transcription profiles, but nevertheless claims in the Results that similarity of the profile of corrected cells to that of Mock EP cells (i.e. uncorrected sickling cells) and dissimilarity to that of created β -thal cells (and here, why one state would be preferable to the other is not clear), would be “confirming that [the authors’] optimized process represents a substantial improvement on existing approaches.”

Altogether, the authors should critically re-read their Results section and reduce or move speculative statements that will otherwise undermine the credibility of their work.

(3)

The Discussion should consider, at least briefly, general efforts of enhancing HDR-based repair (e.g. PMID 34713224) and consider the best as well as the worst indel frequencies that have been observed in (not necessarily HbS) HSPCs, and it should draw more specific comparisons to more recent DSB-based studies addressing recombination events (e.g. PMID 37070192) as well as to DSB-independent alternatives (as found in current references 64-66), beyond what is currently presented in lines 541-550 and before a combined summary (“In summary” line 535) and conclusion (“To conclude” line 551) of the findings. This could also consider the genotoxicity (e.g. PMID 37679541) of DSB-less systems. In the same vein and possibly juxtaposed to results achieved by other systems, the Discussion should reiterate key percentages achieved in terms of HDR-based correction, chimerism and their combination, to allow the conclusion in lines 538-540 that the current work supports therapeutic application of TALEN-based editing for SCD.

(4)

[Supplementary Figure 4]

A major point merely for its importance to be fixed, rather than for its impact on the data, make sure to consider what appear to be late rearrangements of the figure panels in the labels of the legend and in the main text. It appears that panel S4A is currently described as panel S4D, S4C as S4B and S4D as S4C.

Minor Comments

General remarks:

This is a single-pass correction, with apologies for errors inadvertently introduced in the process.

Throughout, use “FACS” only for cell sorting applications and instead apply “FCM” for mere analyses by flow cytometry, in text, figures and figure legends. This applies e.g. in the figure, panel S4B x3 and in the legends of Figures S5 – S8.

Throughout, it appears that “template” and “templates” would be the conventional terms to use in place of “matrix” and “matrices”, respectively, starting from line 161.

Throughout, use μg instead of ug and μL instead of uL.

Define the precise meaning of β_0 , which is currently defined as “indel allele” and the meaning of an “indel allele” throughout the manuscript. E.g. are these out-of-frame indels or 21+ nt indels (to go by the definition of Synthego’s KO Score) or any indels?

In the figure legends, separating “n=3” or the like off as separate sentences is not suitable; rephrase suitably throughout, e.g.:

- “as mean \pm SEM. n=3. Two-way ANOVA” -> “as mean \pm SEM, for n=3. Two-way ANOVA” or “as mean \pm SEM (n=3). Two-way ANOVA”
- “as mean \pm SEM. n=3 independent biological experiments” -> “as mean \pm SEM, for n=3 independent biological experiments”

Also, keep “n” in minor case (e.g. in the legend for S4E)

Throughout, use “Hb” in the main text instead of “hemoglobin” after its initial definition.

For the following, text in [square brackets] indicates line numbers (for the main article PDF) or section/figure labels (for supplementary materials), and {curved braces} indicate notes; other text indicates requested text changes (old -> new) or insertions.

[1]

correct -> corrects

[49]

proportion -> proportions

[50]

Therefore -> This suggests that {This is open to interpretation.}

[51]

due to -> linked to

[61]

βS -> βS (NM_000518.4 (HBB):c.20A>T)

[62]

{Define sickle hemoglobin here instead of line 79.}

[69][101]

ex-vivo -> ex vivo {Latin compound adjectives usually remain without hyphens.}

[79]

Adeno Associated Virus 6 -> adeno-associated virus 6 vectors {See also line 160.}

[80]

single stranded -> single-stranded

[89]

concerns of AAV6 -> concerns over AAV6

[96]

corrected HBB alleles -> corrected, HBB alleles

[112]

long-term repopulating stem cells (LT-HSCs) -> long-term hematopoietic stem cells {For consistency and in line with the abbreviation. While “long-term–repopulating hematopoietic stem cells” would be a more precise term, elsewhere in this article, including in the title, “long-term hematopoietic stem cell” is used, which is of equally common usage in the field.}

[126-148]

End sentences with a full stop.

[132][347]

Cation Exchange -> cation-exchange

Reverse Phase -> Reversed-phase

[141]

{Specify by rephrasing here and in the supplemental M&M section whether busulfan-based conditioning was applied only to NCG mice, or to both NCG and NBSGW mice.}

[148]

{Supply the GEO accession number.}

[160]

adeno-associated viral vector (AAV6) -> AAV6

[164]

HBB gene correction -> β S gene correction

[165]

digital droplet -> digital droplet PCR

[168]

mRNA-encoding TALEN -> TALEN-encoding mRNA

[172]

indels -> insertions and deletions (indels)

[173][196][Figure 1C in the figure]

HDR/Indel -> HDR/indel {Make sure to correct the plural version of "HDR/Indels" to singular throughout.}

[179]

HDR-Enh0129-31 which -> HDR-Enh0129-31, which

[204]

Also give the SD (or, if it is retained, the SEM) for the average percentage values (n=9), to give an idea of the sample variation in the main text.

[209]

(> 95% mean CD34+ cells assessed by flow cytometry) -> (> 95% mean CD34+ cells), as assessed by flow cytometry 4 days after editing {Referral to the time point is critical in the context of earlier measurements of viability.}

[215-218]

{This is confusing. Rephrase to clarify which number(s) refer to clonogenic potential and which to "mild reduction". Is this a reduction in the BFU-E percentage or a reduction in the overall yield of colonies in the assay, and in the latter case, what 100% control value does the percentage refer to?}

[221]

high level -> frequent

[223]

long-term engrafting HSCs -> LT-HSCs {See also line 112.}

[226][403]

long-term repopulating hematopoietic stem cells (LT-HSCs) -> LT-HSCs {See also line 112.}

[235]

consisted of -> generated/produced

[239]

respectively – lower than the relative input but less pronounced than AAV (45% and 50% median HDR frequency, respectively) – and no -> respectively, and a smaller drop in HDR frequency compared to the in vitro input frequency of 45% for dsODN vs. 50% for AAV. No

[244]

13.1 -> 13.1%#

[245][246]

respectively -> respectively,

[248]

in term of -> in terms of

[260]

Azymuth -> Azimuth

[273]

{Make both panels of Figure 3D more clearly part of the same panel. The FCM panel might at present as well be part of panel 3F.}

[278]

at the exception -> with the exception

[357]

ssODN and AAV6 edited -> ssODN- and AAV6-edited

[392]

patients derived engrafted -> patient-derived

[449]

incorporated to -> incorporated in

[452]

current HBB -> current DSB-based HBB

[457-464]

{Clearly state the comparisons and inferences drawn. Is the comparison for corrected HbS cells vs β -thal cells, or for correction of HbS cells with vs without HDR-Enh01?}

[Supplementary Figure 2]

homology directed repair -> homology-directed repair {x2}

ddPCR mobilized -> ddPCR for/in/on/using mobilized {x2}

[478]

amounts -> numbers

[488]

AAV6 mediated -> AAV6-mediated

[489]

to IDLV mediated gene delivery which -> to IDLV-mediated gene delivery, which

[507]

edited-HSC -> edited HSC

[528]

HDR edited -> HDR-edited

[568]

AMi. -> Ami

[623][625][633][644][669][672][683][688][714][716][740][754][757]

{Add page numbers to or otherwise correct the reference.}

[Figure 1]

{Panel 1F, align the “+” and “-” symbols underneath the x axis or indicate their range with a bracket.}

[777]

seeding -> seeding,

[782]

{Which parameter was compared using the one-way ANOVA?}

[Figure 2]

[Panel 2A] Bone Marrow human chimerism -> Bone marrow human chimerism

[Panel 2A] Edited Alleles frequencies -> Edited allele frequencies

[784][810]

plerixafor mobilized -> plerixafor-mobilized

[799]

by by -> by

[803]

HDR edited -> HDR-edited {also in panel 2F}

[853]

CD34 expressing -> CD34-expressing

[857]

seeding normalized -> seeding, normalized

[858]

. one-way -> . One-way

[863]

enabling to assess -> enumerating {If that is what is meant.}

[927]

day4 -> day 4

[928]

bone marrow engrafted cells -> bone-marrow-engrafted cells

[Figure 5]

{Panel 5B, correct x axis label} hb expression -> globin expression

{Panel 5E, arrange the legend colors in the same order as the stacked bar elements.}

{Panel 5D, in the legend and M&M, and possibly in the main text, explain which time points were chosen for sampling and (if different from simple averaging of values, line 888) how they were integrated into a single image for each sample.}

[Figure 6]

{Panel 6B, Give the correct statistics test (One-way ANOVA is unlikely.) and specify *** instead of ns.}

[Supplementary Figure 3]

phosphorylation -> phosphorylation {in the figure, panel S3E}

plerixafor mobilized -> plerixafor-mobilized

in vivo injection.). Normalized -> in vivo injection. Normalized

stem cells profiles -> stem cell profiles

selected subpopulation (HSC-enriched) -> selected subpopulation (hematopoietic stem cell (HSC)-enriched) {remove subsequent definition of "HSC" x2 in this legend}

{Remove 2nd definition of "HPC" in this legend}

The color gradient of the dot indicates the average expression of the gene. -> The color of each dot indicates the average expression of the gene according to the scale shown. {Show scale.}

[Supplementary Figure 4]

{See major point 4 for the legend}

Erythroid -> Erythroid {in the figure, panel S4B x3}

FACS -> FCM {in the figure, panel S4B x3; then define "FCM" as flow cytometry in the legend. "FACS" is not what was done here.}

{Choose a color or brightness that more clearly distinguishes the currently light blue arrows from the grey background; in the figure, panel S4H}

{Choose larger x-axis font; in the figure, panel S4I}

CD233+/CD49d- indicating -> CD233+/CD49d- (center plot) indicating

protocol are in shown -> protocol are shown {in the legend}

[Supplementary Figures 5 & 6]

{Clean up the bitmaps shown to either remove or enhance to legible size all labels shown.}

[Supplementary Figures 5 & 7]

{Remove the panel label “A”. Apply “Sentence case” as elsewhere, instead of “Heading Case” to the figure title.}

[Supplementary Figure 6]

{Laudible as the explanatory schematics are above the scatter plot bitmaps, make sure to describe the meaning of the arrows with solid heads for CD233/CD49d and CD36/CD71 plots in S6A.}

[Supplementary Materials & Methods]

[CD34+ cell isolation]

50mL -> 50 mL

To performed CD34+ -> To perform CD34+

[TALENs and HDR donor templates]

viral mediated -> viral-mediated

phosphothioate -> phosphorothioate

1 mg mRNA -> 1 µg mRNA {Or so I would assume.}

4mm -> 4-mm

0.1 mS -> 0.1 ms

0.2 mS -> 0.2 ms

{Define the term “vg”}

500pmol -> 500 pmol

[Evaluation of editing efficiency...]

TALEN cuting -> TALEN cutting

events were performed -> events was performed

ware reported -> were reported

[Deep sequencing / Amplicon sequencing for on and off target detection]

on and off target -> on- and off-target

[HPLC]

Reverse Phase -> Reversed-phase

Cation Exchange -> cation-exchange

[Single -cell RNAseq bioinformatic analysis]

was was carried out -> was carried out

[Transplantation of CD34+ HSPCs...]

intra peritoneal -> intraperitoneal

[Statistical Analysis]

{Specify the test used to decide between parametric and non-parametric testing.}

{This is a much-abbreviated version of my standard response to articles employing SEM as an apparent measure of sample variation. In short, the SD is a descriptive parameter indicating sample variation, while the SEM (usually calculated as SD/\sqrt{n} , with n being the sample size) is an inferential statistics measure indicating the precision of the sample mean estimate. The use of the (tantalisingly smaller) SEM as a measure of sample variability is therefore incorrect. There are many articles on this subject, and the admittedly widespread use of the SEM as a statistical measure of sample variability (instead of reliability of means in inter-group comparisons) in peer-reviewed articles is no vindication of using the SEM in this way, but more likely indicative of the need for clearer guidelines on how to apply statistical measures in biomedical literature.

Even where use of the SEM is argued because of its ability to graphically indicate statistical differences between groups, there the CI would instead be a more informative measure altogether; in particular for small n , typical of most biomedical publications of experimental data, where the CI may become larger than even the SD, and where the SEM may therefore be misleading as a visual clue for statistically significant differences. In the present article most n are around $n=3$, which clearly indicates the use of SEM as problematic.

I therefore strongly recommend (but will not insist on) changing SEM to SD throughout for graphical representations in the main article and in the supplementary data, including for Figure 3H, where the error bars and measure shown are currently not indicated. If the SEM is retained, add a short sentence in M&M justifying or clarifying its use in this study.}

Non-viral DNA delivery coupled to TALEN gene editing efficiently corrects the sickle cell mutation in long-term HSCs. Moiani *et al*

Point-by-point answer to reviewers

Reviewer #1 (Remarks to the Author):

Moiani et al use the TALEN technology to develop a gene editing process for the treatment of sickle cells disease via homology-directed repair. In their work they compare viral vs non viral DNA template for DNA template delivery. Their results convincingly show that non-viral DNA delivery combined to TALEN gene editing reduces toxicity due to viral DNA delivery and results in superior HBB gene correction in long-term HSC. The paper is well written, contains very solid data thanks and experiments have been performed in a rigorous and complete way. Their work represent a significant advance in terms of translational development, considering also that they set up the procedure with GMP grade reagents and in a very thorough way. However, the comparison between viral and non-viral methods for sickle (Romero et al 2019, Patthabi et al, 2019), and the toxicity of AAV have been previously shown and limit somehow the novelty of this paper.

We would like to thank reviewer 1 for their positive feedback about the significance of our work and their fruitful comments.

Major comments

- The authors showed that the optimized editing protocol can decrease indels in the cell product and in the xenograft and thus reduce the probability of generating a beta-thal phenotype in sickle patients. This point is absolutely well taken and their results outperform those reported previously in similar studies (Romero et al; Patthabi et al). Yet, some clarifications should be provided to strengthen it further, as it is one of the most relevant novelty aspects of the paper:

- o Can the authors verify whether the optimized editing protocol is having any impact on the frequency of rearrangements, such as translocations and large deletions? It is possible that NHEJ inhibition favors other DNA repair pathways, such as MMEJ, and thus potentially lead to an increased frequency of genotoxic outcomes.

We agree that our gene editing process optimization could potentially impact the nature of the molecular events promoted by TALEN cleavage activity. We also acknowledge that any genome editing interventions could potentially generate genotoxic outcomes. Our original manuscript already documents the characterization of the on- and off-site TALEN activity and the resulting genomic rearrangements associated to two proximal TALEN cleavages on the same Chromosome 11 (*HBB* and *HBD* loci), obtained in our optimized conditions. However, we recognize that we did not decipher the effect of process optimization, i.e., addition of *Via-Enh01* and *HDR-Enh01*, on such genomic events.

To densify our dataset and address reviewer 1's comment, we first analyzed and compared the molecular events obtained at *HBB* locus by SureSelect and deep sequencing, in the presence of absence of

Via-Enh01 and HDR-Enh01 (n=2 HbSS HSCPs donors, technical quintuplets). Our results enabled to precisely determine the deletion events associated to TALEN cleavage (Fig. 1). These different deletions events were then stratified in two subcategories: non-homologous end joining events (deletions displaying microhomologies < 3 base pairs, named NHEJ hereafter) and microhomology-mediated end joining events (deletions displaying microhomologies \geq 3 base pairs, named MMEJ hereafter).

Fig. 1. Deciphering the effect of Via-Enh01 and HDR-Enh01 on the molecular events promoted by TALEN gene editing at the *HBB* locus. Frequency and length of deletion events obtained by high-throughput DNA sequencing of *HBB* on-site locus after TALEN-*HBB*_{ss} treatment of HbSS patient HSPCs, in presence or absence of HDR-Enh01 and Via-Enh01 (n=2 donor, technical quintuplets). The deletion events displaying microhomology < 3 bp are considered as NHEJ-mediated deletions and those displaying microhomology \geq 3 bp are considered as MMEJ-mediated deletions.

Interestingly, in the absence of Via-Enh01 and HDR-Enh01, the NHEJ-mediated deletions appear significantly more frequent than the MMEJ-mediated deletions (Fig. 2B, 64% and 36% mean deletion events, respectively, Mann Witney non-parametric test, p-value<0.0001) and a NHEJ/MMEJ ratio of 1.8 (Fig. 2C). This trend was found inverted in the presence of Via-Enh01 and HDR-Enh01, with NHEJ-mediated deletions appearing significantly less frequent than the MMEJ-mediated deletions (Fig. 2B, 46% and 54% mean deletion events, respectively, Mann Witney non-parametric test, p-value<0.0001) and a NHEJ/MMEJ ratio of 0.8 (Fig. 2C). Similar results were obtained in an independent experiment performed with a third HbSS HSPCs donor characterized by *HBB*-specific amplicon sequencing (Fig. 2D-E). This specific approach allowed used to obtain a high number of sequence reads focused on *HBB* locus to confirm the first set of data obtained by SureSelect. This trend is consistent with the work of Canny *et al*¹ demonstrating that HDR-Enh01 (named i53 in their manuscript) elicited an increase of CRISPR-CAS9-mediated MMEJ events.

Fig. 2. Deciphering the effect of Via-Enh01 and HDR-Enh01 on the molecular events promoted by TALEN gene editing at the *HBB* locus. **A**, frequency of indels obtained by high-throughput DNA sequencing of TALEN-HBB_{ss} on-target and off-target sites in HbSS patient HSPCs, in presence (+) or absence (-) of HDR-Enh01 and Via-Enh01. **B and C**, frequencies of NHEJ- and MMEJ-dependent deletions and ratio of NHEJ-over MMEJ-dependent events, respectively, detected by high throughput DNA sequencing of *HBB*-specific SureSelect-enriched sequences and computed within all deletion events obtained in presence (+) or absence (-) of HDR-Enh01 and Via-Enh01. On each box plot, the central mark indicates the median value, the bottom and top edges of the box indicate the interquartile range (IQR), and the whiskers represent the maximum and minimum data point. Each point represents the frequency of indels obtained in one independent experiment performed with a given HbSS HSPC donor (n=2, 5 technical replicates ~2500 average usable read/condition). A non-parametric Mann-Whitney test was used for statistical analysis of the data; p-value are indicated. **D and E**, frequencies of NHEJ- and MMEJ-dependent deletions and ratio of NHEJ- over MMEJ-dependent events detected by high throughput DNA sequencing on *HBB*-specific PCR amplicon, respectively, detected and computed within all deletion events obtained in presence (+) or absence (-) of HDR-Enh01 and Via-Enh01. Each point represents the frequency of indels obtained in one experiment performed with one HbSS HSPC donor (n=1, ~30000 average usable reads/condition).

The revised Sureselect dataset analysis is now part of the revised Fig. 7 and for the sake of simplicity, the dataset obtained with the third HbSS HSPCs donor analyzed by *HBB*-specific amplicon sequencing (Fig. 2D and E) are not illustrated.

Second, to avoid missing unpredicted rearrangements associated to the co-editing of *HBB* and *HBD* loci, we further characterized edited HSPCs batches using the CAST-Seq methodology². This unbiased analysis was performed on HSPCs from 3 HbSS patients edited by TALEN-HBB_{ss} in the presence of HDR-Enh01 and Via-Enh01. Our results showed that such treatment elicited deletion, inversion, and translocation events between the *HBB* on-site and *HBD* off-site, without promoting any detectable rearrangement in other part of the genome (Fig. 3). In addition, the identity of the deletion, inversion, and translocation events detected by CAST-SEQ were found consistent with our original predictions, that were used to design our ddPCR probe and primer sets and obtained the results documented in Fig. 7 of our original manuscript.

Fig. 3. Unbiased analysis of the genomic rearrangement in HbSS patient HSPCs edited by TALEN-HBB_{ss} in the presence of HDR-Enh01 and Via-Enh01 by CAST-SEQ methodology. Circos plot representing the editing events occurring at *HBB* on-site (blue) and between *HBB* onsite and *HBD* off-site (pink) in HbSS HSPCs edited in the presence of TALEN-HBB_{ss}, HDR-Enh01 and Via-Enh01 (n=3 donors).

These results were incorporated in the main revised manuscript along with the text below.

“Simultaneous cleavage of HBB and HBD genes is known to elicit genetic rearrangements^{2,3}. We further confirm that aspect in an unbiased and qualitative fashion by using the CAST-Seq methodology² on HSPCs from 3 HbSS patients, edited by TALEN-HBB_{ss} in the presence of HDR-Enh01 and Via-Enh01. Our results showed that such treatment elicited deletion, inversion, and translocation events between the HBB on-site and HBD off-site, without promoting any detectable rearrangement in other part of the genome (Fig. 7d).”

Having confirmed our original predictions of genomic rearrangements associated to TALEN-HBB_{ss} activity in a qualitative and unbiased manner, we then used a rearrangement-specific ddPCR to quantitatively decipher the effect of TALEN, HDR-Enh01 and Via-Enh01 treatments on the translocation, deletion and inversion events occurring between *HBB* and *HBD* loci. This was performed on HbSS patient HSPCs edited with TALEN-HBB_{ss} in the presence or absence of ssODN repair template and in the presence or absence of HDR-Enh01 and Via-Enh01. To evaluate the persistence of such genomic adverse events, we performed this ddPCR analysis on edited cells obtained 4- and 7-days post HSPC thawing as well as those obtained after 16 weeks of engraftment in NBSGW mouse model, as originally explained in the first version of our manuscript (Fig. 4).

As already demonstrated in the original version our manuscript, TALEN-HBB_{ss} cleavage activity promoted formation of deletions, inversions, and translocations between *HBB* and *HBD* loci (Fig. 4C). The frequency of genomic rearrangements was found either unchanged or decreased by the presence of Via-Enh01 and HDR-Enh01. In addition, the frequencies of rearrangement significantly decreased from culture

samples to those obtained from BM of mice, 16 weeks after onset of edited HSPC injection, as noted in our original manuscript.

Fig. 4: Effect of Via-Enh01 and HDR-Enh01 on TALEN-mediated genomic rearrangements after editing at *HBB* locus. **A.** Representative schema of the experience performed to characterize the genomic rearrangements occurring at the *HBB-HBD* locus by ddPCR. ddPCR was performed in multiple HbSS HSPCs edited in the presence or absence of HDR-Enh01 and Via-Enh01 from Day 4 and 7 of liquid culture and 16 weeks after *in vivo* engraftment. **B.** Experimental ddPCR strategy to detect deletion, inversion and translocation events at the *HBB-HBD* locus. Location of TALEN cut site, primers and probes binding are shown. **C.** Frequencies of deletions (left panel), inversions (middle panel), translocations (right panel) at the *HBB-HBD* locus assessed by ddPCR in ssODN edited HSPCs derived from HbSS patients (HbSS) at day 4, day 7 of culture or 16 weeks in hCD45+ bone-marrow-engrafted cells (Wk16). On each box plot, the central mark indicates the median value, the bottom and top edges of the box indicate the interquartile range (IQR), and the whiskers represent the maximum and minimum data point. Each point represents one experiment, performed with a given HbSS HSPC donor in technical triplicate. Mann-Whitney non-parametric unpaired test; *p*-value are indicated.

Similar conclusions were reached with HD HSPCs or HbSS patients' HSPCs edited by our optimized viral or non-viral gene correction process (Fig. 5). Altogether, this comprehensive dataset obtained on multiple HD and HbSS patient HSPC donors and with two format of DNA template (viral and non-viral), suggests that Via-Enh01 and HDR-Enh01 did not potentialize the genomic adverse events occurring

between between *HBB* and *HBD* loci (translocation/deletion/inversion). It also confirmed that such genomic adverse events were selected out as a function of time.

Fig. 5 Effect of Via-Enh01 and HDR-Enh01 on TALEN-mediated genomic rearrangements after editing at *HBB* locus. Frequencies of deletions (top panels), inversions (middle panels), translocations (bottom panels) at the *HBB*-*HBD* locus assessed by ddPCR in ssODN or AAV edited HSPCs derived from healthy donor (HD) or HbSS patients (HbSS) at day 4, day 7 of culture and 16 weeks after NCG engraftment onset, in hCD45+ bone-marrow-engrafted-cells (WK16). On each box plot, the central mark indicates the median value, the bottom and top edges of the box indicate the interquartile range (IQR), and the whiskers

represent the maximum and minimum data point. Each point represents one experiment, performed with a given HD or HbSS HSPC donor in technical triplicate. Mann-Whitney non-parametric unpaired test; *p*-value are indicated (ns = non-significant).

These results were incorporated in the main revised manuscript along with the text below.

“We then quantitatively assessed the frequency of such rearrangements by ddPCR using gDNA from HbSS patients’ HSPCs edited by TALEN-HBB_{ss}, in the presence or absence of ssODN, Via-Enh01 and HDR-Enh01 (Fig. 7e-f and g). TALEN cleavage activity promoted formation of deletions, inversions, and translocations between HBB and HBD with frequencies of 1.3% ± 0.4 (deletion), 0.33% ± 0.04 (inversion) and 0.91% ± 0.71 (translocation), respectively (Fig. 7g, average frequency of events ± SD, obtained at day-4, in the absence of Via-Enh01 and HDR-Enh01). The frequency of genomic rearrangements was found either unchanged or decreased by the presence of Via-Enh01 and HDR-Enh01, and this pattern was observed in the presence or in the absence of the ssODN repair template (Fig. 7g). In addition, it significantly decreased from culture samples to those obtained from BM of mice, 16 weeks after onset of edited HSPC infusion (Fig. 7g). Similar conclusions were reached with HD HSPCs or HbSS patients’ HSPCs edited by our viral gene correction process (Supplementary Fig. 5). This suggested a negative selection for adverse events in LT-HSCs (Fig. 7g, and supplementary Fig. 5, compare day-4 and day-7 to WK16 dataset).”

Nuclease-mediated genome editing has been recently found to promote losses and gains of heterozygosity in primary cells⁴⁻⁹, echoing the question of reviewer 1 “Can the authors verify whether the optimized editing protocol is having any impact on the frequency of rearrangements”.

We thus sought to qualitatively assess that aspect in HbSS patients’ edited cells. For that purpose, HSPCs from 2 HbSS patients were edited by our optimized non-viral gene editing process (TALEN-HBB_{ss}, ssODN, Via-Enh01 and HDR-Enh01) and analyzed by InferCNV scRNA seq, an analytical method enabling the detection of gross genomic rearrangements at a single cell level and in an unbiased manner, by surveying variation of gene transcription across chromosomes. This analysis was performed on edited cells obtained before or after differentiation into RBCs (4- and 15-days post thawing) and compared to untreated cells controls. InferCNV plots showed no evidence of any loss or gain of heterozygosity at the two time points, as opposed to the InferCNV positive control of loss of heterozygosity^{5,6} showing a loss of chromosome 14 induced by TRAC-specific CRISPR-CAS9 treatment of T-cells (see blue lines observed across chromosome 14, Fig. 6). Similar conclusions were reached with HbSS patients’ HSPCs edited by our optimized viral gene editing process (TALEN-HBB_{ss}, AAV, Via-Enh01 and HDR-Enh01, Fig. 7). Altogether, our results suggest that, whatever the DNA template format being used, the TALEN-mediated *HBB* correction treatment incorporating HDR-Enh01 and Via-Enh01, does not generate detectable loss or gain of heterozygosity in non-differentiated and differentiated HbSS patient HSPCs.

Fig. 6. scRNAseq InferCNV analysis of HbSS HSPCs edited with TALEN-HBB_{ss} in the presence of ssODN, HDR-Enh01 and Via-Enh01. **A.** Representative schema of scRNAseq InferCNV to identify the potential large genomic rearrangement including gain and loss of heterozygosity (GOH and LOH, respectively) in HSCP transfected with mRNA encoding TALEN-HBB_{ss}, Via-Enh01, HDR-Enh01 and with ssODN (n=2 HbSS non-mobilized HSPC donors). **B.** Representative schema of experimental design to verify the ability of InferCNV to detect loss of heterozygosity. **C.** InferCNV results obtained from samples gathered 4 days post HSPC thawing (Day 4) and after erythroid differentiation (Day15). Each individual row corresponds to a single cell and each column corresponds to a specific gene and its genomic position, grouped by chromosome. Red color represents increase in gene expression, while blue color represents decrease in gene expression. **D.** InferCNV results obtained from T-cell sample edited by TRAC specific CRISPR-CAS9 and gathered 4 days post PBMC thawing (Day 4). The color code is similar as in C. Typical examples of LOH are illustrated by the blue lines seen across chromosome 14, where the TRAC CRISPR-CAS9 is designed to cut.

A**B**
Fig. 7. scRNAseq InferCNV analysis of HbSS HSPCs edited with TALEN-HbB_{ss} in the presence of AAV, HDR-Enh01 and Via-Enh01. **A** Representative schema of scRNAseq InferCNV to identify the potential large genomic rearrangement including gain and loss of heterozygosity in HSCP, n=2 HbSS donors transfected with mRNA encoding TALEN-HbB_{ss}, Via-Enh01, HDR-Enh01 and transduced with AAV. **B.** InferCNV results obtained from samples gathered 4 days post HSPC thawing (Day 4) and after erythroid differentiation (Day15). Each individual row corresponds to a single cell and each column corresponds to a specific gene and its genomic position, grouped by chromosome. Red color represents increase in gene expression, while blue color represents decrease in gene expression.

These results were incorporated in **Fig. 8** of the main revised manuscript along with the text below.

“Nuclease mediated genome editing has been recently found to promote losses and gains of heterozygosity in primary cells^{4–9}. We thus sought to qualitatively assess that aspect in HbSS patients’ edited cells. For that purpose, HSPCs from 2 HbSS patients were edited by our optimized non-viral gene editing process (TALEN-HbB_{ss}, ssODN, Via-Enh01 and HDR-Enh01) and analyzed by InferCNV scRNAseq (Fig. 8). This analysis was performed on edited cells obtained before or after differentiation into RBCs (4- and 15-days post thawing) and compared to untreated cells controls. InferCNV plots showed no detectable loss or gain of heterozygosity at the two time points (Fig. 8c), in contrast with the InferCNV positive control⁶ showing a loss of chromosome 14 induced by TRAC-specific CRISPR-CAS9 treatment of T-cells (Fig. 8d). Similar conclusions were reached with HbSS patients’ HSPCs edited by our optimized viral gene editing process (TALEN-HbB_{ss}, AAV, Via-Enh01 and HDR-Enh01, Supplementary Fig. 6).”

o In Fig. 6, there is high variability among mice in terms of HDR editing efficiency and indels, ranging from 0 to 60%. Are the mice with the lowest HDR editing efficiency those showing the highest indels frequency, and vice versa? If this is the case, it means that there is a 50% probability of acquiring a beta-thal like phenotype. Viceversa, are the mice with the highest HDR editing also those with the higher indel frequency? This could reflect high variability of the editing procedure and/or extremely low graft clonality, a potentially worrisome finding regarding the safety of this approach.

To address this important point and bring clarity to our *in vivo* dataset illustrated in Fig. 2, we first documented the donor origin of each datapoints illustrated in the Fig. 2B, 2D and 2F of our original manuscript. Then, to answer to reviewer 1 question and render this information easily accessible to the reader, we computed the ratio of HDR/indel alleles obtained for each datapoint/mouse. This ratio allows to visualize at a glance, the respective evolution of Indel and HDR events within the same animal (Fig. 8).

Fig. 8. Long-term engraftment and hematopoietic reconstitution of edited plerixafor- mobilized HSPCs from healthy donors in NCG mice. Frequency of HDR (left plot) or Indels (middle plot) and ratio of HDR/indel (left plot) alleles evaluated in gDNA from hCD45+ cells obtained from BM 16-18 weeks after injection (Output) or in gDNA from HSPCs before injection (Input). Black lines represent median values (n= 18 mice edited group from a total of 3 HSPCs donors differentiated from one another by purple, light and dark blue colors). Two-way ANOVA followed by Bonferroni multi-comparison test; *p*-value are indicated (ns = non-significant).

First, our results show that the ratio of HDR/indel obtained at the input and output are not significantly different from one another, and this, for both non-viral and viral editing processes. This indicates that, overall, both types of events are equally stable during the process of *in vivo* engraftment (whatever gene editing process being used), and that, as mentioned by reviewer 1, “the highest HDR editing also those with the higher indel frequency” and vice versa. There are however subtle differences to be noted when comparing the output of the two processes. We observed that the output ratios obtained for the non-viral editing process remain grouped around the median, whereas those obtained for the viral editing process appear more scattered. When considering the latter process specifically, while HDR/indel ratio around and above 1 could be therapeutically beneficial, low HDR/indel ratio (for instance, those below 0.5), could be potentially worrisome in a safety stem point. This pattern could be due to low graft clonality of viral-edited HSPCs, consistent with their low frequency of engraftment seen in Fig. 2B of our original manuscript, although additional experiments, involving cell barcoding steps, would be needed to definitively conclude on that matter. Because these low ratios appear more frequently in the viral editing process than in its non-viral editing counterparts, our data suggest that that safety profile of the non-viral

edited HSPCs appears preferential. However, the number of animals used for this study doesn't meet the power necessary to reach a definitive conclusion. This interpretation will thus have to be eventually confirmed with a higher number of mice.

Of note, for the sake of consistency, the same dataset treatment and illustrations were applied to Fig. 6, allowing us to reach similar general conclusions regarding the non-viral editing process. These new representations are now implemented in the revised Fig. 2 and 6.

In Fig. 5 and Suppl. Fig. 4, the authors generate and use beta-thal like HSPCs as control of a condition in which high level (>60%) of biallelic knock-out could lead to an undesired phenotype of the erythroid progeny, with pathogenic potential. Yet, this control is somehow an extreme scenario. Using HSPCs edited with the standard editing condition or with previously published editing protocols is the most appropriate control to claim "that our optimized process represents a substantial improvement on existing approaches".

The goal of the β -Thal control was to generate cells harboring the transcriptomic signature of β -Thalassemia, to serve as a reference and gauge the safety profile of TALEN-mediated gene correction processes with viral and non-viral DNA template delivery. Thus, we optimized our TALEN process to promote high knock-out efficiency in HSPC and use this edited batch generated out of two HbSS HSPCs donors, to obtain a β -Thalassemia-like transcriptomic signature, at a single cell level.

Our intention was not to embark in the daunting task of comparing side-by-side, our gene correction approach to others. We understand that the sentence pointed by reviewer 1 does not faithfully represent our aim. To avoid any confusion, this sentence has been removed from our original manuscript.

- In Fig. 2B, one would expect the engraftment of untreated and mock treated animals to be higher than ssODN and AAV ones, in line with the data on colonies in Fig. 1F and with previous publications. Similar observation applies also to Fig 6B. Can the authors add to the text a brief explanation/speculation on why this was not the case? Moreover, since the experiment was performed transplanting cells separately from three different donors, authors could use a color/symbol code to identify mice transplanted with HSPCs from the same donor in each experimental group. This could help interpreting the data.

As stated above, we have now used a color/symbol code to identify mice transplanted with HSPCs from the same donor in each experimental group to help with data interpretation.

While the number of mice in mock and untreated groups is too low to draw robust conclusions, here are the observations we can make:

For donor 1, chimerism was lower in mice receiving untreated cells compared to mice receiving mock-electroporated cells or ssODN-transfected cells.

For donor 2, chimerism values in mice receiving untreated cells and in mice receiving mock-electroporated cells were still in the range of those observed in mice receiving ssODN-transfected cells.

For donor 3, chimerism was variable in mice receiving untreated cells and lower in mice receiving mock-electroporated cells compared to mice receiving ssODN-transfected cells.

Overall, given the low number of mice per donor in control groups and as a corollary, the variability potentially associated to the low number of control mice, we cannot draw definitive conclusions regarding the engraftment of mock and untreated HSPCs relative to ssODN and AAV treated HSPC. Therefore, the only fair comparison is between mice receiving ssODN-treated cells and AAV-treated cells: treatment with ssODN allows a better chimerism compared to treatment with AAV.

However, we agree with the reviewer 1, that a larger fraction of mice have a low chimerism when treated only with electroporation or when not treated at all. We believe that, in our experimental settings, electroporation of the editing tools can affect the proliferation and differentiation of HSPCs (as suggested by our scRNAseq analysis of untreated HSPCs vs cells treated with ssODN or AAV).

- In the untreated control of the scRNA-seq analysis, the % of HSC-enriched cells is very low compared to the treated groups. Can the authors explain and comment on this? While the lower % of quiescent cells could be explained by the more proliferating status of unmanipulated cells (as mentioned in the paper), this observation is more difficult to interpret, despite being in line with the engraftment data shown in Fig. 2B.

While we can't definitively answer to this question, we believe that the electroporation of the editing tools, may impact the proliferation of HSPCs and thus affect the kinetic of differentiation. As the untreated control does not go through the electroporation and doesn't incorporate any editing tool, untouched HSPCs are likely to proliferate and differentiate more than HSPC edited by the viral and non-viral processes.

Minor comments:

- The acronyms HDR-Enh01 and Via-Enh01 should be substituted with the name of the actual protein encoded by the electroporated mRNA.

We renamed these two molecules in a systematic manner for the sake of clarity and readability. The origin of HDR-Enh01 and Via-Enh01 are referenced in the bibliography section and the reader can easily access to the actual sequence of the two proteins in supplementary Table 1. We thus kindly ask to keep this naming strategy in the revised version of the paper.

- The letter "E" for the panel in Fig. 6 is missing.

This is addressed in the revised manuscript.

- Numbers in the statement related to Fig. 6D-E in the main text are wrong.

The numbers were corrected in the text describing the median of edited HSPCs engraftment and of HDR frequencies illustrated in Figure 6. This typo doesn't change our conclusions.

- When discussing base and prime editing strategies, this recent ref could be added: <https://doi.org/10.1038/s41587-023-01915-4>

We thank the reviewer for pointing that important and recent work. This ref is now added in the section discussing the base and prime editors.

Reviewer #2 (Remarks to the Author):

Summary

The present manuscript improves therapeutic double-strand-break-based editing of HSPCs based on the TALEN platform, in the context of sickle-cell disease, using preclinical/clinically relevant materials and methods, and employing comprehensive and cutting-edge analysis technology.

On the plus side, the authors detail TALEN and viability- and HDR-enhancing mRNA sequences, compare viral and non-viral donors to allow reproduction. They explore transition from research- to GMP-suitable reagents and include in their analyses single-cell RNA-Seq and transplantation experiments in immunodeficient mice, besides a range of solid functional analyses and exemplary off-target analyses in therapeutically relevant cells. Efficiencies and tolerability reached and the reported off-target behavior compare very favorably with other recent work on DSB-based therapies.

On the minus side, single-cell RNA-Seq and its conclusions are based on multiplexed sequencing of two sickle-cell patient samples. Also, many of the developments on their own are incremental, and the Results text is at time too speculative, while the Discussion text omits any concrete comparison to efficacy and safety data for competing therapies, e.g. using same-target CRISPR/Cas and DSB-independent platforms.

Overall and despite minor shortcomings, the combination of analyses and improvements presented in the manuscript significantly contributes to our understanding of DSB-based therapies and their greater safety and efficacy. This is all the more important for allowing more generalised statements going forward, the present study being based on data for TALENs instead of the far more commonly used CRISPR/Cas platform.

We thank the reviewer 2 for their positive feedback regarding our significant contribution to the field of DSB-therapies. We hope that the point-by-point answers delineated below, addresses their fruitful questions and comments.

Major Comments

(1)

The use of Via-Enh01 increases survival of cells post-HDR. However, it also increases the likelihood of enriching the edited cell population for p53-pathway-deficient cells. In line with original observations around p53-related changes during DSB-based editing (e.g. PMID 29892067) and with ambitions for clinical application based on GMP-compliant conditions, this should be monitored for the viable post-editing cell population, with and without Via-Enh01, for GMP-compliant culture conditions.

The reviewer 2 raises an important point regarding our study and the field of HSPCs gene therapy in general: the competitive advantage of *p53*^{-/-} over *p53*^{+/+} clones detected after gene editing and over several *ex vivo* cell division cycles. While this phenomenon has been mostly characterized in immortalized hRPE1, hPSC, hiPS and hES cell lines^{10–12}, we believe it was important to verify that aspect in the edited primary HSPCs generated in our study. To address this question, we sought to characterize HSPCs edited by our viral and non-viral engineering process at a genomic and transcriptomic level.

First, we characterized the potential outcome of *p53* mutations in healthy donor HSPCs successfully edited by our GMP compliant viral and non-viral processes in the presence or in the absence of HDR-Enh01 and Via-Enh01 (proofs of editing efficiency could be visualized in **Supplementary Fig. 2a, b** of our revised supplementary manuscript). Mock and edited cells were recovered after a 7-days engineering and culture process and their genomic DNA was analyzed using a *p53*-specific exome sequencing. Our results obtained in three independent HSPCs donor showed no detectable enrichment of *p53* variant with respect to the mock control groups (Fig. 9a). Of note, a single nucleotide *p53* variant (G>C) was identified in the unedited and edited experimental groups of donors 4, 5 and 6. It corresponds to a natural variant frequently identified in healthy donors (ns1042522, worldwide median prevalence of 66.4%, Figure 9b).

Fig 9. Characterization of TALEN-mediated genomic adverse events occurring at *p53* after viral and non-viral editing at HBB locus of HSPCs in GMP compliant conditions. a *p53* exome sequencing obtained from

the genomic DNA of Mock HD HSCPs or of HD HSCPs edited in GMP conditions by the viral and non-viral gene editing process in the presence or absence of HDR-Enh01 and Via-Enh01. Mock and edited cells were gathered 7 days post-editing and their genomic DNA was analyzed using a *p53*-specific exome sequencing. The frequency and exonic position of *p53* variant obtained in each donor (n=3 independent biological experiments and donors) and in each experimental group are indicated. The detection threshold was set to 1% according the *p53* exon sequencing kit's manufacturer guidelines. **b** The G->C *p53* variant identified in the *p53* exon 4 of the Mock and edited experimental groups of donors 4, 5 and 6 corresponds to a natural single nucleotide variant frequently identified in healthy donors (ns1042522, worldwide median prevalence of 66.4%, one dot corresponds to the frequency of prevalence obtained in one SNP database. Database names are indicated in the legend).

Because potential expansion and enrichment of *p53*-deficient clones may arise after several cell divisions, we compared the *p53* exome of Mock HSCPs to the ones obtained for successfully edited HSCPs gathered 4 days post editing and 16 weeks post NCG mouse engraftment onset (proofs of editing could be visualized in the main Fig. 2d, left and middle panels, of our revised manuscript). Our *p53* exome sequencing results obtained with 3 HSPC donors didn't show any detectable enrichment of common driver mutations between mock and edited conditions, using the recommended detection threshold of 1% (Fig. 10). Of note, the single nucleotide *p53* variant ns1042522 was also identified in the mock and edited experimental groups of donors 1 and 3. Together, these data suggest that our optimized *HBB* gene correction process incorporating Via-Enh01 and HDR-Enh01 doesn't generate detectable alteration of *p53* gene.

Fig 10. Characterization of TALEN-mediated genomic adverse events occurring at *p53* after optimized non-viral editing at *HBB* locus of HSCPs and long-term engraftment in NCG mouse model. *p53* exome sequencing obtained from the genomic DNA of Mock HD HSCPs (Mock in vitro) or of HD HSCPs edited in R&D conditions by the optimized non-viral gene editing process and gathered 4 days post editing (edited in vitro) or 16 weeks after NCG mice engraftment onset (edited in vivo). The frequency and exonic position of *p53* variant obtained in each donor (n=3 independent biological experiments and donors) and in each experimental group are indicated. The detection threshold was set to 1% according the *p53* exon sequencing kit's manufacturer guidelines. The G->C *p53* variant identified in the *p53* exon 4 of the Mock

and edited experimental groups of donors 1 and 3 corresponds to a natural single nucleotide variant frequently identified in healthy donors (ns1042522, worldwide median prevalence of 66.4%).

Second, we analyzed the transcriptomic status of edited HSPCs recovered after a short culture process (4 days) and after erythroid differentiation using InferCNV, an analytical method enabling the detection of gross genomic rearrangements at a single cell level and in an unbiased manner, by surveying variation of gene transcription across chromosomes. Our InferCNV dataset of edited cells documented earlier in Fig. 6 and 7 of this rebuttal letter, does not show any detectable loss or gain of heterozygosity in chromosome 17, where *p53* is located, and in any other chromosomes commonly altered, as a bystander effect of *p53* mutations occurrence¹³. This conclusion was reached when analyzing HSPCs few days after their editing (viral and non-viral optimized editing process) and after multiple rounds of cell division during their differentiation in erythroid progenies.

Altogether, our results suggest that the TALEN-mediated gene correction treatment incorporating HDR-Enh01 and Via-Enh01, neither generates detectable loss or gain of heterozygosity nor promotes enrichment of detectable *p53* driver mutation in HSPCs. We added a paragraph in the results section as well as created a new figure (**Fig. 8** of the revised manuscript) documenting the scRNAseq Infer CNV and *p53* exon sequencing obtained in R&D conditions. For the sake of simplicity and readability, the data obtained in GMP conditions were excluded from the revised manuscript.

*“Nuclease mediated genome editing has been recently found to promote losses and gains of heterozygosity in primary cells⁴⁻⁹. We thus sought to qualitatively assess that aspect in HbSS patients’ edited cells. For that purpose, HSPCs from 2 HbSS patients were edited by our optimized non-viral gene editing process (TALEN-HBB_{ss}, ssODN, Via-Enh01 and HDR-Enh01) and analyzed by InferCNV scRNAseq (**Fig. 8**). This analysis was performed on edited cells obtained before or after differentiation into RBCs (4- and 15-days post thawing) and compared to untreated cells controls. InferCNV plots showed no detectable loss or gain of heterozygosity at the two time points (**Fig. 8c**), in contrast with the InferCNV positive control⁶ showing a loss of chromosome 14 induced by TRAC-specific CRISPR-CAS9 treatment of T-cells (**Fig. 8d**). Similar conclusions were reached with HbSS patients’ HSPCs edited by our optimized viral gene editing process (TALEN-HBB_{ss}, AAV, Via-Enh01 and HDR-Enh01, **Supplementary Fig. 6**).*

*Nuclease mediated genome editing was also reported to promote enrichment of *p53*-deficient clones in edited cell population after several rounds of divisions¹⁰⁻¹². While this phenomenon has been mostly characterized in immortalized hRPE1, hPSC, hiPS and hES cell lines, we believe it was important to verify that aspect in the edited primary HSPCs generated in our study, given their potential for clinical applications. To address this question, we sought to characterized HSPCs edited by our non-viral optimized engineering process incorporating HDR-Enh01 and Via-Enh01, by high throughput sequencing of their *p53* exome (**Fig. 8e**). Because potential expansion and enrichment of *p53*-deficient clones may arise after several cell divisions, we compared the *p53* exome of untreated HSPCs to the ones obtained for edited HSPCs gathered 4 days post editing and 16 weeks post NCG mouse engraftment onset. Our *p53* exome sequencing results obtained with 3 HSPC donors didn’t show any detectable enrichment of common driver mutations between untreated and edited conditions, using the recommended detection threshold of 1% (**Fig. 8e**). Of note, a single nucleotide *p53* variant was identified in the unedited and edited experimental groups of donors 1 and 3. It corresponds to a natural variant frequently identified in healthy donors (ns1042522, worldwide median prevalence of 66.4%). Altogether, our results suggest that the TALEN-mediated gene correction treatment incorporating HDR-Enh01 and Via-Enh01, neither generates detectable loss or gain of heterozygosity nor promotes detectable *p53* driver mutation enrichment in HSPCs.”*

(2)

The authors should refrain from conjecture or excessive interpretation in the Results section, as I will illustrate here for three examples.

First, this applies to some statements in lines 251 to 307, where scRNAseq data for n=2 and duplexed sequencing are at times over-interpreted and are generalised without a disclaimer about the potential limitations.

The limitation of our study associated to the low number of donors analyzed (n=2) is now acknowledged in the discussion section using the following sentence: *“Noteworthy, the different conclusions drawn out of our scRNAseq package are based on the transcriptomic analysis of 2 independent HD HSPCs. Additional work using multiple other HD and HbSS patient HSPCs, is now needed to confirm our conclusions on the mode of action of both DNA delivery strategies and on their impact of HSPCs fitness.”*

If such a disclaimer would be better at home in the Discussion, then so would be an interpretation (currently in Results) going from downregulated mitochondrial metabolism and protein synthesis, over preservation of more quiescent primitive cells, to the assertion that “Overall, these data show that the non-viral mediated DNA delivery strategy preserves more engraftment potential” (lines 303 to 304), ahead of any presentation of data for engraftment later in the manuscript.

We agree with reviewer 2 that the results section contains different interpretations that may be best suited for the discussion section, and we did our best to address this point in the revised manuscript.

We removed the sentences below from the results section:

“This is consistent with the fact that our gene editing process decreases the proportion of HSPCs in S phase compared to UT (Fig. 3e-f), preserving more quiescent primitive cells that do not rely on mitochondrial metabolism and oxidative phosphorylation¹⁴ (Supplementary Fig.3f).”

“Overall, these data show that the non-viral mediated DNA delivery strategy preserves more engraftment potential, as evidenced by the preservation of a larger pool of quiescent primitive HSCs with repopulating capacity and by mitigating p53 activation and maintaining greater fitness of ssODN-edited LT-HSCs compared to AAV6.”

“validating its therapeutic potential” was removed from *“Overall, these data confirmed that our gene editing protocol can correct LT-HSCs derived from patients, validating its therapeutic potential.”*

What is more, besides differences in buffer and cell handling, the template delivered by AAV is much longer than the ssODN, which might have toxicity effects on the treated cells that are unrelated to “viral” characteristics.

We agree with reviewer 2 and acknowledge this point in the discussion using the following sentence:

“Differences of editing processes, cells handling, and length of the DNA templates used in each process could be part of such discrepancy, but other factors are also likely to influence the overall functionalities of edited HSPCs. A comprehensive scRNAseq characterization of the gene edited products before in vivo injection allowed us to decipher some of the underlying reasons.”

Overall, current lines 497-534 are a better place for corresponding conclusions and discussion.

Second, the interpretation in the Results section that observing fewer sickling cells after correction confirms that HbA expression suffices to inhibit Hb polymerization (lines 362 to 363) would be more suited for the Discussion, besides ignoring the likely more fundamental effect of corrected cells producing less HbS.

We agree with this point. This sentence was shortened to only described the results. The comments made on HbA expression and Hb polymerization was move to the discussion section for the sake of consistency.

“Genetic correction of β s in HSPCs from HbSS patient HSPCs translated into an efficient rescue of HbA (Supplementary Fig. 4c, $55.4\% \pm 5.5$ of HbA rescued in RBCs, average frequency \pm SD) and a significant decrease of HbS (Supplementary Fig. 4c, from $93.6\% \pm 2.9$ to $24.5\% \pm 14.4$, average frequency of HbS \pm SD, in RBCs), and of sickle RBCs (Fig. 5c, from $96.2\% \pm 1.2$ to $40.5\% \pm 14.1$, average frequency of sickle RBCs \pm SD) confirming that HbA expression was sufficient to inhibit the Hb polymerization and the sickling process.”

Third, understandably the manuscript does not take a range of major existing protocols (e.g. PMID 37294917) to compare their impact on transcription profiles, but nevertheless claims in the Results that similarity of the profile of corrected cells to that of Mock EP cells (i.e. uncorrected sickling cells) and dissimilarity to that of created β -thal cells (and here, why one state would be preferable to the other is not clear), would be “confirming that [the authors’] optimized process represents a substantial improvement on existing approaches.”

As discussed earlier, our intention was not to embark in the daunting task of comparing side-by-side, our gene correction approach to others. We understand that the sentence pointed by reviewers 1 and 2 does not faithfully represent our aim. To avoid any confusion, this sentence has been removed from our original manuscript.

Altogether, the authors should critically re-read their Results section and reduce or move speculative statements that will otherwise undermine the credibility of their work.

(3)

The Discussion should consider, at least briefly, general efforts of enhancing HDR-based repair (e.g. PMID 34713224) and consider the best as well as the worst indel frequencies that have been observed in (not necessarily HbS) HSPCs, and it should draw more specific comparisons to more recent DSB-based studies addressing recombination events (e.g. PMID 37070192) as well as to DSB-independent alternatives (as found in current references 64-66), beyond what is currently presented in lines 541-550 and before a combined summary (“In summary” line 535) and conclusion (“To conclude” line 551) of the findings. This could also consider the genotoxicity (e.g. PMID 37679541) of DSB-less systems. In the same vein and possibly juxtaposed to results achieved by other systems, the Discussion should reiterate key percentages achieved in terms of HDR-based correction, chimerism and their combination, to allow the conclusion in lines 538-540 that the current work supports therapeutic application of TALEN-based editing for SCD.

Some of the editing results obtained with other approaches were already mentioned in the discussion, however, we understand that it doesn’t illustrate the range of editing activities obtained by the

different groups working on HSPCs gene editing. We thus densify the discussion section and added the most significant contributions to the field, documenting the worst and the best editing activities obtained in the past recent years. We also documented the general efforts made to enhance HDR-based repair using different attributes including small molecules, mRNA or peptides, new chimeric engineered nucleases fused to DNA repair modulator domains as well as processes developed for temporal control of DNA repair. The paragraph below was added in the discussion:

“Using HDR-Enh01 as NHEJ inhibitor is not the only strategy to improve the ratio of HDR/indel during gene correction/insertion processes. Indeed, several encouraging approaches, using small molecules¹⁵⁻¹⁷, engineered nucleases fused to NHEJ inhibitor protein domains^{18,19} or temporal control of DNA repair^{20,21} where recently reported to achieve this goal to variable extents in cells lines, primary differentiates cells and HSPCs. In the context of our study, the inhibition of NHEJ-mediated inactivation of HBB by HDR-Enh01 presents an advantage over DSB-based HBB correction strategies reported earlier²²⁻²⁷. Actually, multiple former studies performed with different engineered nucleases specific for HBB locus (CRISPR-CAS9, and Zinc Finger nuclease) and different HBB repair templates (AAV, IDLV and ssODN), report a wide range of absolute frequencies of HDR and Indels at HBB locus (from 10% to 65-70%, and from 10% to 60%, respectively) and HDR/indel ratio < 1 in most of the cases²²⁻²⁷. Together, these earlier works indicate that the nature of the engineer nuclease, the DNA repair template, the handling of the cells and the timing of the gene editing process, markedly influence the therapeutic potential as well as the safety of edited HSPCs.”

In addition, we reiterated the key values obtained throughout the manuscript to better justify our final conclusions about the therapeutic potential of our approach, by adding the paragraph below in the discussion:

*“Our preferred optimized protocol, based on TALEN and ssODN, achieved high gene correction efficiencies in vitro (**Fig. 1b**, 41.5% ± 5.3 HDR and 19.1% ± 2.1 Indels, average frequency ± SD) and in vivo (**Fig 2b and d**, 29.2% ± 8.7 HDR, 20.6% ± 8 Indels, average frequency ± SD and 58.2% median engraftment) using PLX-mobilized HSPCs from HD. These results were confirmed in vitro (**Fig. 4b and Fig. 6d**, 44.3% ± 5 and 49.3% ± 4 HDR, 21.4% ± 2.3 and 23.7% ± 1 Indels, average frequency ± SD) and in vivo (**Fig. 6d**, 21.0% ± 18.6 HDR, 13.5% ± 13.1 indels, average frequency ± SD and 23.2% median frequency of engraftment) using non-mobilized HSPCs from HbSS patients. Genetic correction of β^s in HSPCs form HbSS patient HSPCs translated into an efficient rescue of HbA (**Supplementary Fig. 4c**, 55.4% ± 5.5 of HbA rescued in RBCs, average frequency ± SD) and a significant decrease of HbS (**Supplementary Fig. 4c**, from 93.6% ± 2.9 to 24.5% ± 14.4, average frequency of HbS ± SD, in RBCs), and of sickle RBCs (**Fig. 5c**, from 96.2% ± 1.2 to 40.5% ± 14.1, average frequency of sickle RBCs ± SD) confirming that HbA expression was sufficient to inhibit the Hb polymerization and the sickling process. Because one corrected HBB allele is sufficient to generate functional RBCs and because a threshold of 20% donor chimerism in the bone marrow is necessary for clinical benefit²⁸⁻³⁰, this work provides evidence that coupling non-viral DNA delivery to TALEN editing has the potential to elicit positive therapeutic outcome for SCD patients.”*

Finally, while DSB-independent alternatives were already cited in the discussion to acknowledge their emergent contribution to the field, we incorporated new references that state important works documenting the development and assessment of such tools, in the field of Gene Therapy (see below).

“For this preclinical proof of concept, we chose the TALEN gene editing technology because of its high specificity and editing efficiency. However, other gene editing tools could theoretically be incorporated in a similar protocol³¹⁻³⁴. Indeed, ex vivo gene therapy strategies based on DSB-free gene editing tools including base and prime editors, also displayed a potential clinical benefit for the treatment of SCD. Base

editing at HBB locus was reported to elicit an average of 44-80% conversion of β^S to β^G allele (Makassar β -globin allele), with indels ranging from 1.2 to 2.8% and bystander editing events <2%³¹, while prime editing elicited an average of 27% conversion of β^S to β allele with an average of indels of 4.6% in plerixafor-mobilized HbSS patient HSPCs in vitro³². Conversions events were found to be maintained after engraftment in immunodeficient mice model indicated that both editing strategies hold therapeutic potential for SCD treatment^{31,32}. However, while these strategies offer a better edit-to-indel ratio, the genotoxic consequences of their DNA and RNA off-target activity has only been recently started to be characterized^{31,32,35-37}. Given the high level of LT-HSC targeting obtained by gene correction or base/prime editor strategies for SCD, the strategy conferring the safest and most therapeutic benefit for the treatment of SCD will have to be determined in clinical trials.”

(4)

[Supplementary Fig. 4]

A major point merely for its importance to be fixed, rather than for its impact on the data, make sure to consider what appear to be late rearrangements of the figure panels in the labels of the legend and in the main text. It appears that panel S4A is currently described as panel S4D, S4C as S4B and S4D as S4C.

This point is now addressed in the figure caption.

Minor Comments

General remarks:

This is a single-pass correction, with apologies for errors inadvertently introduced in the process.

We thank the Reviewer 2 for his thorough review of the manuscript typo and approximations as it helps significantly improving its rigor and quality.

Throughout, use “FACS” only for cell sorting applications and instead apply “FCM” for mere analyses by flow cytometry, in text, figures and figure legends. This applies e.g. in the figure, panel S4B x3 and in the legends of Fig. S5 – S8.

This point is addressed in the manuscript, in the figures and in the figure legends.

Throughout, it appears that “template” and “templates” would be the conventional terms to use in place of “matrix” and “matrices”, respectively, starting from line 161.

This point is addressed in the main manuscript and in the supplementary data file.

Throughout, use μg instead of ug and μL instead of uL.

This point is addressed in the main manuscript and in the supplementary data file.

Define the precise meaning of β^0 , which is currently defined as “indel allele” and the meaning of an “indel allele” throughout the manuscript. E.g, are these out-of-frame indels or 21+ nt indels (to go by the definition of Synthego’s KO Score) or any indels?

We systematically consider Indels as insertions and deletions of 1 to multiple base pairs without discriminating between in-frame (multiple of 3 bp) and out-of-frame mutational events. This is now specified in the following sentence positioned at the beginning of the revised results section: *“However, this correction was associated with a substantial frequency of insertions and deletions of 1 to multiple base pairs (indels) at the HBB locus and an HDR/indel ratio close to 1 with both approaches (Supplementary Fig. 1E).”*

In the figure legends, separating “n=3” or the like off as separate sentences is not suitable; rephrase suitably throughout, e.g.:

- “as mean \pm SEM. n=3. Two-way ANOVA” -> “as mean \pm SEM, for n=3. Two-way ANOVA” or “as mean \pm SEM (n=3). Two-way ANOVA”
- “as mean \pm SEM. n=3 independent biological experiments” -> “as mean \pm SEM, for n=3 independent biological experiments”

Also, keep “n” in minor case (e.g. in the legend for S4E)

This point is addressed throughout the main manuscript.

Throughout, use “Hb” in the main text instead of “hemoglobin” after its initial definition.

This point is addressed in the text.

For the following, text in [square brackets] indicates line numbers (for the main article PDF) or section/figure labels (for supplementary materials), and {curved braces} indicate notes; other text indicates requested text changes (old -> new) or insertions.

[1]

correct -> corrects

This point is addressed.

[49]

proportion -> proportions

This point is addressed.

[50]

Therefore -> This suggests that {This is open to interpretation.}

We kept this word as is.

[51]

due to -> linked to

This point is addressed.

[61]

βS -> βS (NM_000518.4 (HBB):c.20A>T)

This point is addressed.

[62]

{Define sickle hemoglobin here instead of line 79.}

This point is addressed.

[69][101]

ex-vivo -> ex vivo {Latin compound adjectives usually remain without hyphens.}

This point is addressed.

[79]

Adeno Associated Virus 6 -> adeno-associated virus 6 vectors {See also line 160.}

This point is addressed.

[80]

single stranded -> single-stranded

This point is addressed.

[89]

concerns of AAV6 -> concerns over AAV6

This point is addressed.

[96]

corrected HBB alleles -> corrected, HBB alleles

This point is addressed.

[112]

long-term repopulating stem cells (LT-HSCs) -> long-term hematopoietic stem cells {For consistency and in line with the abbreviation. While “long-term–repopulating hematopoietic stem cells” would be a more precise term, elsewhere in this article, including in the title, “long-term hematopoietic stem cell” is used, which is of equally common usage in the field.}

This point is addressed.

[126-148]

End sentences with a full stop.

Full stops were added in this section.

[132][347]

Cation Exchange -> cation-exchange

Reverse Phase -> Reversed-phase

These points were addressed.

[141]

{Specify by rephrasing here and in the supplemental M&M section whether busulfan-based conditioning was applied only to NCG mice, or to both NCG and NBSGW mice.}

This point is addressed.

[148]

{Supply the GEO accession number.}

In the highly competitive domain of Gene Therapy, this dataset is highly sensitive because it can be used to identify gene perturbations associated to the gene editing process and delineate multiple novel interventional optimizations. It thus represents an intellectual property potential that is of high importance for Collectis. We thus kindly ask to keep this raw Transcriptomics data internal, until Collectis files all the patent(s) associated to it. We propose to add the sentence: "Transcriptomic data will be available upon request". We hope the reviewer 2 understands the proposal made by Collectis in the highly competitive nature of the Gene Therapy field.

[160]

adeno-associated viral vector (AAV6) -> AAV6

This point is addressed.

[164]

HBB gene correction -> β S gene correction

This point is addressed throughout the manuscript.

[165]

digital droplet -> digital droplet PCR

This point is addressed.

[168]

mRNA-encoding TALEN -> TALEN-encoding mRNA

This point is addressed.

[172]

indels -> insertions and deletions (indels)

This point is addressed.

[173][196][Figure 1C in the figure]

HDR/Indel -> HDR/indel {Make sure to correct the plural version of “HDR/Indels” to singular throughout.}

This point is addressed throughout the manuscript, figures, and figure captions.

[179]

HDR-Enh0129–31 which -> HDR-Enh0129–31, which

This point is addressed.

[204]

Also give the SD (or, if it is retained, the SEM) for the average percentage values (n=9), to give an idea of the sample variation in the main text.

This point is addressed. And the SD was represented in all figures and documented in figure captions and in main text.

[209]

(> 95% mean CD34+ cells assessed by flow cytometry) -> (> 95% mean CD34+ cells), as assessed by flow cytometry 4 days after editing {Referral to the time point is critical in the context of earlier measurements of viability.}

This point is addressed.

[215-218]

{This is confusing. Rephrase to clarify which number(s) refer to clonogenic potential and which to “mild reduction”. Is this a reduction in the BFU-E percentage or a reduction in the overall yield of colonies in the assay, and in the latter case, what 100% control value does the percentage refer to?}

We remained factual and quantitative and rephrased this section with the sentence below:

“Finally, our gene editing process achieved high frequency of HDR in burst-forming unit-erythroid (BFU-E) progenitors (Fig. 1e, 51.3% ± 7.1 and 68.7% ± 5.0 average frequency of HDR alleles for ssODN and AAV6, respectively), although the process resulted in a 2- and 1.4-fold decrease in clonogenic potential in AAV- and ssODN-edited HSPCs, respectively, compared to untreated control (Fig. 1f, 32.8% ± 6.3 and 48.8% ± 6.0 average frequency of CFU ± SD for AAV and ssODN edited HSPCs, respectively, vs 68.5% ± 8.8 average frequency of CFU ± SD for untreated).”

[221]

high level -> frequent

This point is addressed.

[223]

long-term engrafting HSCs -> LT-HSCs {See also line 112.}

This point is addressed throughout the manuscript.

[226][403]

long-term repopulating hematopoietic stem cells (LT-HSCs) -> LT-HSCs {See also line 112.}

This point is addressed throughout the manuscript.

[235]

consisted of -> generated/produced

This point is addressed.

[239]

respectively – lower than the relative input but less pronounced than AAV (45% and 50% median HDR frequency, respectively) – and no -> respectively, and a smaller drop in HDR frequency compared to the in vitro input frequency of 45% for dsODN vs. 50% for AAV. No

This part has been rephrased for the sake of clarity.

[244]

13.1 -> 13.1%#

This point is addressed.

[245][246]

respectively -> respectively,

This point is addressed.

[248]

in term of -> in terms of

This point is addressed.

[260]

Azymuth -> Azimuth

This point is addressed.

[273]

{Make both panels of Figure 3D more clearly part of the same panel. The FCM panel might at present as well be part of panel 3F.}

This point is addressed by adding a vertical line and a title “Correlation engraftment with FCM and scRNA seq” to link the two plots.

[278]

at the exception -> with the exception

This point is addressed.

[357]

ssODN and AAV6 edited -> ssODN- and AAV6-edited

This point is addressed.

[392]

patients derived engrafted -> patient-derived

This point is addressed.

[449]

incorporated to -> incorporated in

This point is addressed.

[452]

current HBB -> current DSB-based HBB

This point is addressed.

[457-464]

{Clearly state the comparisons and inferences drawn. Is the comparison for corrected HbS cells vs β -thal cells, or for correction of HbS cells with vs without HDR-Enh01?}

[Supplementary Figure 2]

homology directed repair -> homology-directed repair {x2}

This point is addressed.

ddPCR mobilized -> ddPCR for/in/on/using mobilized {x2}

This point is addressed.

[478]

amounts -> numbers

This point is addressed.

[488]

AAV6 mediated -> AAV6-mediated

This point is addressed.

[489]

to IDLV mediated gene delivery which -> to IDLV-mediated gene delivery, which

This point is addressed.

[507]

edited-HSC -> edited HSC

This point is addressed.

[528]

HDR edited -> HDR-edited

This point is addressed.

[568]

AMi. -> Ami

Ami stands for Annarita Miccio. This was kept as is.

[623][625][633][644][669][672][683][688][714][716][740][754][757]

{Add page numbers to or otherwise correct the reference.}

We thank the reviewer for carefully reviewing the references that contained some errors in our original manuscript. They were all reviewed and corrected accordingly.

[Figure 1]

{Panel 1F, align the “+” and “-” symbols underneath the x axis or indicate their range with a bracket.}

This point is addressed.

[777]

seeding -> seeding,

This point is addressed.

[782]

{Which parameter was compared using the one-way ANOVA?}

[Figure 2]

[Panel 2A] Bone Marrow human chimerism -> Bone marrow human chimerism

This point is addressed.

[Panel 2A] Edited Alleles frequencies -> Edited allele frequencies

This point is addressed.

[784][810]

plerixafor mobilized -> plerixafor-mobilized

This point is addressed.

[799]

by by -> by

This point is addressed.

[803]

HDR edited -> HDR-edited {also in panel 2F}

This point is addressed.

[853]

CD34 expressing -> CD34-expressing

This point is addressed.

[857]

seeding normalized -> seeding, normalized

This point is addressed.

[858]

. one-way -> . One-way

This point is addressed.

[863]

enabling to assess -> enumerating {If that is what is meant.}

We agree with this word. It is changed in the text.

[927]

day4 -> day 4

This point is addressed.

[928]

bone marrow engrafted cells -> bone-marrow-engrafted cells

This point is addressed.

[Figure 5]

{Panel 5B, correct x axis label} hb expression -> globin expression

This point is addressed.

{Panel 5E, arrange the legend colors in the same order as the stacked bar elements.}

This point is addressed.

{Panel 5D, in the legend and M&M, and possibly in the main text, explain which time points were chosen for sampling and (if different from simple averaging of values, line 888) how they were integrated into a single image for each sample.}

This point is addressed in the following sentence added in the manuscript: *“UMAP (Uniform Manifold Approximation and Projection) plot comprising aggregated 5’scRNAseq data from differentiated erythroid cells analyzed 11 and 13 days post differentiation onset.”*

[Figure 6]

{Panel 6B, Give the correct statistics test (One-way ANOVA is unlikely.) and specify *** instead of ns.}

We apologize for this error and substituted a Mann-Wittney non-parametric test to the One-way ANOVA. The p -value are indicated in this figure and in all other figures, as required by Nature Communications standard editorial guidelines. For information, in this figure, the p -value=0.44.

[Supplementary Figure 3]

phosphorylation -> phosphorylation {in the figure, panel S3E}

This point is addressed in the hitmap.

plerixafor mobilized -> plerixafor-mobilized

This point is addressed.

in vivo injection.). Normalized -> in vivo injection. Normalized

This point is addressed.

stem cells profiles -> stem cell profiles

This point is addressed.

selected subpopulation (HSC-enriched) -> selected subpopulation (hematopoietic stem cell (HSC)-enriched) {remove subsequent definition of “HSC” x2 in this legend}

This point is addressed.

{Remove 2nd definition of “HPC” in this legend}

This point is addressed.

The color gradient of the dot indicates the average expression of the gene. -> The color of each dot indicates the average expression of the gene according to the scale shown. {Show scale.}

This point is addressed.

[Supplementary Figure 4]

{See major point 4 for the legend}

Erythroid -> Erythroid {in the figure, panel S4B x3}

This point is addressed. The figure was also simplified for the sake of clarity.

FACS -> FCM {in the figure, panel S4B x3; then define “FCM” as flow cytometry in the legend. “FACS” is not what was done here.}

This point is addressed.

{Choose a color or brightness that more clearly distinguishes the currently light blue arrows from the grey background; in the figure, panel S4H}

We tried to do our best to improve this point and hope that the reviewer agrees with our choice.

{Choose larger x-axis font; in the figure, panel S4I}

This point is addressed.

CD233+/CD49d- indicating -> CD233+/CD49d- (center plot) indicating

This point is addressed.

protocol are in shown -> protocol are shown {in the legend}

This point is addressed.

[Supplementary Figures 5 & 6]

{Clean up the bitmaps shown to either remove or enhance to legible size all labels shown.}

We tried to do our best to improve the readability of the main cellular population labels.

[Supplementary Figures 5 & 7]

{Remove the panel label “A”. Apply “Sentence case” as elsewhere, instead of “Heading Case” to the figure title.}

This point is addressed.

[Supplementary Figure 6]

{Laudible as the explanatory schematics are above the scatter plot bitmaps, make sure to describe the meaning of the arrows with solid heads for CD233/CD49d and CD36/CD71 plots in S6A.}

This point is addressed.

[Supplementary Materials & Methods]

[CD34+ cell isolation]

50mL -> 50 mL

This point is addressed.

To performed CD34+ -> To perform CD34+

This point is addressed.

[TALENs and HDR donor templates]

viral mediated -> viral-mediated

This point is addressed.

phosphothioate -> phosphorothioate

1 mg mRNA -> 1 µg mRNA {Or so I would assume.}

4mm -> 4-mm

0.1 mS -> 0.1 ms

0.2 mS -> 0.2 ms

{Define the term “vg”}

500pmol -> 500 pmol

These points above are addressed.

[Evaluation of editing efficiency...]

TALEN cuting -> TALEN cutting

events were performed -> events was performed

were reported -> were reported

These points above are addressed.

[Deep sequencing / Amplicon sequencing for on and off target detection]
on and off target -> on- and off-target

This point is addressed.

[HPLC]
Reverse Phase -> Reversed-phase
Cation Exchange -> cation-exchange

This point is addressed.

[Single -cell RNAseq bioinformatic analysis]
was was carried out -> was carried out

This point is addressed.

[Transplantation of CD34+ HSPCs...]
intra peritoneal -> intraperitoneal

This point is addressed.

[Statistical Analysis]

{Specify the test used to decide between parametric and non-parametric testing.}

{This is a much-abbreviated version of my standard response to articles employing SEM as an apparent measure of sample variation. In short, the SD is a descriptive parameter indicating sample variation, while the SEM (usually calculated as SD/\sqrt{n} , with n being the sample size) is an inferential statistics measure indicating the precision of the sample mean estimate. The use of the (tantalisingly smaller) SEM as a measure of sample variability is therefore incorrect. There are many articles on this subject, and the admittedly widespread use of the SEM as a statistical measure of sample variability (instead of reliability of means in inter-group comparisons) in peer-reviewed articles is no vindication of using the SEM in this way, but more likely indicative of the need for clearer guidelines on how to apply statistical measures in biomedical literature.

Even where use of the SEM is argued because of its ability to graphically indicate statistical differences between groups, there the CI would instead be a more informative measure altogether; in particular for small n , typical of most biomedical publications of experimental data, where the CI may become larger than even the SD, and where the SEM may therefore be misleading as a visual clue for statistically significant differences. In the present article most n are around $n=3$, which clearly indicates the use of SEM as problematic.

I therefore strongly recommend (but will not insist on) changing SEM to SD throughout for graphical representations in the main article and in the supplementary data, including for Figure 3H, where the

error bars and measure shown are currently not indicated. If the SEM is retained, add a short sentence in M&M justifying or clarifying its use in this study.}

We agree with reviewer 2 and changed SEM to SD throughout the figure section, in the main text and in the supplementary data section.

References

1. Canny, M. D. *et al.* Inhibition of 53BP1 favors homology-dependent DNA repair and increases CRISPR-Cas9 genome-editing efficiency. *Nat. Biotechnol.* **36**, 95–102 (2018).
2. Turchiano, G. *et al.* Quantitative evaluation of chromosomal rearrangements in gene-edited human stem cells by CAST-Seq. *Cell Stem Cell* **28**, 1136–1147.e5 (2021).
3. Long, J. *et al.* Characterization of Gene Alterations following Editing of the β -Globin Gene Locus in Hematopoietic Stem/Progenitor Cells. *Mol. Ther.* **26**, 468–479 (2018).
4. Cullot, G. *et al.* CRISPR-Cas9 genome editing induces megabase-scale chromosomal truncations. *Nat. Commun.* **10**, 1136 (2019).
5. Tsuchida, C. A. *et al.* Mitigation of chromosome loss in clinical CRISPR-Cas9-engineered T cells. *Cell* **186**, 4567–4582 (2023).
6. Nahmad, A. *et al.* Frequent aneuploidy in primary human T cells after CRISPR–Cas9 cleavage. *Nat. Biotechnol.* **40**, 1807–1813 (2022).
7. Alanis-Lobato, G. *et al.* Frequent loss of heterozygosity in CRISPR-Cas9-edited early human embryos. *PNAS* **118**, e2004832117 (2021).
8. Leibowitz, M. L. *et al.* Chromothripsis as an on-target consequence of CRISPR–Cas9 genome editing. *Nat. Genet.* **53**, 895–905 (2021).
9. Papathanasiou, S. *et al.* Whole chromosome loss and genomic instability in mouse embryos after CRISPR-Cas9 genome editing. *Nat. Commun.* **12**, 5855 (2021).
10. Merkle, F. T. *et al.* Human pluripotent stem cells recurrently acquire and expand dominant negative P53 mutations. *Nature* **545**, 229–233 (2017).
11. Ihry, R. J. *et al.* p53 inhibits CRISPR–Cas9 engineering in human pluripotent stem cells. *Nat. Med.* **24**, 939–946 (2018).
12. Haapaniemi, E., Botla, S., Persson, J., Schmierer, B. & Taipale, J. CRISPR–Cas9 genome editing induces a p53-mediated DNA damage response. *Nat. Med.* **24**, 927–930 (2018).
13. Rodriguez-Meira, A. *et al.* Single-cell multi-omics identifies chronic inflammation as a driver of TP53-mutant leukemic evolution. *Nat. Genet.* **55**, 1531–1541 (2023).
14. Filippi, M.-D. & Ghaffari, S. Mitochondria in the maintenance of hematopoietic stem cells: new perspectives and opportunities. *Blood* **133**, 1943–1952 (2019).
15. Wimberger, S. *et al.* Simultaneous inhibition of DNA-PK and Pol θ improves integration efficiency and precision of genome editing. **14**, 4761 (2023).
16. Selvaraj, S. *et al.* High-efficiency transgene integration by homology-directed repair in human primary cells using DNA-PKcs inhibition. *Nat. Biotechnol.* (2023) doi:10.1038/s41587-023-01888-4.
17. Stieger, K. *et al.* Global and Local Manipulation of DNA Repair Mechanisms to Alter Site-Specific Gene Editing Outcomes in Hematopoietic Stem Cells. *Front. Genome Ed.* **2**, 601541 (2020).
18. Jayavaradhan, R. *et al.* CRISPR-Cas9 fusion to dominant-negative 53BP1 enhances HDR and inhibits NHEJ specifically at Cas9 target sites. *Nat. Commun.* **10**, 2866 (2019).

19. Carusillo, A. *et al.* A novel Cas9 fusion protein promotes targeted genome editing with reduced mutational burden in primary human cells. *Nucleic Acids Res.* **51**, 4660–4673 (2023).
20. Lomova, A. *et al.* Improving Gene Editing Outcomes in Human Hematopoietic Stem and Progenitor Cells by Temporal Control of DNA Repair. *Stem Cells* **37**, 284–294 (2019).
21. Shin, J. J. *et al.* Controlled Cycling and Quiescence Enables Efficient HDR in Engraftment-Enriched Adult Hematopoietic Stem and Progenitor Cells. *Cell Rep.* **32**, (2020).
22. Hoban, M. D. *et al.* Correction of the sickle cell disease mutation in human hematopoietic stem/progenitor cells. (2015) doi:10.1182/blood-2014.
23. Dewitt, M. A. *et al.* Selection-free Genome Editing of the Sickle Mutation in Human Adult Hematopoietic Stem/Progenitor Cells HHS Public Access. *Sci Transl Med* **8**, 360–134 (2016).
24. Patabhi, S. *et al.* In Vivo Outcome of Homology-Directed Repair at the HBB Gene in HSC Using Alternative Donor Template Delivery Methods. *Mol Ther Nucleic Acids* **17**, 277–288 (2019).
25. Park, S. H. *et al.* Highly efficient editing of the β -globin gene in patient-derived hematopoietic stem and progenitor cells to treat sickle cell disease. *Nucleic Acids Res.* **47**, 7955–7972 (2019).
26. Magis, W. *et al.* High-level correction of the sickle mutation is amplified in vivo during erythroid differentiation. *iScience* **25**, 104374 (2022).
27. Lattanzi, A. *et al.* Development of β -globin gene correction in human hematopoietic stem cells as a potential durable treatment for sickle cell disease. *Sci. Transl. Med.* **13**, eabf2444 (2021).
28. Walters, M. C. *et al.* Stable mixed hematopoietic chimerism after bone marrow transplantation for sickle cell anemia. *Biol. Blood Marrow Transplant.* **7**, 665–673 (2001).
29. Magnani, A. *et al.* Extensive multilineage analysis in patients with mixed chimerism after allogeneic transplantation for sickle cell disease: insight into hematopoiesis and engraftment thresholds for gene therapy. *Haematologica* **105**, 1240–1247 (2020).
30. Fitzhugh, C. D. *et al.* At least 20% donor myeloid chimerism is necessary to reverse the sickle phenotype after allogeneic HSCT. *Blood* **130**, 1946–1948 (2017).
31. Newby, G. A. *et al.* Base editing of haematopoietic stem cells rescues sickle cell disease in mice. *Nature* **595**, 295–302 (2021).
32. Everette, K. A. *et al.* Ex vivo prime editing of patient haematopoietic stem cells rescues sickle-cell disease phenotypes after engraftment in mice. *Nat. Biomed. Eng.* **7**, 616–628 (2023).
33. Chu, S. H. *et al.* Rationally Designed Base Editors for Precise Editing of the Sickle Cell Disease Mutation. *Cris. J.* **4**, 169–177 (2021).
34. Brusson, M., Antoniou, P. & Miccio, A. Base and Prime Editing Technologies for Blood Disorders. *Front. Genome Ed.* **1**, 618406 (2021).
35. Fiumara, M. *et al.* Genotoxic effects of base and prime editing in human hematopoietic stem cells. *Nat. Biotechnol.* (2023) doi:10.1038/s41587-023-01915-4.
36. Liang, S.-Q. *et al.* Genome-wide profiling of prime editor off-target sites in vitro and in vivo using PE-tag. *Nat. Methods* **20**, 898–907 (2023).

37. Anzalone, A. V *et al.* Search-and-replace genome editing without double-strand breaks or donor DNA. *Nature* **576**, 149–157 (2019).

REVIEWERS' COMMENTS

Reviewer #1 (Remarks to the Author):

The authors did an excellent work in responding to all issues and the paper can be accepted in my opinion. It is rigorous and solid and worth publishing.

Reviewer #2 (Remarks to the Author):

Summary

The authors have made painstaking efforts experimentally and in writing to address the numerous points raised by this reviewer in their first round of revisions. The manuscript has greatly improved in the process and pending the corrections indicated below will be a solid addition and point of reference for others.

Major Comments

(1) The response to reviewers in reference to potential p53 enrichment (Major Comment 1 for the original submission) refers to and shows a legend and panels 9a and 9b for a new Figure 9, and a new Figure 10. However, the new manuscript only contains 8 Figures, including image material and legend text, shown for Figure 10 in the response letter, as new panel 8e. The new manuscript or supplementary file do not refer to Figures 9 or 10 in their text, and text and image material for proposed Figure 9 are missing. Moreover, proposed Figure 9 shows data for Donors 4 to 6, while the associated legend text in the response letter refers to Donors 3 to 5. I downloaded materials twice, initially assuming a confusion of file versions, but this appears to be an error in the submission. After confirming the above, the meaning of the concluding sentence for the response "For the sake of simplicity and readability, the data obtained in GMP conditions were excluded from the revised manuscript." became apparent to this reviewer, for whom the use of labels 9 and 10 for Figures in the response letter was bound to cause confusion.

Though, as the authors state, DSB-related observations for p53 enrichment after editing were mainly made in cell lines, GSEA indicated p53-related signalling as differentially expressed between different DNA donor types in the current study and therefore already shone the spotlight on this pathway, so that it is gratifying to see suspicion of selection for p53 deficiency laid to rest for the established methodology here. In credit to the additional work done also for GMP-complaint conditions, I propose inclusion of what was called Figure 9 in the response letter as an additional Supplementary Figure (including also minor correction), as follows:

The material for proposed Figure 9 should be included in the Supplementary Materials as a new Supplementary Figure, correcting minor errors for the legend (HSCPs -> HSPCs [twice]; of p53 variant -> of p53 variants; according the -> according to the; the frequency of prevalence obtained -> the prevalence obtained; ns1042522 -> rs1042522) and correcting the donor IDs in legend and/or figure. In conjunction with reference to Figure 8e, the main text should then refer to that new Supplementary Figure. Likewise, the Figure 8e legend should refer to that new Supplementary Figure, at least in reference to prevalence of rs1042522 to panel b of that new Supplementary Figure.

(2) As indicated before, but maybe too obscurely, please, replace [630] “confirming that HbA expression was sufficient to inhibit the Hb polymerization” with “confirming that substituting HbS with HbA expression was sufficient to inhibit the Hb polymerization.” This is a minor change in the text but a major change in concept. Overexpression of β -globin and formation of HbA alone would not be therapeutic in SCA; it is instead the replacement of β^S alleles with β alleles at sufficient efficiency that prevents sickling. Arguably, at the editing efficiencies achieved, the vast majority of HSPCs would have at least 50% β alleles, i.e. would resemble HSPCs of an HbS carrier.

Further contributing to a confusion of concepts here for uninduced readers is reference to Supplementary Figure 4c data in lines [627] and [628] in the same sentence as showing “average frequency” instead of “relative concentration” for CE-HPLC HbS data (see Minor Comments below; by contrast, line [629] in describing sickling RBCs vs normal RBCs correctly refers to “frequency”).

Minor Comments

General remarks:

Indication of standard variations and the like for percentages should include the measure of variation as a percentage. This should be done throughout, either as e.g. “(41.5 \pm 5.3)%” or, preferably and more readily for the current text, as “41.5% \pm 5.3%” [lines 178-219, 240-254, 275-278, 289-292, 318-324, 352-371, 406-446, 618-629]

For the following, text in [square brackets] indicates line numbers (for the main article PDF) or section/figure labels (for supplementary materials), and {curved braces} indicate notes; other text indicates requested text changes (old -> new) or insertions.

[234][1107][1118][1119] HSCP-> HSPC

[518] cells lines -> cell lines

[518] differentiates cells -> differentiated cells

[521] for HBB locus -> for the HBB locus

[523] at HBB locus -> at the HBB locus

[525] engineer nuclease -> engineered nuclease

[612] impact of HSPCs fitness -> impact on HSPC fitness

[625] HSPCs from HbSS patient HSPCs translated -> HSPCs from HbSS patients translated

[627][628] average frequency -> relative concentration

[634] elicit positive -> elicit a positive {or: elicit positive outcome -> elicit positive outcomes}

[644] Conversions events -> Conversion events

[645] mice models -> mouse models

[648] been recently -> recently been

[1121] variant -> variants

[1123] according the -> according to the

[1126] ns1042522 -> rs1042522

NCOMMS-23-36573A

Non-viral DNA delivery coupled to TALEN gene editing efficiently corrects the sickle cell mutation in long-term HSCs. Moiani *et al*

REVIEWERS' COMMENTS

Reviewer #1 (Remarks to the Author):

The authors did an excellent work in responding to all issues and the paper can be accepted in my opinion. It is rigorous and solid and worth publishing.

We thank the reviewer #1 for acknowledging the quality of our revised manuscript.

Reviewer #2 (Remarks to the Author):

Summary

The authors have made painstaking efforts experimentally and in writing to address the numerous points raised by this reviewer in their first round of revisions. The manuscript has greatly improved in the process and pending the corrections indicated below will be a solid addition and point of reference for others.

We thank the reviewer #2 for acknowledging the quality of our revised manuscript.

Major Comments

(1) The response to reviewers in reference to potential p53 enrichment (Major Comment 1 for the original submission) refers to and shows a legend and panels 9a and 9b for a new Figure 9, and a new Figure 10. However, the new manuscript only contains 8 Figures, including image material and legend text, shown for Figure 10 in the response letter, as new panel 8e. The new manuscript or supplementary file do not refer to Figures 9 or 10 in their text, and text and image material for proposed Figure 9 are missing.

In our original first rebuttal letter, we decided to show a substantial amount of additional data to explain our results with a high degree of granularity and in a didactic manner. This approach was followed to make sure the reviewers had all relevant and factual elements to judge our work. Some of these additional datasets, including additional figures, were then evaluated for their potential to improve the clarity, readability and granularity of our manuscript as well as their capacity to broaden the impact of our work. Although most of the figures displayed in our rebuttal letter were incorporated in our revised manuscript, some were left aside for the sake of clarity. As an example, we found that the Figure 1 and Figure 9 of the rebuttal letter would complexify our message without bringing significant added value to our revised manuscript. We then clearly stated in the rebuttal that we didn't find relevant to add these figures in the main manuscript.

Moreover, proposed Figure 9 shows data for Donors 4 to 6, while the associated legend text in the response letter refers to Donors 3 to 5. I downloaded materials twice, initially assuming a confusion of file versions, but this appears to be an error in the submission. After confirming the above, the meaning of the concluding sentence for the response “For the sake of simplicity and readability, the data obtained in GMP conditions were excluded from the revised manuscript.” became apparent to this reviewer, for whom the use of labels 9 and 10 for Figures in the response letter was bound to cause confusion.

Though, as the authors state, DSB-related observations for p53 enrichment after editing were mainly made in cell lines, GSEA indicated p53-related signalling as differentially expressed between different DNA donor types in the current study and therefore already shone the spotlight on this pathway, so that it is gratifying to see suspicion of selection for p53 deficiency laid to rest for the established methodology here. In credit to the additional work done also for GMP-complaint conditions, I propose inclusion of what was called Figure 9 in the response letter as an additional Supplementary Figure (including also minor correction), as follows:

The material for proposed Figure 9 should be included in the Supplementary Materials as a new Supplementary Figure, correcting minor errors for the legend (HSCPs -> HSPCs [twice]; of p53 variant -> of p53 variants; according the -> according to the; the frequency of prevalence obtained -> the prevalence obtained; ns1042522 -> rs1042522) and correcting the donor IDs in legend and/or figure. In conjunction with reference to Figure 8e, the main text should then refer to that new Supplementary Figure. Likewise, the Figure 8e legend should refer to that new Supplementary Figure, at least in reference to prevalence of rs1042522 to panel b of that new Supplementary Figure.

Although we still believe adding this figure to the supplementary material section doesn't significantly improve the point made in the Fig. 8e of the main manuscript, we decided to comply with the editor proposal. This figure is now part of the supplementary material as Supplementary Fig. 7. The additional edits proposed by the editor are also incorporated in the main manuscript and in the supplementary materials.

(2) As indicated before, but maybe too obscurely, please, replace [630] “confirming that HbA expression was sufficient to inhibit the Hb polymerization” with “confirming that substituting HbS with HbA expression was sufficient to inhibit the Hb polymerization.” This is a minor change in the text but a major change in concept. Overexpression of β -globin and formation of HbA alone would not be therapeutic in SCA; it is instead the replacement of β S alleles with β alleles at sufficient efficiency that prevents sickling. Arguably, at the editing efficiencies achieved, the vast majority of HSPCs would have at least 50% β alleles, i.e. would resemble HSPCs of an HbS carrier.

This sentence is now revised in the discussion section.

Further contributing to a confusion of concepts here for uninduced readers is reference to Supplementary Figure 4c data in lines [627] and [628] in the same sentence as showing “average frequency” instead of “relative concentration” for CE-HPLC HbS data (see Minor Comments below; by contrast, line [629] in describing sickling RBCs vs normal RBCs correctly refers to “frequency”).

This point is now addressed.

In the same paragraph, we also added an information that was missing in our original revised manuscript. It summarizes the editing results obtained in BFU-E colonies. This new information appears in bold in the following sentence:

*“These results were confirmed in vitro in HSPC population (Fig. 4b and Fig. 6d, $44.3\% \pm 5\%$ and $49.3\% \pm 4\%$ HDR, $21.4\% \pm 2.3\%$ and $23.7\% \pm 1\%$ indels, average frequency \pm SD), **in BFU-E colonies (Fig. 4g, 58% of clones harboring mono and biallelic HDR and 9% of clones harboring biallelic indels)..”***

Minor Comments

General remarks:

Indication of standard variations and the like for percentages should include the measure of variation as a percentage. This should be done throughout, either as e.g. “ $(41.5 \pm 5.3)\%$ ” or, preferably and more readily for the current text, as “ $41.5\% \pm 5.3\%$ ” [lines 178-219, 240-254, 275-278, 289-292, 318-324, 352-371, 406-446, 618-629]

This point is addressed throughout the manuscript.

For the following, text in [square brackets] indicates line numbers (for the main article PDF) or section/figure labels (for supplementary materials), and {curved braces} indicate notes; other text indicates requested text changes (old -> new) or insertions.

[234][1107][1118][1119] HSCP-> HSPC

This point is addressed.

[518] cells lines -> cell lines

This point is addressed and also corrected in the supplementary materials section.

[518] differentiates cells -> differentiated cells

This point is addressed.

[521] for HBB locus -> for the HBB locus

This point is addressed.

[523] at HBB locus -> at the HBB locus

This point is addressed.

[525] engineer nuclease -> engineered nuclease

This point is addressed.

[612] impact of HSPCs fitness -> impact on HSPC fitness

This point is addressed.

[625] HSPCs from HbSS patient HSPCs translated -> HSPCs from HbSS patients translated

This point is addressed.

[627][628] average frequency -> relative concentration

This point is addressed.

[634] elicit positive -> elicit a positive {or: elicit positive outcome -> elicit positive outcomes}

This point is addressed.

[644] Conversions events -> Conversion events

This point is addressed.

[645] mice models -> mouse models

This point is addressed.

[648] been recently -> recently been

This point is addressed.

[1121] variant -> variants

This point is addressed.

[1123] according the -> according to the

This point is addressed.

[1126] ns1042522 -> rs1042522

This point is addressed.